# The Philosophy of Nature of the Natural Realism. The Operator Algebra from Physics to Logic

**Gianfranco Basti**

Faculty of Philosophy, Pontifical Lateran University – Vatican City 00120; basti@pul.va

**Abstract:** This contribution is an essay of formal philosophy—and more specifically of formal ontology and formal epistemology—applied, respectively, to the philosophy of nature and to the philosophy of sciences, interpreted the former as the ontology and the latter as the epistemology of the modern mathematical, natural, and artificial sciences, the theoretical computer science included. I present the formal philosophy in the framework of the category theory (CT) as an axiomatic metalanguage—in many senses "wider" than set theory (ST)—of mathematics and logic, both of the "extensional" logics of the pure and applied mathematical sciences (= mathematical logic), and the "intensional" modal logics of the philosophical disciplines (= philosophical logic). It is particularly significant in this categorical framework the possibility of extending the operator algebra formalism from (quantum and classical) physics to logic, via the so-called "Boolean algebras with operators" (BAOs), with this extension being the core of our formal ontology. In this context, I discuss the relevance of the algebraic Hopf coproduct and colimit operations, and then of the category of coalgebras in the computations over lattices of quantum numbers in the quantum field theory (QFT), interpreted as the fundamental physics. This coalgebraic formalism is particularly relevant for modeling the notion of the "quantum vacuum foliation" in QFT of dissipative systems, as a foundation of the notion of "complexity" in physics, and "memory" in biological and neural systems, using the powerful "colimit" operators. Finally, I suggest that in the CT logic, the relational semantics of BAOs, applied to the modal coalgebraic relational logic of the "possible worlds" in Kripke's model theory, is the proper logic of the formal ontology and epistemology of the natural realism, as a formalized philosophy of nature and sciences.

**Keywords:** quantum field theory; Kripke model theory; physical causality principle

## 1. Introduction: From Logic to Physics and Vice Versa

*1.1. A Methodological Premise: Mathematical Logic and Philosophical Logic*

The final aim of this contribution is to develop the formal ontology and epistemology of the *natural realism* (NR), as a formalized philosophy of nature and a formalized philosophy of science. They are interpreted, respectively, with the former as the ontology, and the latter as the epistemology of the modern natural and artificial science, the theoretical computer science (TCS) included, using the *category theory* (CT) as metalanguage of logic and mathematics. In this framework, the NR formal ontology is a categorical interpretation of the so-called *ontic structural realism* (OSR) approach to the philosophy of quantum physics (see [1,2]), or more generally of the *ontic interpretation* of the $\psi$-wave function (see [3] for an updated discussion).

Now, the proper *modal relational semantics* of the NR-formal ontology is that in it (the complex Boolean structures of), the propositional formulas of a descriptive ontology of the physical systems/processes can be validated "by homomorphisms up to isomorphisms" directly onto (the complex algebraic structures of) the mathematical models of the physical systems to which the ontological formulas descriptively refer. This depends, ultimately on the extension of the *operator algebra formalism* from physics to logic (see Sections 3.2 and 5.1), and then on the algebraic relational interpretation of the meaning function $[\cdot]$ in CT logic,

for which the extension of a complex formula $\varphi$ of the propositional calculus making it true, i.e. $[\varphi]$, is not defined by operations onto set-subset partial orderings such as in the set-theory (ST) logic, but primarily by operations onto a complex algebra (algebra-subalgebras) structure, in the common framework of the *operator algebra* formalism, extended from the mathematical physics to the Boolean logic, i.e., the so-called *Boolean algebra with operators* (BAO) (see [4,5] and below Section 5.1.4). All this is synthesized into the motto that "meaning is homomorphism" because meaning is based on a structure preserving mapping or homomorphism from the algebraic complex structure of a physical object in its mathematical model, onto the algebraic complex structure of the logic of a predicative sentence, in the descriptive language of ontology, which just because of this homomorphism is "referring to" or "signifying" this physical object [6].

Effectively, in this way, I want to emphasize the relevance of R. Goldblatt's suggestion synthesizing the main difference between the CT and the ST metalanguage in the slogan "arrows instead of epsilon" (see [7], pp. 37–74). Specifically, in ST, we suppose Russell's *set-elementhood* principle expressed in the *Principia* [1] for avoiding in axiomatic ST Frege's and Cantor's antinomies, and then we are supposing the predicate logic making of the *set-membership* relation $\in$ is a primitive in ST. On the contrary, in CT, we can formalize Peirce's pioneering intuition of a triadic algebraic construction of the predicate domains, making *morphisms* (arrows) the primitive of CT, with the consequent categorical notion of the set as *hom-set* (see Section 3).

The CT metalanguage is particularly suitable, therefore, for formalizing the constructive power of nature in constituting *dynamically* new domains of predication as it is required by an *evolutionary* approach not only in biology but also and primarily in cosmology. This is based on the universal mechanism of the (infinitely many) *spontaneous symmetry breakings* (SSBs) of the quantum fields at their ground state (i.e., the so-called *quantum vacuum* (QV) condition) in the *quantum field theory* (QFT), conceived as fundamental physics. This holds, both at the *microscopic* level of relativistic quantum physics of the *standard model* (SM) of elementary particles (see Section 3), and at the *macroscopic* level of the *condensed matter physics* of the chemical and biological systems (see [8–10] for a synthesis).

In this CT framework, the subcategory in the category **Set** of the *non-well-founded* (NWF) sets, violating the "set-elementhood" principle (see Note 1) because it satisfies P. Aczel's *anti-foundation axiom* [11] by which set *self-membership* is allowed, is particularly suitable for our aims. Specifically, for modeling in CT logic and mathematics the notion of *emergence* of new physical systems as a result of as many SSBs of the QV, i.e., as many *phase coherence domains* of the quantum fields at their ground state, which can be modeled in NWF-set theory as new "self-containing wholes", irreducible to the simple "combinatorics" of elements according to the famous expression "more is different" that was coined by the Nobel Prize Ph. Anderson precisely for characterizing any *phase transition* in fundamental physics [8].

Finally, both in ST and CT logics, the distinction holds between the *mathematical* and the *philosophical* logics that in its modern form is due to the American logician Ch. I. Lewis in his criticism of the application of the *extensional*, *truth-functional* mathematical logic of the *Principia* to the analysis of the philosophical, especially metaphysical, theories [12], thus criticizing *ante litteram* the core of Wittengstein's *Tractatus*. The philosophical logic is, indeed, the *modal logic* (ML), the logic of necessity and possibility, of "must be", and "may be" of which Lewis first proposed an axiomatic version by adding new modal *symbols* (essentially, the *necessity* $\square$ and the *possibility* $\lozenge$ operators) and *axioms,* respectively, to the alphabet and to the axioms of the standard propositional calculus of mathematical logic to define for the first time in the history of logic a formal *modal calculus* (MC) [13]. Therefore, by combining in a proper way the modal axioms, we can obtain as many *modal systems* as the proper syntax of different philosophical theories (see [12–14], for a complete presentation of the axiomatic approach to the MC, and Section 5.2.3 below for a partial exemplification). In this way, the distinction, and at the same time the relationship between

*mathematical* and *philosophical* logics, started to take its actual form using the rigor of the axiomatic method.

Indeed, saying that ML is the logic of necessity and possibility—a distinction per se that is meaningless in mathematical logic—means, using S. Kripke's many-worlds *modal relational semantics* [15,16], that in the *modal model theory*, we are dealing with truth or falsity of propositions not concerning only one state-of-affairs, or "actual world", as in the standard Tarskian model theory in mathematical logic [17], but also with truth or falsity in other *possible states-of-affairs* or "possible worlds" that possess *some relation* with the actual one. An approach that, also intuitively, is compliant with an evolutionary cosmology, based on the physical *causality principle* of the special relativity (SR) "light-cone" that holds both in general relativity (GR), and QFT (see below Section 1.2), and where, therefore, "cosmogony is the legislator of physics", according to J. A. Wheeler's intriguing statement about quantum gravity in cosmology [18]. Consequently, in ML, a proposition will be *necessary* in a world, if it is true in *all possible worlds related* to that world, and *possible*, if it is true *at least in another world*, relatively to the former one. This implies, of course, that in ML, the logical connectives (propositional predicates) are not *truth-functional,* at least in Frege's sense related to the usage of the *truth-tables* for the propositional connectives/predicates ("not", "and", "or", "if…then", …) [2].

To sum up, the *different meanings* of the modal operators correspond to as many *different semantics* and then to as many *truth criteria,* ruled by suitable axioms, for the interpretations of the MC, by which formalizing in a proper way, and then comparing, different philosophical theories, their consistency, and their effectiveness in solving the problems for which they were developed and defended by the respective supporters. Now, the main semantics of the MC generally admitted in ML are the following:

1.　The *alethic logics*, where the meaning of the modal operators is *possibly/necessarily true* in descriptive theories of the world states, in the different senses of the *logical,* and the *ontological* (physical and metaphysical) truth. Specifically, without confusing the *logical* (linguistic, abstract) and the *ontic* (causal, real) possibility/necessity, and their relationships. Historically, this distinction is the core of the classical Aristotelian philosophy and it was reintroduced in the contemporary analytic philosophy debate by S. Kripke at the end of the XX cent (see [19] and Section 6.2). Of course, the *onto*-logical alethic interpretation of MC is the proper logic of the *formal ontology.*

2.　The *epistemic logics,* where the meaning of the modal operators possible/necessary is related to different levels of knowledge *certainty*, and then to the distinction between *opinion/science* (*dóxa/epistéme*, in the Platonic language of the classic philosophy) [20–22]. Therefore, the necessity operator is interpreted in epistemic contexts as the "knowledge operator" **K**, and the possibility operator is here interpreted as the "belief operator" **B**. The possible worlds concerned here are the *believed representations* of the world relatively to a knowing (conscious) singular/collective *communication agent*, *x*. Additionally, the passage from "believing for *x* that *p*", **B**(*x,p*), to "knowing for *x* that *p*", **K**(*x, p*), depends on the satisfaction of a *foundation clause* **F**, i.e., $\mathbf{K}(x, p) \Leftrightarrow \mathbf{B}(x, p) \wedge \mathbf{F}p$, in the sense that the *sound* (true) beliefs or scientific knowledges are those founded in the real world. Of course, the clauses **F** will be different for different epistemologies, and for different underlying ontologies, which in this way can be rigorously compared and discussed (see [20] and Section 1.4).

3.　The *deontic logics*, where the meaning of the modal operators possible/necessary is related to different levels of ethical/legal *obligation*, and then the necessity/possibility operators of MC must be interpreted as the deontic operators of *obligation* **O**, and *permission* **P** [20,23,24]. The possible worlds concerned here, namely, the "ideal worlds" of the *ought to be,* as distinct from the "real world" of the *to be,* are those related to the ethical values or "goals" to be pursued. Or, more precisely, they are related with the axiological *optimality/maximality* criteria of "goodness" for actions to be satisfied according to the different ethical/legal systems. This means imposing ethical/legal constraints or "obligations" for the *effective pursuing* of the goals in the "real world"

by the human agents in terms of ethical optimality/maximality goodness constraints being satisfied [3]. Where, of course, the distinction between *moral* and *legal* obligations, and then between the *individual* and the *common* good(s) is fundamental [20]. From the standpoint of the history of philosophy, the distinction between the "alethic" and the "deontic" semantics of ML gives a formal foundation to the so-called "Hume problem" of the distinction between the "world of facts" ("to be": alethic logic) and the "world of values" ("ought to be": deontic logic), well known to the Middle Age logic but lost during the Renaissance and recovered by Hume. Moreover, in the case of the deontic obligatoriness being distinct from the logical necessity, the "possible worlds" *x* concerned are the *optimal states s of the world* (so introducing the "optimality operator" **Op** of the axiological logic (the "logic of values")), for a *given (individual, collective) subject x*, i.e., **Op** $(x, s)$ [4]. Therefore, the ethical obligatoriness expressed by the moral/legal norm *p*, i.e., **Ob** *p*, ruling the behavior for pursuing effectively in the real world a given optimal state *s* by *x*, i.e., **Ob** $p(x,s)$, satisfies the following axiomatic scheme: **Ob**$p := (\mathbf{Op}\,(x,s) \wedge c_a \wedge c_{ni}) \leftrightarrow \mathbf{Ob}\,p(x,s)$, where the two clauses $c_a$ and $c_{ni}$ express, respectively, the "condition of acceptance" by the individual/collective subject *x* of the optimal ordering **Op,** and the "condition of non-impediment" for *x* of effectively pursuing *s* in the real world [20].

4. Finally, in the MC semantics, it is possible to also formalize *intensional objects* and *predicates*, and not only intensional interpretations of modal operators, as we did till now, sometimes denoted as *individual concepts* ([14], p. 332). Generally, indeed, the "possible worlds" are modeled as classes of objects satisfying given modal rules. For this reason, MC is normally formalized in ST using **NBG** as its metalanguage but with the remembered restrictions and distinctions characterizing the different modal object domains [25]. However, it is also possible to model possible worlds by considering, for defining the truth evaluation functions of the modal semantics, the *individuals* within *a partition* of possible worlds of the universe (i.e., of the set of all possible worlds) considered. In this way, in the validation procedure, the *contingent identity* can also be considered, that is, *the identity of individuals* satisfying different predicates in different possible world partitions. In this sense, the ML semantics, because of its high flexibility, appears to be able to formalize the *intensional* (with "s") *logics* also in the sense of the *subject–object intentional* (with "t") *relationship* of the phenomenological inquiry [26]. Specifically, it expresses the *singular/plural first-person ("I"/"we") language* of individual/collective *intentional agents*, i.e., the "belief systems" of the different individuals and cultural groups in a society. This means that—against the dominating "relativism"—using the intensional logic formalization, it becomes possible to compare different visions of the world, in ontology, ethics, epistemology. . . , as far as each group, each "we", makes the effort of formalizing what they "intend" with their respective doctrines, i.e., in their "intensional logics". Then, according to the synthetic but effective account of John Searle, we can summarize by saying that the *intensional* (with "s") logic is also the proper logic of the cognitive, subjective *intentionality* (with "t") [27].

We can conclude, therefore, that the main distinction between *philosophical logic* and *mathematical logic* reduces to that between *modal (intensional)* and *extensional* logics, respectively [20,21], against the reductionist program of the early Neo-Positivistic approach to the philosophical analysis. Moreover, we must recall that the "philosophical logic" is not the same as the *philosophy of logic,* that is, the philosophical enquiry about the foundations of the formal (mathematical and philosophical) logic.

Finally—and this brings us back to the formal core of this paper—in addition to the early Lewis' *axiomatic* approach, and Kripke's *relational* approach to MC and ML, both based on the ST metalanguage, today, the more fruitful approach to MC and ML is the *algebraic* approach to Kripke's modal relational semantics that applies both to mathematical and philosophical languages in the framework of CT metalanguage. The algebraic approach

is, indeed, based on a categorical modal interpretation of BAOs. For this taxonomy of the different ML approaches, see [28] and Section 5 of this paper.

*1.2. The Logical Issue of Whichever Formal Ontology and Epistemology of Natural Sciences*

For our aims, the relevance of a categorical formalization of ML emerges clearly when we reflect on the main issue of whichever formal ontology and epistemology of the natural sciences. For this, we can refer to the teaching of W. V. O. Quine, and more specifically to his criticism of the axiomatic approach to ML developed by Ch. I. Lewis in its pretension of being the proper logic of ontology and metaphysics:

> What the resulting Lewis' systems describe are actually modes of *statement composition*—revised conditionals of a non-truth-functional sort—rather than implication relations between statements. If we were willing to reconstrue statements as names of some sort of entities, *we might take (metaphysical) implication as relation between those entities rather than between the statements themselves*; and correspondingly for equivalence, compatibility, etc. ([29] p, 32. Italics are mine).

In a word, what Quine is rightly vindicating as a proper foundation of the modal logic of the metaphysical *implication* (premise-conclusion) in a *formal* ontology and/or metaphysics is the necessity that the modalities of the *logical* relations between statements be able *to denote* ("to name") in some proper way the modalities of the *real* (causal) relations among the extra-linguistic entities, to which an ontological/metaphysical statement pretends to refer. However, Lewis' modal logic system is not able, in principle, to satisfy this requirement!

As I synthesized elsewhere [30–32] and we discuss at length in this contribution, the more direct and elegant way to satisfy Quine's deep requirement is to justify in a naturalistic ontology the *functorial dual equivalence* $\xleftarrow[\Omega^*/\Omega]{\rightleftarrows}$ in a categorical setting, between the *logical entailment* for which "it is impossible that the premise is true, and the consequence is false" ($\neg\diamond(\alpha^* \wedge \neg\beta^*)$) on the *logical* side of the descriptive statements of the ontological language, and the dual *causal modal entailment* "it is impossible the effect without its cause" ($\neg\diamond(\beta \wedge \neg\alpha)$) on the *ontic* side of the physical objects to which the descriptive statements refer. Here, the latter must be considered in some proper way as the semantic extension on which it validates *dually* the propositional formulas of the former.

As we see, this *dual relationship* between the *logical* and the *causal* entailments is the core of the Aristotelian theory of the *demonstrative syllogism* (premise → conclusion), where the soundness of its premise is founded dually by homomorphism on the conclusion of the *causal syllogism* (cause ← effect). This is a theory that can only be justified in the categorical framework of the theory of the *functorial bounded morphism* between Kripke models, respectively, on the ontic and on the logical sides of the NR formal ontology, as discussed in Section 5 (see Sections 5.1.3 and 5.2.3). We return to this specific point in the conclusive Section 6 of this work when we examine our theoretical proposal developed in this paper in a historical perspective. To conclude, it is worth emphasizing that this distinction between the causal and the logical necessitations (entailments) was reproposed in recent times by Kripke in his seminal *Naming and Necessity* book [19], even though it received its proper formal justification only in the CT framework of a coalgebraic semantics of Kripke model theory (see Section 5.2.3).

What indeed immediately excited the interest of scholars in Kripke's proposal is the evidence that the causal entailment, in Kripke's many worlds *relational semantics,* appears to be the ML version of the *causality principle* in fundamental physics based on the so-called "light-cone" of special relativity (SR). In fact, and it is important to recall this in our context, this causality principle holds, both for the three quantum interaction force fields of the relativistic QFT, and for the gravitational force field of general relativity (GR), as one of the most distinguished theoretical physicists of our time, the 1979 Nobel Laureate in Physics Steven Weinberg (1933–2021), also recently pointed out. He, indeed, in his last published book dedicated to the *Foundations of Modern Physics,* in the paragraph about the *causality* in fundamental physics, stated (see Figure 1):

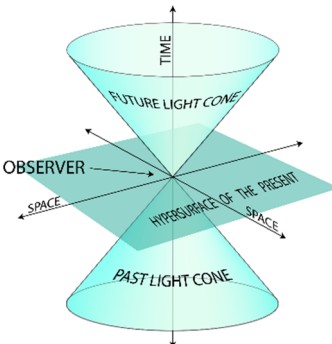

**Figure 1.** Intuitive representation of Minkowski's light-cone of special relativity, which is defined on complex numbers, where the future-directed states are defined on real numbers and the past-directed states are defined on imaginary numbers.

> We saw (…) that no Lorentz transformation acting on a body at rest could give it a speed greater than *c*, the speed of light. We can derive a stronger result, that no influence whatever can travel faster than light. This is not just a confession of technological inadequacy, but a consequence of an assumption of causality, that *effects always come after causes*" ([33], pp. 121–122 (italics mine)).

Intuitively—but overall formally (see Section 5.2.3)—it is evident that Kripke's many-world relational semantics is the proper model theory of the causal light-cone granting *a dynamic partition criterion* among the possible world-states in terms of their "causal accessibility" from/to a given past/future physical event. Moreover, this evidence acquires a precise formal justification in the categorical formalization of the relativistic quantum physics (QFT) when we reflect upon the evidence that the "causal relations" from/to past/future events satisfy the dual definition of *morphisms* (arrows) from/to an *initial/terminal object*, characterizing, respectively, the categories of *algebras* and *coalgebras* in CT logic and mathematics (see Section 2.4 and especially Definition 7. and Note 12).

Particularly, it is worth emphasizing from the ontological standpoint that, when we consider the "future light-cone" on the cosmological scale of the *universe evolution* by which it populates progressively itself of ever more complex objects and structures in the "hot big-bang hypothesis", because of the strongly non-linear character of the causal processes involved (symmetry breakings) [10], the *logical/mathematical unpredictability* of the "effects" (future events) as to their common "cause" implies that the only morphisms that logically make sense are those from the effects as to their common cause. It, therefore, categorically plays the role of the common *terminal object,* to which all the "arrows" relating the effects to their cause are directed. This notation emphasizes the coalgebraic nature of the "future" light-cone, and then the "coinductive" nature of the "causal entailment" (see Section 5.2).

Only from this simple reflection does the *mathematical* and *logical* relevance of the *category of coalgebras (coproducts)* functorially represented appear, which we discuss in Section 2.4, but also their *physical* relevance, which we discuss in Section 4.5. Indeed, the (Hopf) coproducts and coalgebras play a fundamental role in the QM and QFT calculations over lattices of quantum numbers. In fact, in this case the coproducts—in terms of summations for calculating the total energy of a superposition of particles (fields) in a quantum state (phase)—are the fundamental way for knowing how many and which type of particles are superposed in a quantum state. Effectively, this is how many and which type of matter fields (of which the "elementary particles" or "fermions" are their quanta) stay in a coherent phase. In this case, for coming back to the causal light-cone, its "local accessibility relations" among physical states (phases) in the physical space-time become as many *phase transitions allowed* among quantum fields in QFT. We present in Sections 3 and 4 a sketch of the categorical formalization of the representation theory of these phase transitions (or unitarily inequivalent representations of the quantum fields dynamics) in QFT, according to its different interpretations and models.

Therefore, coming back to our philosophical discussion, a possible significant solution of Quine's conundrum about the same possibility of a formal ontology on a naturalistic basis (effectively, a formal ontology and epistemology of QFT as fundamental physics) is given in the framework of the CT logic. Namely, according to a categorical *(co-)algebraic relational semantics* of the *meaning function* [·] mapping a formula [*φ*] of the Boolean propositional calculus into its coalgebraic *extension φ, validating* "making *φ* true". This relational semantics was inaugurated by Jónsson's and Tarski's application to Boolean logic of the *operator algebra* formalism, already extensively applied in quantum and classical physics, i.e., the so-called *Boolean algebra with operators* (BAO), in the framework of the celebrated Stone's *representation theorem for Boolean algebras* (RTBA) (see Section 5.1 and Appendix B). Indeed, the *topologies* of RTBA in logic and quantum physics *are ultimately the same,* so that RTBA is the theoretical foundation of any possible bridging between physics and logic, and then of any possible formal ontology of quantum physics.

Without anticipating here all the passages of the argumentation given in Sections 4 and 5 of this paper, we can emphasize two essential points. Before all, it is possible to demonstrate in CT logic the completeness of Kripke's relational semantics using a coalgebraic interpretation over trees on NWF-sets defined in the Stone spaces, in the framework of the *functorial dual equivalence* between the category of Stone coalgebras and the category of the modal Boolean algebras with operators (MBAO), **SCoalg(Ω) ≃ MBAO(Ω\*)** [6]. Secondly, it is possible to extend this categorical dual equivalence to the category of the Hopf coalgebras in physics for the Bogoliubov functor *B* **qHCoalg(*B*)** [34].

Effectively, this relational semantics is only a significant application of the more general principle characterizing the CT logic, according to which "a statement *α* is *true* in/about a category C if and only if its dual *α*^op (i.e., obtained from *α* by reversing all the arrows and their compositions) is *true* in/about the opposed category C^op" (see Section 2.3).

On the other hand, all this offers a categorical solution not only for Quine's *ontological* conundrum but simultaneously for the other modern conundrum of the justification of the *soundness* of the premises (sufficient conditions) in the hypothetical reasoning, afflicting the epistemology of modern sciences from Galilei to Popper (see Section 6.2). Indeed, Karl R. Popper (1902–1994) so synthesized the problem in his masterpiece *The logic of scientific discovery* (1935):

> If we distinguish, with Reichenbach, between a 'procedure of finding' and a 'procedure of justifying' a hypothesis, then we have to say that the former—the procedure of finding a hypothesis—cannot be rationally reconstructed. Yet the analysis of the procedure of justifying hypotheses does not, in my opinion, lead us to anything which may be said to belong to an inductive logic. For a theory of induction is superfluous. It has no function in a logic of science ([35], p.307).

Evidently, in the light of what we just said, we try to demonstrate in this contribution that there exists in CT logic a rational procedure for justifying the *soundness* of the hypothesis in the hypothetical deductive method of modern sciences.

*1.3. A Scheme of this Paper*

In the Section 2, I summarize some elements of CT as a formal metalanguage of the mathematical and logical theories in a systematic comparison with set theory (ST), with which philosophers (and physicists) are generally more acquainted than with CT. Particularly, I emphasize that CT is particularly suitable as a formal metalanguage of the *operator algebra* and then of the *topological* approach in mathematics and physics, and logic and computer science because of the development of the so-called BAO by Jónsson and Tarski [4,5] in the framework of the celebrated Stone's *representation theorem for Boolean algebras* (RTBA) [36] (see below Section 5.1 and Appendix B).

In Section 3, I summarize some elements of the QM formalism, particularly the completion of the original Von Neumann formalization of QM, via the so-called *Gelfand–Naimark–Segal (GNS) construction* that inaugurated the operator algebra approach to QM. Afterward, I present some basic notions of the QFT formalism in the framework of special

relativity theory (SR), according to the original Dirac's interpretation of QFT as a *second quantization* (SQ) with respect to QM.

In Section 4, I summarize the core of the extension of the QFT system representation theory to the modeling of quantum dissipative systems (or dissipative QFT) persistently in far-from-equilibrium conditions because of passing through different phases. This is based on the *Bogoliubov transform* mapping between different phases of fermionic and/or bosonic quantum fields, both in the relativistic QFT (at the physical *microscopic* level) and QFT of the condensed matter physics (at the *macroscopic* level of the chemical and biological systems). Because of the necessary *non-commutative* character of the Hopf algebra *coproducts* in calculations over lattices of quantum numbers in the case of open systems [5], the mathematical formalism of dissipative QFT implies the necessity of the *algebra doubling*, and then of the doubling of the *state (phase) spaces*, and finally of the same *Hilbert spaces* for recovering the canonical (closed) *Hamiltonian representation* of the total system.

This is obtained by also inserting in the Hamiltonian—through the method of the algebra doubling—the thermal bath degrees of freedom, with which any quantum dissipative system is necessarily *entangled*, so to grant a far from equilibrium *energy balance*, and the "closed" character of the resulting system as required by the Hamiltonian "canonical" representation.

The main mathematical result of this approach is then the so-called principle of the *doubling of the degrees of freedom* (DDF), by which it is possible for the system itself "to decide dynamically", which is the proper finite number of the degrees of freedom of the statistical expectations in the Hamiltonian for a faithful representation of the system dynamics.

This means that the ground state of the quantum fields in the dissipative QFT, i.e., their *QV condition*, because it is necessarily at a temperature $T > 0$, allows *different phase-coherence domains*, non-interfering with each other, to coexist in the same balanced (0-summation free energy) ground state of the quantum fields. Each of these phase coherences of the quantum fields at their ground state—according to the fundamental *Goldstone theorem*—corresponds to a *spontaneous symmetry breaking* (SSB) of QV, from which new properties in physical systems emerge. Each SSB corresponds indeed to the spontaneous instauration of *long-range correlations* among quantum fields at their ground state (QV), and it is therefore univocally indexed by the *unique value* $\mathcal{N}$ of the condensate of the so-called *Nambu–Goldstone* (NG) bosons, i.e., the quanta of the long-range correlations among the quantum fields.

Because of the stability of these collective modes of quantum fields that do not require any further energy contribution since all coexist at the same balanced ground state (0-sum energy) of a dissipative system, it is possible to justify a *dynamic partial ordering* of them. All this is the core of the *QV-foliation* principle that can be formalized in CT using the *colimit* operation (see Appendix A), which therefore appears to be the fundamental tool used by nature for generating *complex systems* and for justifying at its fundamental physical level the notion of *memory* in biological and neural systems [34,37,38].

Effectively, in biology and neurosciences, the QV-foliation allows the proposal of an original solution of the debated issue of the *long-term memories* in mammals' brains, modeled as *dissipative brains* entangled (balanced) with their environment (i.e., with the rest of the body, and through it, with the outer environment), in the framework of the *intentional* interpretation of cognitive tasks [39,40]. Intentionality has its biological foundation, therefore, in the *homeostasis* characterizing all living systems as dissipative systems, according to A. Damasio's original proposal [41]. In this way, this neurophysiological application becomes one of the main empirical supports of QFT as the fundamental physics of biological systems.

In Section 5 of this paper, therefore, to arrive at the presentation of the coalgebraic foundation of the Kripke modal relational semantics as the proper logic of the NR formal ontology and epistemology, we start from a synthetic illustration of the momentous Stone's *representation theorem for Boolean algebras* (RTBA: see also Appendix B). From this the consequent development of BAOs derives, and then a *relational semantics* based on the algebraic interpretation of the meaning function in CT logic. In the CT approach to ML, this is based

on the *dual equivalence* between the category of the coalgebras of NWF-sets defined in Stone spaces, **SCoalg,** and the category of the modal Boolean algebras with operators **MBAO,** for the contravariant application of the Vietoris functor $\mathcal{V}$. This constitutes the core of the coalgebraic justification of Kripke's modal relational semantics in CT logic [6,42,43]. Now, starting from the evidence that the Stone spaces in logic are the same topological spaces of the C*-algebras of Hilbert spaces in physics (see [44,45], and Appendix B), it is possible to define the Kripke relational semantics of NR-formal ontology directly in the category of physical coproducts for the contravariant application of the Bogoliubov functor $\mathcal{B}$ [34]. The Vietoris coalgebraic construction, indeed, grants—as the Bogoliubov functor $\mathcal{B}$ does *dynamically* via the DDF construction in the category of coalgebras for QFT systems—a *selection criterion of admissible sets* on which the semantics of the Boolean modal algebras are defined, analogously to the ultrafilters of Stone's RTBA. In both cases, indeed (the *physical* one (Bogoliubov) and the *logical* one (Vietoris)), the set indexing is performed by the *colimit operation* over categories of coproducts on NWF-sets. For this reason, we can write the categorical dual equivalence that is the core of the modal logic of the NR-formal ontology: **SCoalg**$(\mathcal{B}) \simeq$ **MBAO**$(\mathcal{B})^*$, just as in logic, we write **SCoalg**$(\mathcal{V}) \simeq$ **MBAO**$(\mathcal{V})^*$ [6].

Particularly, it is worth emphasizing that in the case of Kripke relational structures defined on NWF-sets, only *local modal truths* are allowed by the powerful notion of *functorial bounded morphism* between Kripke models, respectively, on the logic ($\mathfrak{M}$) and on the physical ($\mathfrak{M}'$) side, i.e., $\mathfrak{M} \underset{\rightleftarrows}{} \mathfrak{M}'$, as we illustrate in Sections 5.1.3 and 5.2.3. This semantics is indeed exactly what we need for formalizing a descriptive ontology of an evolutionary cosmology where "cosmogony is the legislator of nature".

Finally, the concluding Section 6 is dedicated to two fundamental metaphysical and epistemological issues to which the NR formal ontology could suggest a solution. At the beginning of the Modern Age, Immanuel Kant in his famous booklet *Prolegomena to any Future Metaphysics that will be able to come forward as a Science* [46] published in 1783, even though it was originally conceived as an Introduction to Kant's masterpiece *Critique of the Pure Reason*, stated that the future of a naturalistic metaphysics as science will pass necessarily through a new foundation of the *causality principle* in physics and metaphysics. Indeed, the modern Galilean and overall Newtonian physics, of which Kant's *Critique* wanted to constitute the epistemology, confuted the Aristotelian and Scholastic causal view of nature, and the causal justification of its laws. At the same time, it confuted the core of the Aristotelian epistemological realism, for which the *logical relations* among *objects* in reasoning (i.e., in the language of mind), *depend on*, and then *refer to* because *abstracted from* the *causal* (*real*) *relations* among *things* in nature. We summarize in which sense the categorical duality between the causal and logical modal entailments presented since the beginning of this *Introduction* (see Section 1.1), and formally justified in the rest of this paper, is in continuity with the *relational natural realism* of the NR formal ontology (see Section 1.4) in the framework of the CT logic presented in this contribution.

### 1.4. A Taxonomy of the Different Formal Ontologies in Western Thought

As a conclusion of this *Introduction* devoted to illustrating the theoretical and historical background of my proposal of a renewed philosophy of nature as a formal ontology of the natural sciences, let us sketch briefly which are *the main formal ontologies* in the history of Western thought to immediately locate my proposal in this schematic survey.

Today, the term "formal ontology" is widely used in the computer science environment, particularly in the so-called knowledge engineering realm for the development of *semantic databases*. In this sense, ontologies refer to the fundamental conceptual categories by which different linguistic groups organize their knowledges about the objects of their specific environments, that is, their representations of reality. It is often forgotten, however, that this usage of the term "formal ontology" in computer science refers implicitly to the origins of this term in the phenomenological philosophy [47,48].

Historically, indeed, Edmund Husserl introduced the terms "formal ontology" in the contemporary philosophical jargon for signifying the *ante-predicative foundation* of predicates

in formal logic that he developed in his *transcendental logic,* based on the notion of the *intentional transcendental subject* [26] [7]. Specifically, against the *formalism* of René Descartes' *cogito,* and Immanuel Kant's *Ich denke überhapupt* ("I think in general"), which made the *self-conscious evidence* of the pure thinking the *conceptualist* foundation of the logical truth. In his criticism of the epistemic formalism of Descartes and Kant, Husserl, following his teacher Franz Brentano [49], vindicated that any psychical act as such (believing, thinking, willing, sensing...) is *evidently* characterized by an intrinsic *aboutness* or "reference to an object". In this sense, the pure *cogito* cannot exist or the Kantian "I think in general" since "I/we think (believe, will, sense...) always *something*", i.e., a given object. Conversely, no object can exist in logic or mathematics, according to Husserl, without supposing an implicit reference to a knowing (individual/collective) *subject.* To sum up, the modern principle of evidence has an intrinsic *intentional constitution,* based on the *transcendental relationship subject–object.*

Effectively, in the *Third Logical Investigation*, Husserl defends this *ontological* foundation of the *logical* truths because knowledge can access *real* beings/things only as objects-for-a-subject". Particularly, in the "Introduction" to this *investigation,* Husserl refers to the notion of *formal ontology* as the "*pure (a priori) theory of objects as such*" (see [50], p.3).

Indeed, this reference to the ontology, because of his criticism of the *formalism* typical of the modern "reshaping" of mathematics by the *axiomatic method* ([51], pp. 21–23), constitutes the main motivation of Husserl's phenomenological method since the very beginning of his career. Namely, since his PhD work (1891) in mathematics concerning the "calculus of variations", Husserl introduced the notion of *Inhaltlogik.* This is the "logic of contents", or "intensional logic", as he denoted it, for correcting the formalistic, purely syntactic nature of the calculus in modern extensional logic and mathematics [52], according to Frege's *Begriffsschrift* [53].

Now, in this light, it is important to compare Husserl's and Peirce's criticisms they independently made about Ernst Schröder's first volume of his treatise on the *Algebra of Logic* [54], published in 1890, which was the first historical proposal of a mathematical logic, before Frege's logistic or predicative one, based on his logic of classes [55]. A comparison between Husserl and Peirce is relevant for us because both agree independently about the insufficiency of Schröder's *dyadic* algebra of logic for justifying a satisfactory theory of *signifying* in logic. However, while Husserl vindicated the necessity of the reference to *an intentional subject* for giving the algebraic formulas of Schröder's calculus the capacity of signifying something [56], Peirce, in his famous review paper on Schröder's book, *The Logic of Relatives* [57], introduced the necessity of an irreducible *algebraic triadic relation* for "signifying" the dyadic relation between subject–predicate in the linguistic tokens. In other papers, Peirce denoted this third term of the "semiotic" (signifying) relation as an *interpretant*, with a neologism invented for excluding—against any conceptualist view on the foundations of logic—any necessary reference to a knowing subject, or *interpreter,* for justifying the capacity of signifying a predicative formula in logic. In Peirce's words:

> This definition [of semiotic relation] no more involves any reference to human thought than does the definition of a line as the place within which a particle lies during a lapse of time ([58], p. 52).

On this *triadic algebra of relations* is based, therefore, Peirce's semiotic notion of *sign* as a being-for (*esse per*, in Scholastic Latin) a third term, by which the dyadic relation of being-to (*esse ad*) between the two terms subject–predicate of a predicative relation acquires the capacity of *signifying*. On this theory, Peirce's famous theory of the three *semiotic categories* of "firstness", "secondness", and "thirdness" [59] is also based. These are *ante-predicative algebraic categories,* in the sense that any classical predicative theory of *logic* categories (i.e., intended as the most general and then irreducible predicates of a given language) supposes these three semiotic *algebraic* categories.

We see in Section 5.1 how, through the axiomatization of Peirce's naïve algebra of relations into an *axiomatic calculus of relations,* by A. Tarski (see below Note 11) and then the development by the same Tarski of a BAO with its algebraic relational semantics (see

Section [5.1] and Appendix [B]), Peirce's pioneering work is in the background of whichever ontology of the *relational natural realism,* my NR-formal ontology included.

Given this necessary historical background, let us now illustrate shortly a taxonomy of the *main formal ontologies* proposed in the history of Western thought. This synthesis is inspired by a similar one developed by my colleague and friend Nino B. Cocchiarella [60,61], a logician and philosopher of logic, now Emeritus at the Philosophy Dept. of the Indiana University at Bloomington (USA). What I share with him—apart from some significant differences—is the general idea that the main ontologies of whichever philosophy and culture can be interpreted, in formal philosophy, like many *theories of predication,* as far as predication is not reducible to the only class/set membership relation ∈. The main theories of predication are, indeed, in the history of logic, the *nominalism,* the *conceptualism,* and the *realism,* which historically can be viewed like many *theories of universals.* By "universal" we intend, again with Cocchiarella, "what can be predicated of a name", according to Aristotle's classical definition (*De Interpretatione,* 17a39).

To sum up [60–62], we can *synchronically* distinguish along the centuries of the (Western) history of thought (generally distinct into Ancient, Middle, and Modern Ages) at least *three types of ontologies,* with the last one subdivided into two others (see Figure [2]). For each of these subdivisions, I quote indicatively in parenthesis some authors, who belong indifferently to one of the three main ages of the Western tradition [8].

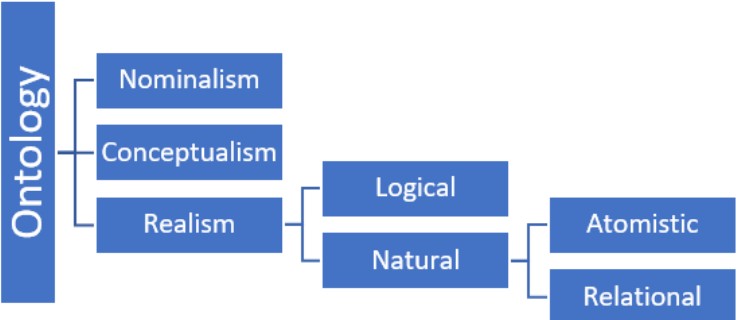

**Figure 2.** Scheme of the main ontologies in the Western tradition.

1.  *Nominalism*: the predicable universals are reduced to the predicative expressions of a given language that, *by its conventional rules*, in the referential usages of predicative sentences, determines the truth conditions of the ontological propositions (Sophists, Roscellinus, Ockham, Hobbes, Quine, etc.).
2.  *Conceptualism*: the predicable universals are expressions of *mental concepts*, so that the *laws of thought*, in the referential usages of predicative sentences, determine the truth conditions of the ontological propositions (Descartes, Leibniz, Kant, Husserl, Stein, etc.).
3.  *Realism*: the predicable universals are expressions of *properties and relations* existing independently of the linguistic and/or mental capacities in:
    a.  *The logical realm*: we have, therefore, the ontologies of the so-called *logical realism*, where the *logical relations*, in the referential usage of predicative sentences, determine the truth conditions of the ontological propositions, independently of human linguistic and mental capacities (Plato, Guillaume de Champeaux, Frege, Russell, Fraenkel, Gödel, etc.).
    b.  *The physical realm*: we have then the ontologies of the so-called *natural realism*, or "naturalism". In turn, naturalism can be of two types:
        - *Atomistic*: without natural kinds, where the *logical-mathematical laws* with their *empirical fulfilment* by measurements on physical events, in the referential usages of predicative sentences, determine the truth conditions of the ontological propositions (Democritus, Newton, Laplace, Wittenstein's *Tractatus*, Carnap, etc.).

- *Relational*: with "natural kinds", where the *real relations* (causes) among "things" in nature determine the *logical relations* among "objects", in the referential usages of predicative sentences in language, and then determine the truth conditions of the ontological propositions (Aristotle, Aquinas, Poinsot, Peirce, Kripke, NR, etc.).

## 2. Some Elements of the Category Theory and Its Relational Semantic in Logic

As we anticipated, this section is devoted to acquainting philosophers with the basic notions of CT, discussed in their relationships with the correspondent notions in ST. The strong interdisciplinary character of formal philosophy is even more evident when we consider the actual *algebraic formalization* of ML in the context of CT logic and mathematics [28], by which the very same algebraic relational structures appear to be at the *common roots* of the mathematical and the philosophical logics. For my synthetic exposition, I refer essentially to [63], which is addressed explicitly to introduce physicists and philosophers into CT, while I refer to [64] and [65] as two CT textbooks addressed mainly to professional mathematicians and computer scientists.

### 2.1. The Ante-Predicative Definition of Category in Category Theory

As we recalled since the beginning of this contribution, the proper formal character of the CT metalanguage as to the (standard) ST metalanguage consists in not taking $\in$ of the set-membership as a primitive, so to limit the *constructive* approach in logic and mathematics to the *inductive* one, extended to infinite sets (transfinite induction), based on Von Neumann's "cumulative hierarchy of *ranks* of ordinals" and then on Zermelo's "well-ordering theorem" because of the "foundation axiom" in **ZF(C)** set theory [66]. In this sense, given the strict dependence of ST on the predicate logic, which is the deep reason underlying the fact of taking $\in$ as a primitive, CT can be defined as an algebraic ante-predicative theory on the foundations of logic and mathematics. Therefore, in the following exposition, I compare systematically some basic CT notions with the corresponding set-theoretic ones, with which we are more acquainted, to emphasize the differences and contact points. Of course, this is without supporting any non-sensical opposition between ST and CT in the foundations of logic and mathematics.

Indeed, it is well-known that it is possible to interpret CT at the foundational level within **NBG** set theory, even though not within **ZF** because of the presence of "large" categories requiring "classes" with a cardinality greater than *V* ("large cardinals") and then supposing Gödel's "generalized continuum hypothesis". Nevertheless, what is evident is that CT, initially meant to organize certain fields of mathematics in a systematic way (such as algebraic topology and homological algebra), categories soon became objects of study in their own right ([45], p. 805).

What I want to emphasize in this work is that the CT metalanguage allows not only the working mathematician, as S. Mac Lane suggested [64], but also the working philosopher, as S. Abramsky first suggested [67,68], to discover and formalize axiomatically *structural* similarities between theories; in our ontological case, between logical and physical theories, in which an exclusive "predicative" interpretation of the category notion that takes the $\in$ of the membership relation as a primitive would be forbidden as an inconsistent "category jump".

Indeed, in CT, the *primitives* are:

1. *Morphisms* or *arrows*, *f*, *g*,—intended as a (purely relational) generalization of notions such as "function", "operator", "map", etc.
2. The i*dentity arrow*, such that, for any object *A*, there is an identity arrow or reflexive morphism $\mathrm{Id}_A = \mathbf{1}_A \colon \mathrm{A} \to \mathrm{A}$ [9].
3. Two *maps* or *operations* from arrows to objects, $\mathrm{dom}(\cdot)$, $\mathrm{codom}(\cdot)$, assigning a domain or *source* and a codomain or *target* to each arrow.
4. The *compositions of arrows*, written as *g*, *f*, or *f* o *g*, in which the codomain of *g* is the domain of *f*, that is, for any three objects *A, B, C* in the theory, there exists a morphism

composition $f \circ g$, that is, $A \xrightarrow{g} B \xrightarrow{f} C = A \to C$, satisfying a *transitive* property among arrows.

These primitives must satisfy two axioms regulating compositions and identities among morphisms by which domains and codomains match appropriately:

**Axiom 1.** *(Associativity Law):* $h \circ (g \circ f) = (h \circ g) \circ f.$

**Axiom 1.** *(Identity or Unity Law):* $f \circ \mathrm{Id}_A = f = \mathrm{Id}_B \circ f.$

Therefore:

**Definition 1.** *(Category, C): Any structure-preserving collection of «arrows» (or «morphism»), «objects», «compositions», and the two «mappings» dom(f), cod(f), assigning to each morphism f its domain-codomain of objects, and satisfying associativity and identity, constitutes a category C in CT.*

In this way, it becomes possible to locate the algebraic, ante-predicative notion of category among the other algebraic structures more used by the working mathematicians with their defining axioms, according to the following Table 1.

**Table 1.** Main algebraic structures with respect to their defining axioms.

| | | Closure | Associativity | Identity | Invertibility | Commutativity |
|---|---|---|---|---|---|---|
| ▪ | Semigroupoid | *Unneeded* | Needed | *Unneeded* | *Unneeded* | *Unneeded* |
| ▪ | Category | *Unneeded* | Needed | Needed | *Unneeded* | *Unneeded* |
| ▪ | Groupoid | *Unneeded* | Needed | Needed | Needed | *Unneeded* |
| ▪ | Magma | Needed | *Unneeded* | *Unneeded* | *Unneeded* | *Unneeded* |
| ▪ | Quasigroup | Needed | *Unneeded* | *Unneeded* | Needed | *Unneeded* |
| ▪ | Loop | Needed | *Unneeded* | Needed | Needed | *Unneeded* |
| ▪ | Semigroup | Needed | Needed | *Unneeded* | *Unneeded* | *Unneeded* |
| ▪ | Inverse Semigroup | Needed | Needed | *Unneeded* | Needed | *Unneeded* |
| ▪ | Monoid | Needed | Needed | Needed | *Unneeded* | *Unneeded* |
| ▪ | Group | Needed | Needed | Needed | Needed | *Unneeded* |
| ▪ | Abelian Group | Needed | Needed | Needed | Needed | Needed |

Furthermore, if we add the algebraic notion of *homomorphism* as a *structure-preserving mapping* between algebraic structures—not to be confused with the notion of *homeomorphism* denoting an *isomorphism* (i.e., an invertible homomorphism) between topological spaces—

we can give the following examples of typical categories in mathematics useful for our aims, each characterized by specific objects and specific arrows:

- **Set** (sets and functions);
- **Grp** (groups and homomorphisms);
- **Mon** (monoids and epimorphisms), where "monoids" are "one-object categories" and "epimorphisms" are the categorical counterpart of "surjective functions" in ST;
- **Top** (topological spaces and continuous functions/paths);
- **Vect**$_k$ (vector spaces defined on a numerical field $k$ and linear functions).

Moreover, in CT, the formal tool for *calculating* and *demonstrating* and then to grant *universality* and *truthfulness* to CT constructions are the *commutative diagrams* of the algebraic calculus of relations. In this way, to continue with Abramsky, the "arrow-theoretic" way of reasoning consists essentially in a *diagrammatic way of reasoning* [63] (p. 10) [10]. Following step by step his useful exemplification, it is asserted that the equations $g \circ f = h$ and $g \circ f = k \circ h$ correspond to the *commuting triangle* and *commuting square diagrams,* respectively, which are the basic commutative diagrams in CT, i.e.,

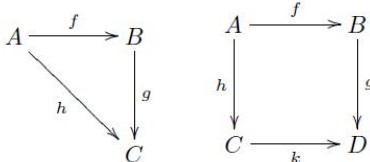

Similarly, the two equations asserting the "associative" and the "identity" laws above, i.e.,

$$h \circ (g \circ f) = (h \circ g) \circ f; \; f \circ \mathrm{Id}A = f = \mathrm{Id}B \circ f$$

characterizing a *category C* in universal algebra (e.g., *groups* also satisfy "closure" and "invertibility" axioms) can be expressed by the two diagrams below, respectively:

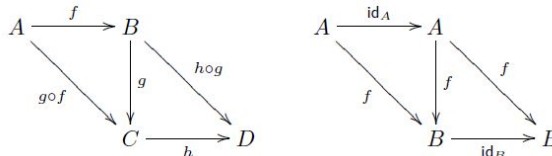

The definition of the commutative diagram can be the following one, which is a simplified version of the rather cumbersome one given by Abramsky and Tzevelekos ([63], p. 11):

**Definition 2.** *(commutative diagram): A commutative diagram in a category C is a directed graph, whose nodes are objects in C, and edges are morphisms in C. This diagram commutes, if any, two paths with a common source and target that are equal, where at least one of them has a length greater than 1. Specifically, given paths:*

$$A \overset{f_1}{\to} C_1 \overset{f_2}{\to} \ldots C_{n-1} \overset{f_n}{\to} B \; \text{and} \; A \overset{g_1}{\to} D_1 \overset{g_2}{\to} \ldots D_{m-1} \overset{g_m}{\to} B$$

*if max(numb) > 1, then $f_n \circ \ldots \circ f_1 = g_m \circ \ldots \circ g_1$. This commutativity property immediately grants the uniqueness of the diagram concerned, and then the universality of a diagrammatic demonstration.*

**Remark 1.** *From the historical standpoint, the commutative diagrams in CT are the formalization in the framework of Tarski's axiomatic algebraic calculus of relations [69] of the naïve intuition of Peirce, who first introduced diagrams as a calculation tool in the earliest stage of the algebra of relations he inaugurated. Moreover, the evidence that the commuting triangle satisfying the equation $g \circ f = h$ is the more fundamental diagram confirms Peirce's intuition that the triadic relations, and not the dyadic ones, are the irreducible relations in algebra. Specifically, they are the basic structure of "semiotics", i.e., of any "signifying" structure in logic and mathematics [11].*

*2.2. The Categorical Definition of Sets as Hom-Sets*

If all this justifies Abramsky's intriguing statement that "we will refer to any concept which can be defined purely in terms of compositions and identities as *arrow-theoretic*" ([67], p. 3), this perspective change is made explicit when we consider the categorical notion of set as *hom-set*. Indeed:

> Category theory can be seen as a "generalized theory of functions", where the focus is shifted from the pointwise, set-theoretic view of functions to an abstract view of functions as *arrows* ([67], p. 8).

In fact, given in a category $\mathcal{C}$ the two collections of arrows (or morphisms), Ar($\mathcal{C}$), and objects, Ob($\mathcal{C}$), characterizing the category definition, we can also define the arrow-theoretic notion of *hom-set* for a category *C*, where the prefix *hom-* stays for *homomorphism,* i.e., a structure-preserving mapping between pairs of objects, as the arrow-theoretic interpretation of a "function". Specifically, for each pair of objects $A, B \in$ Ob ($\mathcal{C}$), we define the set:

$$\mathcal{C}(A, B) := \{f \in Ar(\mathcal{C}) | f : A \to B\}.$$

We therefore refer to $\mathcal{C}$(*A,B*) as a "hom-set", where distinct hom-sets are *disjoints* ([63], p. 9).

*2.3. The Notions of Functors as Morphisms between Categories and Natural Transformations as Morphisms between Functors*

Another fundamental notion of CT we have already used is the notion of *functor F*, that is, a «morphism between categories» ([63], p. 28):

**Definition 1.** *(Functors). A functor F: C → D is given by:*

- *An object-map assigning an object A of D to every object A of C.*
- *An arrow-map assigning an arrow Ff: FA → FB of D to every arrow f: A → B of C, in such a way that compositions and identities are preserved: $F(g \circ f) = Fg \circ Ff$; $Fid_A = id_{FA}$.*

In this way, a functor justifies a *homomorphism* or a «structure-preserving mapping» between the categories $\mathcal{C}$ and $\mathcal{D}$. Of course, for each category $\mathcal{C}$, there exists an *endofunctor* mapping a category onto itself: $\mathcal{C} \to \mathcal{C}$ and an *identity functor* Id$_C$ by which all identities (objects) in C are given, i.e., ([63], p. 31):

$$\mathrm{Id}_\mathcal{C} : \mathcal{C} \to \mathcal{C} := A \mapsto A, f \mapsto f$$

For our aims, other significant types of functors are:

- The *inclusion functor I: $\mathcal{C} \hookrightarrow \mathcal{D}$* between a category $\mathcal{C}$ and its sub-category $\mathcal{D}$. Of course, this is achieved by taking the identity map both for object-maps and arrow-maps.
- The *forgetful functor U:* **Mon** → **Set,** which sends monoids to their set of elements, "forgetting" the algebraic structure, and sends a homomorphism to the corresponding function between sets.

Moreover, the application of each functor can be covariant if it also preserves between the two categories, in addition to all the objects, the directions of morphisms and the orders of compositions. On the contrary, the application of a functor G is contravariant if it preserves all the objects but reversing all the directions of the morphisms (i.e., from $A \to B$, to $GA \leftarrow GB$), and the orders of their compositions (i.e., from *f* o *g* to *Gg* o *Gf*). In this case, the target category of the functor is the opposite of the source category. In a word:

**Definition 1.** *(Contravariance). Let C, D be two categories. A contravariant functor G from C to D is a functor G: $C^{\mathrm{op}} \to D$ (or equivalently $C \to D^{\mathrm{op}}$).*

Finally, another fundamental notion of CT is the notion of *natural transformation,* i.e., of morphisms between functors that are fundamental for a categorical *representation theory*.

**Definition 1.** *(Natural transformations). Let $F, G : \mathcal{C} \to \mathcal{D}$ be functors, either both covariant or both contravariant. A natural transformation $t : F \to G$ is a family of morphisms in $\mathcal{D}$ indexed by objects A of $\mathcal{C}$*

$$\{t_A : FA \to GA\}_{A \in Ob(\mathcal{C})}$$

*such that for all f: $A \to B$, the following diagram commutes:*

    *This condition is known as naturality. If each $t_A$ is invertible and then it is an isomorphism, t is a natural isomorphism:*

$$F \xrightarrow{\cong} G$$

*i.e., F and G are naturally isomorphic written $F \cong G$ ([63], p. 36).*

    Of course, as far as a natural isomorphism between functors *F* and *G* is given, the isomorphism between the relative categories, is given too, both in the covariant $\mathcal{C} \cong \mathcal{D}$ and the contravariant case $\mathcal{C} \cong \mathcal{D}^{op}$. From this, the definitions of equivalence and dual equivalence, respectively, are derived between categories ([63], p. 40):

**Definition 1.** *(Equivalence between categories). Two categories $\mathcal{C}$ and $\mathcal{D}$ are equivalent, $\mathcal{C} \simeq \mathcal{D}$ if there are functors F: $\mathcal{C} \to \mathcal{D}$ and G: $\mathcal{D} \to \mathcal{C}$, and natural isomorphisms with the identity functors of the two categories:*

$$G \circ F \cong \mathrm{Id}_{\mathcal{C}}, \;\; F \circ G \cong \mathrm{Id}_{\mathcal{D}}$$

    *If the two functors F, G are contravariant, we have the dual equivalence between the relative categories, i.e., $\mathcal{C} \simeq \mathcal{D}^{op}$.*

    The notion of opposite categories being functorially defined leads us to the categorical interpretation of the *principle of duality* that has a secular tradition in the history of logic (think only of the duality between $\wedge$ and $\vee$ in the De Morgan laws and then in a Boolean lattice), mathematics (think only of the duality between a function $f(x)$ and its inverse $f^{-1}(x)$), and physics (think only of the duality between a function $f$ and its Fourier transform $\hat{f}$. See also [70] for a survey about the notion of duality in mathematics and physics). Now, one of the more significant applications in quantum physics of a natural transformation between contravariant functors—emphasizing the radiographic power of CT in mathematics—concerns the categorical interpretation of the *GNS-construction* for a family of Hilbert spaces—effectively, for a sub-category of the category **Hilb** of the Hilbert spaces, those satisfying the Stone–Von Neumann theorem of the "finitely many unitarily equivalent representations of a quantum system" in QM—based on the *double contravariant application of the Gelfand functor* ([45], p. 807), as we discuss in Section 3.2.

    Finally, this hint to the GNS-construction in the mathematical formalism of QM, which historically inaugurated the operator algebra approach in quantum physics, introduces us to the application of the *functorial dual equivalence* between opposed categories to CT logic, as far as we were made to extend the operator algebra formalism from physics to logic because of the fundamental Stone's RTBA (see Section 5.1.2), which allowed Jónsson and Tarski to define the powerful construction of BAOs [4,5] (see Section 5.1.3). In fact, the dual equivalence between statements means that in CT logic, a statement $\alpha$ is *true* in/about a category $\mathcal{C}$ if its dual $\alpha^{op}$ (i.e., obtained from $\alpha$ by reversing all the arrows and their compositions) is *true* in/about the opposed category $\mathcal{C}^{op}$. This means that in CT logic, truth is *invariant* for the reversal of arrows and of the arrow composition orders ([63], p. 40).

*2.4. The Dual Equivalence between the Categories of Algebras and Coalgebras*

In this subsection, we briefly introduce the fundamental categorical duality between algebras and coalgebras because coalgebraic structures are becoming ever more significant in several fields of modern sciences, from mathematics—think only of the notion of *colimit* as a categorical counterpart of the notion of *direct limit* in mathematical analysis (see Appendix A, and especially [71,72] discussed in it)—to physics, logic, and computer science [6,73,74]. Indeed, in standard ST, the set-membership primitive is strictly related to (Cartesian) *products* among sets and then with algebraic structures $A \times A \to A$. The coalgebraic structures $A \to A \times A$ are, on the contrary, characterized by the dual operation of *coproducts*, effectively disjoint sums or set disjoint unions.

Indeed, to limit ourselves to the more interesting cases for us, direct products, categorically defined, correspond ([63] p. 21):

- In **Set**, to Cartesian products.
- In **Pos,** to Cartesian products with a pointwise order.
- In **Top,** to Cartesian products with a topological order.
- In **Vect**$_k$, products are direct sums.
- In a poset, seen as a category, products correspond to the *greatest lower bounds*.

Now, coproducts are the dual notion as the products in the sense that, formally, "coproducts in C are just products in C$^{op}$, interpreted back in C" ([63] p. 23). In fact, coproducts, categorically defined, correspond to direct sums, that is ([63] p. 24):

- In **Set,** to disjoint unions.
- In **Top,** to topological disjoint unions.
- In **Vect**$_k$, direct sums are coproducts
- In a poset, seen as a category, coproducts correspond to the *least upper bounds*.

The other starting point for illustrating the categorical duality algebras-coalgebras is the duality, with which we already met in illustrating the two past/future light-cones of the causality principle in fundamental physics, between *final* objects that are *initial* and *terminal* objects, respectively [12]:

**Definition 1.** *(Initial and terminal objects). An object I in a category $\mathcal{C}$ is "initial" if for every object A in $\mathcal{C}$, there exists a unique arrow from I to A, which we write as $\iota_A : I \to A$. An object T in a category $\mathcal{C}$ is "terminal" if for every object A in $\mathcal{C}$, there exists a unique arrow from A to T, which we write as $\tau_A : A \to T$ ([63], pp. 17–18).*

Of course, initial and terminal objects are *dual* in the sense that if $A$ is initial in $C$, e.g., in the category of algebras **Alg,** it is terminal in $C^{op}$, e.g., in the category of coalgebras **Coalg,** and vice versa. Indeed, algebras are characterized by products and initial objects, and coalgebras by coproducts and terminal objects.

Let us now illustrate arrow-theoretically the dual categorical characterization of algebras $A \times A \to A$ and coalgebras $A \to A \times A$ for the contravariant application of the same functor $\Omega$. Following Y. Venema ([6], pp. 394–395), we recall that an *algebra* $\mathbb{A}$: $A \times A \to A$ over a signature $\Omega$ is a set $A$ with an $\Omega$-indexed collection $\left\{ f^{\mathbb{A}} \middle| A^{ar(f)} \to A \right\}$ of operations, i.e., polynomial functions indexed by their ariety *ar*, that is, by the number of their arguments. These operations may be combined into a single map constituting the signature of a given algebra $\mathbb{A}$ i.e., $\alpha : \sum_{f \in \Omega} A^{ar(f)} \to A$, where $\sum_{f \in \Omega} A^{ar(f)}$ denotes the *coproduct* (or *sum* or *disjoint union*) of the sets $\left\{ A^{ar(f)} \middle| f \in \Omega \right\}$. It is easy to verify that a map $f : A \to A'$ is a homomorphism between the algebras $\mathbb{A} = \langle A, \alpha \rangle$ and $\mathbb{A}' = \langle A', \alpha' \rangle$ if the following diagram commutes:

$$A \xrightarrow{\;\; f \;\;} A'$$

$$\alpha \uparrow \qquad\quad \alpha' \uparrow$$

$$\Omega A \xrightarrow{\;\; \Omega f \;\;} \Omega A'$$

where it is obvious the signature $\Omega$ is interpreted as the *polynomial* set functor $\sum_{f \in \Omega} \mathcal{I}^{ar(f)}$, emphasizing that $\Omega$ operates on functions between sets. From this, it is possible to generalize to the notion of the algebra category for a given endofunctor $\Omega$, i.e., **Alg($\Omega$).**

**Definition 1.** *(Category of algebras for an endofunctor $\Omega$). Given an endofunctor $\Omega$ on a base-category C, an $\Omega$-algebra is a pair $\mathbb{A} = \langle A, \alpha \rangle$ where $\alpha : \Omega A \to A$ is an arrow in C. A homomorphism from the $\Omega$-algebra $\mathbb{A}$ to the $\Omega$-algebra $\mathbb{A}'$ is an arrow f: $A \to A'$, such that $f \circ \alpha = \alpha' \circ (\Omega f)$. We denote the induced category as* **Alg($\Omega$)**.

Similarly, but dually for coalgebras, $A \to A \times A$.

**Definition 1.** *(Category of coalgebras for an endofunctor $\Omega$).* Given an endofunctor $\Omega$ on a category C, an $\Omega$-coalgebra is a pair $\mathbb{A} = \langle A, \alpha \rangle$ where $\alpha : A \to \Omega A$ is an arrow in C. A homomorphism from the $\Omega$-coalgebra A to the $\Omega$-coalgebra A' is an arrow f: $A \to A'$, such that $f \circ \alpha = \alpha' \circ (\Omega f)$, and for which the following diagram commutes:

$$A \xrightarrow{\;\; f \;\;} A'$$

$$\alpha \downarrow \qquad\quad \alpha' \downarrow$$

$$\Omega A \xrightarrow{\;\; \Omega f \;\;} \Omega A'$$

It is evident that "the collection of coalgebra homomorphisms contains all identity arrows and it is closed under the arrow composition. Hence, the $\Omega$-coalgebras with their homomorphisms form a category: **Coalg($\Omega$)**" ( [6], p. 394), where the category C is called the base category of **Coalg($\Omega$)**.

In this way, the clear similarities between the structures of the algebra and the coalgebra categories can be made formally precise in CT. In fact, from observing the two diagrams above, the basic idea, which also explains the name "coalgebra", is that a coalgebra $\mathbb{C} = \langle C, \gamma : C \to \Omega C \rangle$ over a base category C might also be seen as an algebra in the opposite base category $C^{op}$, i.e., [6], p. 417:

$$\textbf{Coalg}(\Omega) = (\textbf{Alg}(\Omega^{op})^{op}$$

Specifically, the category of $\Omega$-coalgebras is *dually isomorphic* to the category of algebras over the functor $\Omega^{op}$ (i.e., acting like $\Omega$ but on the opposite category $C^{op}$). From this, the *dual equivalence* between the categories **Coalg($\Omega$)** and **Alg($\Omega^{op}$)**, for the contravariant application of the same functor $\Omega$, immediately derives, i.e.,

$$\textbf{Coalg}(\Omega) \simeq \textbf{Alg}(\Omega^{op})$$

To conclude, it is fundamental to recall with Venema himself ([6], p. 395) a *fundamental difference* between the categories of algebras and coalgebras functorially defined. Indeed, while in the category of coalgebras we are dealing with *arbitrary* set functors that can be whichever type of *homomorphic mapping*, in the category of algebras, we are constrained to dealing with functors that are *polynomials*. This difference is made clear when we reflect on the fundamental functorial coalgebraic construction of the *colimit operation* (see Appendix A), which plays a fundamental role in the categorical formalization of the notion of the "QV-foliation" in QFT and then for the formalization of the notions of *emergence* in the NR-formal ontology and the philosophy of nature, particularly in its application to a topology of NWF-sets (see Section 4.6 and [38]).

### 2.5. Non-Wellfounded Sets in the Category of Coalgebras as Causal Sets

All the *standard* axiomatic set theories, **ZF**, as far as sharing the *membership relation* $\in$ taken as a *primitive*, also share the *axiom of extensionality* and the *well-founded* character of set membership, granted by the *axiom of foundation* in its different versions. For instance, Zermelo's *axiom of regularity* grants the well-founded character of sets by not allowing a set to contain itself, so forbidding infinite chains of set inclusions. The axiom of regularity states that every non-empty set *A* contains an element that is disjoint from *A*. In its FOL formulation, it reads: $\forall x(x \neq \varnothing \rightarrow \exists y(y \in x \land y \cap x = \varnothing))$. In this way, by the prohibition of unbounded chains of set inclusions, set *total ordering*, and finally Zermelo's *well-ordering theorem* are granted too—even though definitively by adding the *axiom of choice* AC in **ZFC** (see [66], pp. 320–321 and pp. 360–372 for further explanations).

As Adam Rieger recalls in his monograph about NWF-set theories ([75], pp. 181–182), this means that, given well-ordering, the inductive constructive mechanism of new sets in all well-founded set theories is in terms of the construction of sets that at each stage *S* are formed as a collection consisting of sets formed at stages before *S* (see Figure 3 left). This is an inductive procedure that in **ZF** is extended to infinite sets by including in it Von Neumann's construction of the *cumulative hierarchy* of ordinal numbers as *ranks* of *well-founded* sets, i.e., as ranks or *stages* of a hierarchy of sets having a *minimal* element, to make axiomatically consistent Cantor's transfinite induction [76].

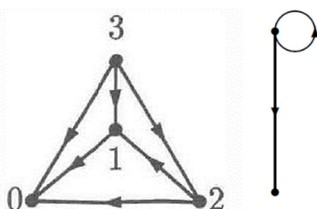

**Figure 3.** (**Left**) Set-tree representation of the number "3" as the root of a well-founded oriented graph of subsets satisfying von Neumann's number construction in ZF: $0 = \{\} = \varnothing$; $1 = \{\varnothing\}$; $2 = \{\varnothing, \{\varnothing\}\}$; $3 = \{\varnothing, \{\varnothing\}, \{\varnothing, \{\varnothing\}\}\}$. Note that in such a representation, the empty set is the node to which all the morphisms are pointing while the unary set is always the central node (from [11], p. 3). (**Right**) Representation of a non-well-founded oriented graph where the self-containing set $\{\{\cdot\}\}$ (reflexive graph) is allowed by the anti-foundation axiom, and where the symbol $(\cdot)$ emphasizes that the construction holds for sets of whichever cardinality (from [75], p. 9).

All this implies that in well-founded set theories, "when we are forming a set *z* by choosing its members, we do not yet have the object *z*, and hence we cannot use it as a member of *z*". Or, more synthetically, a set is a collection of previously given objects. In this sense, as Rieger says, referring this time to G. Boolos [77], it is evident that a set must *include* itself as a subset, like the same symbol of set inclusion $\subseteq$ signifies, but this is not the same as saying that a set *contains* itself as a *member*. In well-founded set theories, to satisfy the set-elementhood principle (see Note 1), each set can be a member/element only of another higher rank set. In a word, *self-inclusion is not self-membership*! In other terms, it is perfectly consistent in set theory writing: $\exists x(Sx \,\&\, x \in x)$, where $Sx$ stays for "*x* is a set", but if we take $\in$ as meaning strictly "'it is a member/element of' it is very, very peculiar to suppose it true". This peculiarity is precisely what characterizes Peter Aczel's NWF-set theory with its *anti-foundation axiom* [11]. In it, set self-membership, in the sense of a *self-containing set* $\{\{\cdot\}\}$ (see Figure 3, right), is allowed and then infinite chains of set inclusions are allowed too, so that no set total ordering but only set *partial orderings* are allowed in NWF set theory [11].

On the other hand, as again Rieger ([75], p. 178) but also as Aczel himself [11] recall, the Russian mathematician Dmitry Mirimanoff was the first in 1917 [78,79] to introduce the distinction between *ordinary* sets not admitting infinite descending membership chains (and then satisfying Zermelo's foundation axiom) and *extraordinary* sets admitting such infinite chains (not satisfying the foundation axiom) without *per se* being antinomic [13]. However, in

the light of the well-founded set theory and the role played in it by the transfinite induction in the construction of Von Neumann's *cumulative hierarchy* applied in **ZF** to the universe *V* (*proper class*) of sets—and in **NBG** also to *V* extensions with a cardinality higher than *V* because of Gödel "generalized CH"—it is hard not to agree with Von Neumann's statement of the "superfluous" character of non-well-founded sets [80].

Rieder, in his survey, emphasizes the recent revival of interest in NWF-set theories ignited by Peter Aczel's NWF-set theory based on the strong *anti-foundation axiom* [11], for its wider applications in TCS. It models, indeed, *parallel and concurrent computations* and *data streaming*, and it is applied to the categorical formalization of Kripke's *model theory of modal logic* (see [75], pp. 184–185, and overall [81] for a synthesis).

However, what completely escapes Rieger's (and Von Neumann's) treatment of NWF-set theories is that the proper formalization of Aczel's NWF-set theory requires the formal apparatus of the CT metalanguage to be fully expressed and justified. This dependence of NWF-set theory on CT formalization with the notion of set as hom-set (see Section 2.2) is, on the contrary, the starting point from which Aczel moves (see [11], 71–102). Before all, for justification of the powerful *final coalgebra theorem* for NWF-sets (see [11], 81–90 and [82]), this demonstrates that all the trees of NWF-sets share the same root as a common terminal object in the category of coalgebras (see Definition 7.).

This theorem, indeed, *mutatis mutandis*—where the main difference is that no *set total ordering* is here admitted but only an *infinite arbitrary branching* of trees of posets—plays the same role in the NWF-set theory based on the anti-foundation axiom that Zermelo's well-ordering theorem plays in well-founded set theories (see on this regard [83] and Section 5).

On this regard, the core difference in a categorical setting between (1) well-founded sets, admitting set total and well-ordering, and (2) NWF sets, where only set partial orderings are admitted, is synthesized in a very effective way by Aczel himself in the following way (see [11], Chapter 6, especially p. 77):

1.  In the recursive induction (the transfinite induction included) of well-founded set theories the *continuous set operators*, i.e., satisfying the CT primitive of the morphism composition, have only one *least* and one *greatest* fixed points, i.e., the *empty set* $\varnothing$ and the *universal collection V* (Von Neumann's *proper class*), respectively.

2.  In the recursive constructions of NWF-sets, where several arbitrary partial orderings are admitted, *the continuous set operators have many fixed points*—effectively, many possible lower and upper bounds of different recursive algebraic-inductive *upward directed* $\{\uparrow\}$, and coalgebraic-coinductive *downward directed* $\{\downarrow\}$ poset construction procedures (see below the application to *concurrent computations* in TCS for modal BAOs in Section 5.2.2 and Appendix A for applications to mathematical analysis in the CT framework).

More intuitively from a logical and epistemological standpoint, what is typical of standard ST based on set total ordering and well-ordering is the formalization of an inductive procedure, and then a *generalization procedure.* When we generalize, indeed, the recursive construction of ever more inclusive collections makes sense, i.e., sets of higher cardinalities that are typical of Boolean algebras, which have the property of recursively constructing the numerical sets on which their operations are defined.

On the contrary, the dual coalgebraic construction of *coinduction* (see [72] for a coalgebraic interpretation of the mathematical continuum and Section 5.2.2), based on *set-trees* "*unfolding*" from a common root according to *reciprocally irreducible unfolding paths* of the different posets, is aimed at epistemologically formalizing a *specification procedure.* In epistemology, this is typical of the logical/ontological theory of the *natural kinds* (genus/species) on a causal basis, as Kripke first emphasized [19].

Using a biological intuitive example of the natural kind logic, a "genus" (e.g., "the mammals") does not "include" $\subseteq$, in a proper set-theoretical sense, its different species (e.g., "elephants", "dolphins", "squirrels", "humans", ...) like a set its subsets but simply "admits" $\ni$ them. Indeed, the different species evolved as different, reciprocally irreducible

branches of the *ascendant-descendants' evolutionary trees* from their common "mammalian-root" (that is, from some (hypothetical?) common ancestor of all mammals). Effectively, $\ni$ is significantly also the symbol of the coalgebraic "co-membership relation" that is *dual* to the "membership relation" $\in$ in the CT relational semantics of the *modal* BAOs (see Sections 5.1 and 5.2). Indeed, as it is trivially evident from this biological example, no well-ordering relationship, and much less *no common metrics* justifying a common ordering relation $\leq$, is shared by the different species of mammals.

Now, as we see immediately, in QFT, this distinction genus-species also applies to physical objects such as the three different "generations" (not "sets"!) of fermions and gauge-bosons of the SM hat have in SSBs of the quantum fields at their ground state (="quantum vacuum condition") their common "branching mechanism" in an evolutionary cosmology (see Section 4). In other terms, this logic and mathematics is compliant with the evolutionary quantum-relativistic cosmology, based on the universal mechanism of the *symmetry breaking,* by which our universe progressively "populated itself" of ever more complex systems and structures (see [10] and Section 4).

Not casually, to formalize set-theoretically these strongly non-linear processes related to the causal light-cone in the universe evolution, some authors, e.g., R. D. Sorkin, proposed the so-called *causal set theory* as the proper set theory of quantum cosmology, with quantum gravitation included [84]. What characterizes Sorkin's trees of causal sets is indeed that they admit only *partial order relations* (i.e., reflexive, transitive, anti-symmetric, and locally finite order relations $\leq$) among sets, where the order relations are interpreted like the many *causal relations* in the Lorentzian manifold of the causal light-cone. The non-acceptable price to be paid for justifying the causal set theory in Sorkin's version is the supposition of the *discrete character* of the space-time manifold of the relativistic universe, which would mean renouncing the formal apparatus of GR in cosmology, and, finally, the same topological approach to the theoretical quantum and relativistic physics, "string theory" included (see [85] for a synthesis).

On the other hand—and this is the deep reason of Sorkin's theory, it would be non-sensical, if not contradictory at all, to suppose the set total or well-ordering in a causal set theory used for modeling the strongly non-linear, unpredictable character of the universe evolutionary branching processes, with each based on the universal mechanism of the *symmetry breaking* (phase transitions) with respect to the preceding universe states of the universe dynamics.

It is evident, however, that the same result of the limitation to posets when we speak of causal sets in physics can be obtained by modeling them in the framework of the NWF-set theory in a *categorical coalgebraic setting*, in a way that is perfectly compliant with the topological formalism of operator algebra and the string theory (continuous set operators) in relativistic and quantum physics and cosmology. Indeed, while in Sorkin's causal set theory for limiting the constructions to posets it is necessary to suppose the discrete character of the spatial-temporal manifold on which they are defined, in the NWF-set interpretation of causal set trees, this is neither necessary nor allowed. In it, indeed, we might limit the construction to causal posets simply because the NWF-sets naturally satisfy reflexivity and not totality in their ordering relations. Finally, their categorical coalgebraic setting is perfectly compliant with the interpretation of the causal event as *a terminal object* in the category of coalgebras for all the other events (effects) referring to it as to their common cause, and then belonging to the (subcategory of the) future-oriented light-cone with respect to their cause (see Section 2.4, Appendix A and overall [72]).

In fact, one of the more significant results of our categorical approach to QFT is the demonstration that the coalgebraic sub-category of the *doubled Hilbert spaces* **DHilb**—differently from the category of Hilbert spaces **Hilb** to which it belongs—satisfies the powerful *cocompleteness theorem* and then a *compactness condition* of the underlying topological space (see Section 4.8). In this modeling, indeed, each pair of doubled Hilbert spaces corresponds to a different dissipative quantum system, modeled as the result of SSB of QV

and then a phase transition among quantum fields at their ground state, with a necessary change in metrics that each phase transition implies in physics.

Historically, indeed, in the Aristotelian ontology of the natural kinds that applies to all physical entities and not only to the biological ones, each species (individual) with respect to the common genus (species) to which it belongs in the proper sense of "from which it *causally* derives" or "from which it is *generated*" adds a "specific *difference*". Thus, one of the greater and more influential Aristotelian and logically skilled philosophers and theologians of the Middle Age, Thomas Aquinas (1274–1323), in his *Commentary* to Aristotle's *Metaphysics* book, stated on this regard that:

> the predicate 'there exists' is said 'as many times as these differences are' (see [86], *Sententia libri Metaphysicae*, VIII, ii, 1694) [14].

In modern terms, this means that in the logic and ontology of the natural kinds, the existence of an object—either an individual, or a collection of individuals—is not related to the "set-elementhood principle" for which each object $x$ for existing must be a member (element) of another set (class) of a higher ordinal rank (or "higher type": see Note 1). In the limit, it must be an element of the universal class $V$, as the consistent usage of the "existential quantifier" exemplifies in ST predicate logic [15]. On the contrary, the modal existence predicate (not quantifier!) $E(x)$ in natural kind logic [60] is justified wherever a new identity relation $\mathrm{Id}_x$ and then a new unitary relation $1_x$ "*causally* emerges as a self-containing new whole" from within a collection of previously given objects. This logic and ontology, however – and in this I completely disagree with Cocchiarella –, can be formalized only in the CT metalanguage, where the assignment of a domain/codomain of objects to each morphism (predicate) depends on the primitive of the $\mathrm{dom}(\cdot)/\mathrm{cod}(\cdot)$ maps, and not on the membership relation taken as a primitive (Section 2.1). For this reason, "existence" can be a predicate $E(x)$ and not a simple quantifier ($\exists x$) like in ST logic.

In the natural kind logic based on coalgebras of NWF sets, in other terms, it is like what happens for the "non-normal classes" of the Russell paradox. Specifically, the *non-normal classes*, according to Russell's definition, "contain themselves as members", as it is exemplified in the well-known linguistic "Richard paradox". The class of all "polysyllables", indeed, *contains* the predicate "being polysyllable", i.e., the denotation of the identity shared by all the elements of the class, because "polysyllable *is polysyllable*", differently from the class of the monosyllables, given that "monosyllable *is not monosyllable*". The logical relevance of NWF-sets is that by the self-containing sets $\{\{\cdot\}\}$ of whichever cardinality, we can justify this notion of a "self-containing collection of elements" in a consistent axiomatic set-theory within CT based on Aczel's "anti-foundation axiom". On the other hand, it is trivial but perhaps significant to recall that NWF-sets are not classes but proper sets because if they do not satisfy set total and well-ordering, they satisfy *partial ordering* relations.

This means that NWF-sets make sense wherever we must model in a suitable ST the "emergence" of a *new property* $\mathrm{Id}_x$ shared by all the elements $x$ of a self-containing set and/or a *new object* $1_x$, which are irreducible to the simple combinatorics ("summation") of previously given elements because it is a new emerging self-containing "whole", as happens wherever we must deal in physics and more specifically in QFT with *phase transitions*: "more is different"! [8]. Think, for instance, of the well-known phase transition between the non-ferromagnetic and ferromagnetic phases of a metal (see Figure 4). In the QFT interpreted as the fundamental physics of the condensed matter systems, this process corresponds to the *symmetry breaking* of the quantum fields of the atoms of the metal in its non-ferromagnetic phase, characterized by the fact that the atom "magnetization vectors" (the momenta of the magnetization dipoles) are pointing in whichever direction (randomly aligned, so satisfying a spherical symmetry).

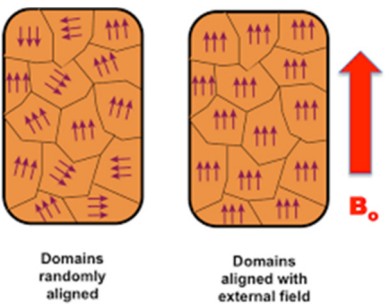

**Figure 4.** Intuitive representation of the phase transition between the non-ferromagnetic (**left**) and ferromagnetic phases (**right**) of a metal.

By the action of an external field, the phase transition occurs, by which the magnetization vectors are suddenly aligned along only one direction (symmetry breaking), from which the new collective property/predicate $Id_x$ emerges dynamically (causally) of "being magnet", effectively as a new *ordered phase-coherence domain* of the quantum fields. A collective property of the metal that can be lost dynamically by a phase decoherence of the quantum fields because of a change in the boundary conditions, that is, by raising the temperature beyond a given threshold.

Because of the universal character of the symmetry breaking mechanism in quantum physics and cosmology, we dedicate the next Sections 3 and 4 of this contribution to illustrate this on the physical-mathematical side of QFT. Then, we dedicate Section 5 to a discussion of the correspondent logic and ontology of this physics, the NR-formal ontology, all modeled in the unifying formal metalanguage of CT.

## 3. Some Elements of the Quantum Mechanics Formalism in a Categorical Setting

As a premise to this Section, a synthetic view of the so-called *standard model* (SM) of the quantum elementary particles (extended also to the *gravitons* of quantum gravity, now only hypothetical) is shown in Figure 5. SM (without gravitons) is actually one of the two sources, together with the *cosmological standard model* (CSM) of the general relativity theory (GR), of the evolutionary quantum-relativistic cosmology (see [10], for a synthetic overview). Effectively, SM is one of the more meaningful results achieved by QM and QFT during the XX cent., of which formalism I in this section provide some elements of a historical reconstruction to help the philosopher's understanding.

### 3.1. The Stone–Von Neumann Theorem in the Quantum Mechanics Formalism

As everybody knows, the main difference between the classical (Newtonian) mechanics and the quantum mechanics (QM) is that, because of the *uncertainty principle,* we cannot have "deterministic" but *irreducibly* only "statistical" representations (measurements) of the two *canonical variables*—position $x$ and momentum $p$—identifying univocally the *physical state* of a particle in the *state space* of a mechanical system. In other terms, in QM, the two canonical variables are no longer representable as independent of each other such as in classical mechanics. This means that in QM, when we represent the time evolution of a particle dynamics in its *state space,* we cannot have the two canonical variables as its orthogonal (independent) dimensions *commuting* with each other [16]. Indeed, in QM, because of the uncertainty principle:

$$\Delta p \Delta x \geq \hbar/2 \tag{1}$$

where $\hbar$ is the Planck constant, they are *conjugate variables,* i.e., their measurements depend on each other, so that they *do not commute* such as in the classical mechanics. In fact, the uncertainty principle means that a higher precision in determining the position $x$ means *necessarily* a lower precision in determining the momentum $p$, and vice versa.

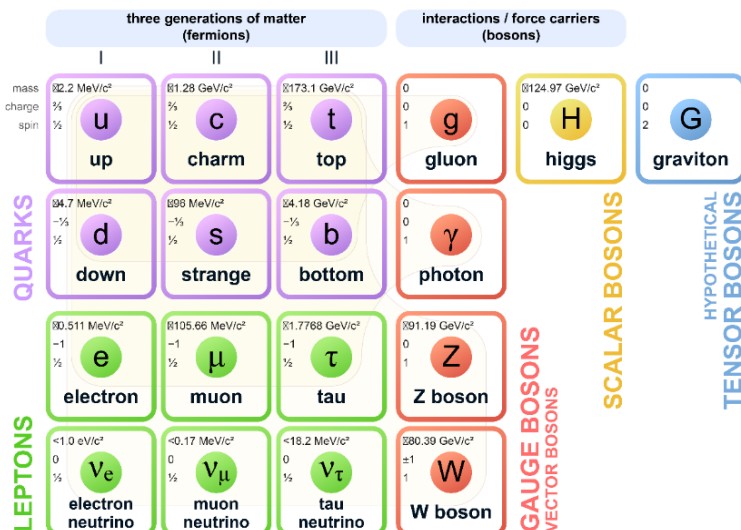

**Figure 5.** An updated scheme of SM extended to the hypothetical *gravitons.* The particles, interpreted in QFT as quanta of the relative (material and interaction) fields (see text), are subdivided into: (1) *Three generations (families) of "fermions"*, i.e., the quanta of the respective "matter fields", obeying the *Fermi-Dirac statistics*, i.e., because all have a fractional "spin", they occupy distinct levels of energy at their ground (minimum energy) state, such as electrons in the atom. Each generation is constituted by two massive "quarks" and two massive but lighter "leptons". The first generation of fermions, manifesting themselves at the universe lower energies (temperatures), are particles constituting the atoms of our ordinary experience (quarks "up" and "down" constituting protons and neutrons ("baryons") in the atom nucleus, plus the "electrons" and the "electronic neutrinos" of atoms). (2) *Three types of massive and massless "bosons",* i.e., all obeying the *Bose–Einstein statistics*. Specifically, because all having an integer spin, at the ground state, they all occupy the same energy lower level. They are (a) the "gauge (carrier) bosons" of the three quantum force fields: "strong" (massless *gluons*), "electromagnetic" (massless *photons*), "weak" (massive *Z, W bosons*); (b) the "scalar bosons" of the *Higgs fields* (massive *Higgs bosons*); and (c) the "tensor bosons" of the *gravitation field* (massless *gravitons*) of the quantum gravity theory, now only hypothetical.

The *non-commutative* character of the quantum canonical variable measurements seemed, therefore, at the beginning of XX sec., to preclude the usage of a geometric representation of the dynamics of a quantum system. Effectively, in his famous work on the foundations of geometry *Grundlagen der Geometrie* (1899) [87], David Hilbert, for the first time in the history of thought, gave a complete axiomatic version of the Euclidean geometry, demonstrating that geometrical spaces are *commutative varieties.* However, Hilbert himself suggested the solution of the conundrum for the nascent quantum physics formalism by his following works on the *functional analysis.* In it he introduced the notion of the *infinite dimensional* Euclidean spaces endowed with an *inner structure* ("inner product": see Note 19), afterward developed and denoted by J. Von Neumann as *Hilbert spaces.* Indeed, the spatial-temporal position of particles in QM is no-longer "pointwise", but it corresponds to a "field", i.e., to a "volume" of the possible spatial position distribution for each "instant" of time because of the uncertainty principle. This is a notion that Hilbert applied to his further discovery of the *spectral theory* in the algebra of matrices [88].

Because Hilbert's abstract spectra also apply unexpectedly well to the study of the observed atomic spectra of the hydrogen atom electromagnetic emissions, and then to Werner Heisenberg's *matrix mechanic* formulation of QM, all this led J. Von Neumann—at the time, the assistant of Hilbert at Göttingen—to write in 1932 his famous book on the foundations of the mathematical formalism of QM [89]. Indeed, at the beginning of the 1930s, as any historical reconstructions of the birth of QM recall, there existed two different

theoretical interpretations (models) of the nascent QM, both having at that time its essential observational basis in the spectra of the light emission of the "excited" hydrogen atom. The two models recalled are: (1) *Werner Heisenberg's* matrix *mechanics* formulation of QM [90], completed in 1925 with the contributions of Max Born and Pascual Jordan [91,92]; and (2) *Erwin Schrödinger's statistical wave mechanics* formulation of QM, published in its complete form in 1926 [93]. Now, in his momentous book, Von Neumann demonstrated the *algebraic isomorphism* and then the *mathematical equivalence* of these two early formulations of QM. To achieve this result, the essential step was the demonstration that it is sufficient to interpret quantum states as *vectors of a Hilbert space* $\mathcal{H}$, and the *quantum observables x* (position) and *p* (momentum) as *operators* acting on this vector space and constituting in turn another Hilbert space $\mathcal{H}^*$, *isomorphic and dual* as to $\mathcal{H}$.

Effectively, the possibility of using the Hilbert vector spaces satisfying the commutativity condition in QM depends on the fact that the statistical wave function *f* of each canonical variable, *x* or *p*, in QM *commute* with the *Fourier transform* $\hat{f}$ of the statistical wave function of the other canonical variable, i.e., $\hat{p}_x$ or $\hat{x}_p$, respectively. From this, the fundamental notion in quantum physics of the *canonical commutation relation,* CCR, derives, each identifying univocally a quantum state in QM, i.e.,

$$[\hat{x}, \hat{p}_x] = i\hbar \qquad (2)$$

where *i* is the imaginary unit, given that the Fourier transform and then the Hilbert vector spaces are defined on the complex numerical field $\mathbb{C}$. In this way, it is easier to understand, at least intuitively, why a Hilbert space is connoted as a *functional space.* In it, indeed, the orthogonal dimensions of the space, on which the "coordinates" of the quantum states are defined, are the outputs of a particular class of statistical distribution functions, those obeying the CCR principle. What is fundamental to recall for our theoretical aims is that the core of Von Neumann mathematical formalism is related to the momentous theorem, formulated simultaneously by Von Neumann himself and by the American mathematician Marshall Stone between 1930 and 1932 [94–97]. Specifically, the so-called *Stone–Von Neumann theorem.* Stone is the same Marshall Stone of the RTBA in *logic*, published in 1936, which we already recalled in this contribution, and we discuss in Section 5.1.2. The Stone–Von Neumann theorem indeed demonstrated the *uniqueness* of a *finite* number of *unitarily equivalent CCRs* and then of a family of Hilbert spaces to faithfully represent a quantum system in QM.

Finally, because of the *superposition principle* in QM, while a *pure state* regarding the statistical wave function of a single particle corresponds to a *ray* in the Hilbert space, a *vector* in the Hilbert space corresponds to the *quantum state* as superposition of several wave functions [17], i.e., it corresponds to a *mixed state,* relative to many particles "occupying" the same quantum state that are anyway indistinguishable because of the uncertainty principle to constitute an irreducible "whole".

### 3.2. The Completion of Von Neumann's Formalism by the Gerlfand–Neimark–Segal Construction and Its Categorical Interpretation

As Klaas Landsman emphasizes very well in his recent textbook [45] on classical and quantum mechanics re-interpreted within the unified formalism of the operator algebra, Von Neumann's early formalization of QM did not satisfy the ideal of *absoluteness* pursued by Von Neumann himself in his construction. Effectively, indeed, his fundamental demonstration of the *reversed isomorphism* $V \rightarrow V^* \rightarrow V$ between a finite Hilbert (vector) space $\mathcal{H}$ and the dual Hilbert space $\mathcal{H}^*$ of its observables (operators) [18] holds only "locally" for a given Hilbert space, and not "generally" for a *whole family* of Hilbert spaces, so to make Von Neumann's early formalism not per se applicable to the "relativistic QM", or QFT, that is, a quantum theory, in which the special relativity principles hold.

Therefore, to complete our survey, it is necessary to recall the momentous *Gelfand–Naimark–Segal (GNS) construction* that systematically extended the Fourier *duality* between a function *f* and its Fourier transform $\hat{f}$, from the infinite *function spaces* of functional

analysis to the infinite *vector spaces* such as per se the Hilbert spaces. Effectively, indeed, the relevance of the GNS-construction for the completion of the QM formalism is strictly related to the attempt started by Von Neumann himself to model the algebras of the physical observables, i.e., the operators over Hilbert spaces. Historically, this is related to the abstract characterization given in 1943 by Israel Gelfand and Mark Naimark of the so-called *C\*-algebras* [19] in a momentous paper [98], in which they refer to Hilbert spaces but without any reference to *operators* on a Hilbert space. This extension, in addition to the term and the notion of "C\*-algebra", was introduced in 1947 by Erwin Segal [99] to signify the norm-closed subalgebras of the Banach algebra of a Hilbert space *B(H)*, namely the space of bounded operators on some Hilbert space $\mathcal{H}$, where *C* of C\* stays for "closed". Therefore, Segal defined a *commutative* C\*-algebra of a Hilbert space as a "uniformly closed, *self-adjoint* algebra (the symbol \*, in C\*) of bounded operators on a Hilbert space", where self-adjoint elements are those satisfying the condition $x = x^*$ (see Note 19).

Theoretically, this had a fundamental double consequence on the QM formalism. On the one hand, it grants that an *algebra of observables*, i.e., a finite number of operators on the Hilbert space, as far as *irreducible,* is sufficient for faithfully representing a quantum system. On the other hand, it mathematically grants the existence of *a finite orthonormal basis* of the Hilbert space for each quantum system. Thus, it generalizes the Von Neumann demonstration of the "reversed" *isomorphism* (= dual equivalence) between a Hilbert space $\mathcal{H}$ and its dual operator space $\mathcal{H}^*$ to a whole *family* of Hilbert spaces. Following Landsman's reconstruction in the framework of CT formalism (see [45], pp. 807–808), in the case of vector spaces such as the Hilbert spaces, the GNS-construction generalizes the "local" or, using the CT jargon, the "unnatural" isomorphism $V \rightarrow V^* \rightarrow V$ discovered by Von Neumann, to a whole subcategory of the category of Hilbert spaces **Hilb.** Namely, these satisfy the Stone–Von Neumann theorem in the representation theory of a quantum system in QM.

Indeed, by the *double contravariant application* of the *Gelfand transform,* so constituting the *functor* of this subcategory of Hilbert spaces, we obtain, by the consequent *natural transformation* (see Definition 5.), the *natural* isomorphism: $V \rightarrow V^* \rightarrow (V^*)^* \rightarrow V$. Therefore, the algebraic GNS-construction determined, from the second half of the last century on, the development of the *operator algebra* formalism as the proper formalism of quantum physics in the framework of Hilbert space modeling, which we can formalize in CT in a particularly effective way.

### 3.3. The Interpretation of QFT as "Second Quantization" with Respect to QM

As we know, *quantum superposition* is a fundamental principle of QM that has no correspondence in classical mechanics. It states that any two or more quantum states, as represented by statistical wave functions $\psi$, can be "added together" or *superposed* to obtain another valid quantum state. The physical analogue of this property is the "constructive" or "destructive" interferences among two or more wave forms, which in the classical "double-slit experiment" of QM reveals the *wave-like* behavior of all quantum objects. Mathematically, the superposition principle is a consequence of the *linearity* of the Schrödinger equation $\Psi$, for which any linear combination of solutions of the system will also be a solution.

The superposition principle was originally suggested by Paul Dirac to extend the QM formalism to model the quantum behavior not only of single particles but also of many particles "occupying" the same quantum state to determine a sort of "ontological shift" in quantum physics. Namely, this is from considering particles as the fundamental objects to considering *oscillating fields* $f(x_N, t)$, that is, different spatial distributions varying in time of $N$ particles, as the fundamental object of inquiry in QFT.

In this framework, we must consider "particles" as *quanta* of the respective oscillating *material* fields, just like photons are quanta of the *interaction* electromagnetic field. The same Dirac suggested the name of "Second Quantization" (SQ) to this approach to QFT [100] because, in the case of non-relativistic systems where the number of particles is *fixed* and

*finite* but too large to use the Schrödinger wave function, it gives a formal alternative for the computations in QM. Effectively, SQ taken in this sense is an "algorithm", as C. M. Becchi states in his useful survey about the SQ formalism, to which we mainly refer in this subsection [101]. In the QM case, indeed, the approach is immediately compatible with the finite number of the degrees of freedom depending on the Stone–Von Neumann theorem just recalled.

However, when we pass into the *relativistic* realm of the quantum field theory (QFT), there exists the problem that the *number of the degrees of freedom* of the system cannot be considered finite any longer but *in(de)finite.* Effectively, this is the core of the famous *Haag's theorem* in QFT [102], which admits different solutions for this problem. We briefly illustrate two of them. Namely, (1) the SQ solution for QFT systems staying mainly *in only one phase* ("closed" QFT systems), which is the object of this subsection, and (2) the solution for QFT systems *passing through different phases* ("open" or "dissipative" QFT systems) in Section 4.

For our aims, following the reconstruction of the SQ method applied both to QM and QFT given in [101], it is important to consider *two new concepts* consequently introduced by Dirac in the general formalism of QM and QFT [100] and that, as such, have a general significance in quantum physics. Indeed, Dirac developed the construction of SQ originally for the treatment of the superposition among finitely many identical *bosons* since his early work in 1927. It concerned the emission and absorption of the electromagnetic radiation, and it was afterward extended to *fermions* by a successive work of Pascual Jordan and Eugene Wigner in 1928 [103] (see Figure 5).

These two notions, using the further contribution of the Russian mathematician Vladimir Fock [104], are: (1) the *Fock space,* as a particular type of Hilbert space generated by the *tensor product* of several one-particle Hilbert spaces to satisfy the superposition principle; and (2) the dual *creation-annihilation operators* acting on the Fock space. Because these operators are "Hermitian conjugates" to each other, they are denoted as $A$, $A^\dagger$, respectively. Effectively, these operators do not act directly on a state of a $N$-bosons (or fermions) Hilbert space, but to use them, we need to extend the (Hilbert) state space $\mathcal{H}_S^{(N)}$—where the subscript $S$ stays for the type of symmetry (bosonic or fermionic) the algebra satisfies (see below)—to the *direct sum* of $N$ state-spaces. Each of them, in turn, is the *tensor product* of $N$ single-particle Hilbert spaces to constitute a "Fock space" $\mathcal{F}(\mathcal{H})$. Intuitively, we can see at the "Fock space", like a "book (summation of pages)", whose "pages" are "Fock states", namely, Hilbert spaces, each of which is a tensor product of $N$ one-particle Hilbert spaces [20].

Effectively, the original intuition behind the Dirac treatment of the SQ algorithm in QM [100] is the strict relationship existing between a composite system and the *tensor product* of the related Hilbert spaces. In other words, "the state space of an assembly of systems is identified with the tensor product of the state spaces of each system" [101].

At this point, always following [101], we can informally define the "Fock space" notion as the "sum" of a set of Hilbert spaces representing zero-particle states, one-particle states, two-particle states, and so on. Therefore, using Dirac *matrix bra-ket symbolism* to denote the wave-function of a quantum state $|\psi\rangle$, the SQ algorithm for $N$-identical superposed particles works in the following way. We start from the 1-particle state $|1\rangle$. We apply the "annihilation operator" and obtain the 0-particle state or the *quantum vacuum* (QV) state $|0\rangle$. Then, we apply one-time, two-times, three times . . . $N$-times the "creation operator", and we obtain the 1-particle state again $|1\rangle$, and then the 2-particles $|2\rangle$, 3-particles $|3\rangle$, . . . $N$-particles $|N\rangle$ quantum states, where $N$ therefore denotes the particle *occupation number* of each state of the resulting Fock space. On this basis, given the definitions of the "tensor product" $\otimes$ and "direct sum" $\oplus$ between vector spaces (see Note 20), we can give a more formal characterization of the Fock space $\mathcal{F}(\mathcal{H})$ as the (Hilbert) direct sum of tensor products of $N$ copies of single-particle Hilbert spaces $\mathcal{H}$, i.e.,

$$\mathcal{F}_\nu(\mathcal{H}) = \mathcal{H}^{(0)} \oplus \mathcal{H}^{(1)} \bigoplus_{N=2}^{\infty} \mathcal{H}_{S_\nu}^{(N)}, \text{ where } \bigoplus_{N=2}^{\infty} \mathcal{H}_{S_\nu}^{(N)} = (S_\nu(\mathcal{H} \otimes \mathcal{H})) \oplus (S_\nu(\mathcal{H} \otimes \mathcal{H} \otimes \mathcal{H})) \oplus \ldots \quad (3)$$

As we see in the formula, the number of the composing Hilbert spaces—and then of the system *degrees of freedom*—can go, in principle, to infinity, so subtracting any algorithmic (i.e., finitary) value to Dirac's SQ construction. Moreover, in Equation (3) defining the notion of Fock space, the further operator $S_\nu$ appears. It makes the tensor *symmetric* or *anti-symmetric*, depending on whether the Hilbert space representing $N$ particles is obeying *bosonic* ($\nu = +$) or *fermionic* ($\nu = -$) statistics. Indeed, the *Pauli exclusion principle* in QM states that two or more *identical fermions* cannot occupy the same quantum state *simultaneously*. For instance, in the case of the electrons in atoms, for which Wolfgang Pauli originally formulated his principle, this means it is impossible that in an atom with many electrons, they have the same values for all *four quantum numbers* identifying their quantum states.

Specifically: (1) $n$ (roughly, the quantized electron energy); (2) the "azimuthal quantum number"$\updownarrow$; (3) the "magnetic quantum number" $m_\updownarrow$; and (4) the "spin quantum number" $m_s$. Therefore, if two electrons stay in the same "orbital" or "energy level" of an atom at its ground state, the first three quantum numbers $n$, $\updownarrow$, $m_\uparrow$ must be the same, and then their $m_s$ must be different to satisfy Pauli's *exclusion principle*. Specifically, the electrons must have opposite half-integer spin projections of $\frac{1}{2}$ and $-\frac{1}{2}$, respectively. On the contrary, particles with an integer spin, that is, *bosons,* are not subject to the Pauli exclusion principle. In this way, any number of bosons can simultaneously occupy the same quantum state.

A more rigorous statement of the same principle that applies directly to the explanation of the $S_\nu$ symbol in Equation (3) concerns the permutations among identical particles, such as those characterizing the tensor product of a finite number of single-particle Hilbert space defining a Fock state in the Fock space (see Equation (3) and Note 20). The total wave function is *anti-symmetric* (i.e., it changes its sign) if the particles are fermions while the sign remains unchanged and then the total wave function is *symmetric* if they are bosons. For this reason, we say that fermions satisfy sets of *canonical anti-commutation relations* (CARs) while bosons satisfy sets of *canonical commutation relations* (CCRs) (see Section 3.1).

When we pass to the relativistic QFT, the situation changes. On the one hand, the Fock space with its decomposition (a sort of factorization, indeed) of the composite Hilbert space into a set of component Hilbert spaces—and particularly, the 0-occupation state of the QV-state $|0\rangle$—acquire a precise physical meaning, losing their purely algorithmic flavor. Indeed, to use Becchi's words in his review of the SQ approach [101], in the relativistic QFT, "particles are produced and absorbed", i.e., continuously "created" and "annihilated", from/to the QV.

On the other hand, all this means that we cannot suppose that the Stone–Von Neumann theorem of QM holds *generally* also in QFT, as the Haag theorem demonstrated [102]. Indeed, per se, in QFT, we are faced with an indefinite number of degrees of freedom—and/or with *an indefinite number of unitary inequivalent representations of the CCRs (CARs)*—and not with *a finite number of unitary equivalent representations of CCRs (CARs),* as the Stone–Von Neumann theorem states for a single quantum system in QM (see Section 3.1).

The solution proposed firstly by Dirac himself [102], developed systematically during the further 50 years into an SQ approach to QFT [105], and discussed at large in [101] is to model QFT fields as *freely oscillating in the vacuum* to model them as *non-interacting* systems (i.e., energetically closed systems). This makes it possible to also model QFT systems in the standard framework of the *statistical mechanics* of systems that are at *equilibrium* in the *asymptotic (infinite limit) condition*. This means, overall, the possibility of also applying the *perturbative methods* in QFT (like in QM) to model field interactions, having "cut away" all the undesired interactions, so that we can always use the canonical *Hamiltonian representation*, which holds only for *closed* dynamic systems, also for QFT system *dynamics* to define its *total energy*.

The two steps followed by Dirac and leading him to the construction of the *quantum electro-dynamics* (QED)—that is, the quantum theory of the electromagnetic interactions–consists in firstly modeling bosonic fields (photon fields in Dirac QED), and secondly fermionic (electron) fields as *freely oscillating in the vacuum*.

Starting from the *bosonic fields*, both problems emerging in such a QFT modeling (the so-called "zero-energy point" problem and the necessity of using an indefinite number of quantum harmonic oscillators in QFT because of the Haag theorem (see Section 4.1)) can be systematically solved by supposing the reference to (the knowledge of) the only Hamiltonian of the system [101]. Indeed, the supposition of quantum fields freely oscillating in the vacuum means that we are modeling a QFT system in a canonical Hamiltonian way, that is, as a "closed" system.

Similar considerations can be used to define the Fock space for *fermionic fields*, so that "in the general situation we have a mixed bosonic and fermionic Fock space with a unique vacuum state" [101]. Moreover, in the case of QFT for fermionic fields, we have the further advantage, as Dirac first noticed for the electron theory [100], that their normal modes are those whose energy is not bounded from below. Indeed, in the ground state of such models, to satisfy Pauli's exclusion principle, their normal modes "under a certain level (the zero-energy Fermi level) are occupied, while those above this level are empty. This corresponds to the picture of the *Fermi Sea*" [101].

Surely, the main successes of QFT based on the SQ approach and finally on the systematic usage of the perturbative methods of statistical mechanics in fundamental physics are innumerable. The same SM in its actual form is surely the most significant of them. On the other hand, the successes of SM are strictly related to the powerful calculus tool of the so-called *Feynman diagrams*.

The intuitive mechanistic model in the background of the ordinary divulgation of the Feynman diagrams is the following. The particles (fermions) interact by exchanging reciprocally force quanta (bosons, e.g., photons) isolated in the vacuum, such as when two ice-skaters move up/away from each other, by exchanging a basketball, being completely isolated from, i.e., having no interaction with, the crowd of the other skaters populating the icy lake. This is an ideal but *nonrealistic situation* indeed. In fact, Feynman diagrams are rightly considered as the more effective application of the perturbative methods of statistical mechanics to QFT because in these diagrams, all the undesired interactions (undesired interaction branches) are systematically cut away. So, *only* the significant interactions (those satisfying the Hamiltonian in the framework of Feynman's "path integral" formalism) are considered (see [106] for a popular exposition of the theory by Feynman himself).

## 4. The Extension of QFT to Modeling Dissipative Systems in a Categorical Setting

### 4.1. The Theoretical Problem at Issue with the Haag Theorem

At the end of his survey on the SQ formalism in QFT to which we referred to in the previous section [101], Becchi quotes the "alternative statistics" related to the *Bogoliubov transform* that also applies to "open" quantum systems and that we illustrate systematically in this section using the CT formalization.

As we know, the fundamental component of Von Neumann's standard formalism of QM [89] is the so-called "Stone–Von Neumann theorem" recalled in Section 3.1, for which a finite number of *unitarily equivalent CCRs (CARs) and then Hilbert spaces* is sufficient for representing a system in QM. In the SQ approach to QFT, this holds as far as we are representing one or more superposed quantum particle fields but staying in *only one phase*, i.e., as freely oscillating in the vacuum, and then satisfying the Hamiltonian of energy for closed systems.

Now, the Haag theorem demonstrated that this representation—the so-called "Dirac's picture"—does not hold *generally* in QFT because, as far as we interpret a QFT system as an *interacting (not closed) "many-body system"*, i.e., we consider the crowd of the other skaters on the icy lake surface following the just quoted metaphor, it can pass through *different phases*. Indeed, the Haag theorem demonstrated formally that in QFT—in the infinite volume limit of the functional analysis—there exists an infinite number of *unitarily inequivalent representations of the commutation (bosons) and anti-commutation (fermions) relations*, all compatible with the QV ground state $|0\rangle$. More precisely, in QFT, there exists a mismatch

between the *field dynamics* (the Heisenberg matrix equations) and its representation (the Hilbert space of physical states).

In fact:

> the same dynamics (i.e., the same set of Heisenberg equations) may lead to different solutions when unitarily *inequivalent* representations (Hilbert spaces of physical states) are used in computing the matrix elements. *The choice of the representation to describe our system is thus of crucial importance in solving the dynamics:* the same dynamics may be realized in different ways (i.e., in different unitarily inequivalent representations). The choice of the representation may be considered as *a boundary condition* under which the Heisenberg equations have to be solved (see [9], p. 55).

This "boundary condition" could *epistemologically* be the supposition that we know the Hamiltonian of the system, as Becchi taught us in his reconstruction of the SQ approach in QFT (see Section 3.3), but it could also *physically* be the "thermal bath" with which a quantum system is continuously exchanging energy in a *balanced* way, in a dissipative interpretation of QFT as "many-body physics". In such a case, the Hamiltonian is not supposed but *dynamically generated* by the system itself!

*4.2. The Thermal Interpretation of QV*

Generally, in QM, we could imagine the QV ground state as the physical state in which all the quantum fields at their ground state "are zero". Effectively, following, for instance, J. Maldacena's divulgation, the fields take random values around zero because of the uncertainty principle. However, these field fluctuations "happen at short distances" while "in the vacuum, at long distance they average out to zero, so that we recover the classical result where the fields are all zero" [107]. In other terms, by such a mathematical construction, there would be no ultimate distinction between QV and the *mechanical vacuum* of the classical and statistical mechanics. In it, no energy contribution is from the vacuum and then the system can be considered as isolated, i.e., the quantum fields are freely oscillating in the vacuum.

To oversimplify, in this approach, we are considering the unavoidable QV fluctuations in the light of the overall averages of Boltzmann's "molecular chaos" of statistical mechanics, and then in the light of the *second principle of thermodynamics* for "closed" or "isolated" systems. However, it is possible to move a step forward, and consider, more realistically, the unavoidable fluctuations of the QV at the ground state in the light of the *third principle of thermodynamics*. This step forward was made by the Japanese physicist Hiroomi Umezawa during the 1990s of the XX cent., leading him to interpret QFT as a "thermo-field dynamics" (TFD) [108,109]. This interpretation was originally developed by Umezawa for modeling condensed matter physics systems, and specifically the neurodynamics of the brain interpreted as a "dissipative system" or "open" system.

Indeed, the *third principle of thermodynamics* states: "The entropy of a system approaches a constant value as the temperature approaches the absolute 0 °K ($-273$ °C)". It was the Nobel Laureate (1921) Walter Nernst who first discovered that for a given mole of matter (namely, an ensemble of an Avogadro number of atoms or molecules), for temperatures close 0 °K, $T_0$, the variation of entropy $\Delta S$ would become infinite (by dividing by 0). Nernst therefore demonstrated that to avoid this catastrophe, we must suppose that the molar heat capacity $C$ is *not constant* at all but vanishes in the limit $T \to 0$ to make $\Delta S$ finite, as it must be. This means, however, that near the absolute 0 °K, there is a mismatch between the variation of the system inner content of energy and the supply of energy from the outside. We can avoid such a paradox, which would violate the *first principle* of the energy balance in any physical system, by supposing that this mysterious *inner supplier of energy is the vacuum* for whichever physical system. Conversely, this implies that the absolute 0°K is unreachable for whichever physical system. In other terms, *there exists an unavoidable fluctuation of the elementary constituents of matter*, at any level of matter organization.

Therefore, an immediate consequence of the *third principle of thermodynamics* is the association of whichever mole of matter with an oscillating *matter field*, and then in QFT, the consequence of a *thermal interpretation* of the QV as *the universal energy reservoir*, with a *temperature T > 0 °K*. This means that no physical—classical or quantum—system is conceivable as "isolated" or "energetically closed", since it is necessarily "open" to the unavoidable vacuum fluctuations in its background. At the same time, this means that, in QFT interpreted as the fundamental physics, at any level of matter field organization (see the notion of "QV foliation"), the QV at its ground state $|0\rangle$ corresponds to the fluctuating ground state, with temperature >0 °K characterizing a given "heap" of matter fields. The ontological conclusion for the fundamental physics is that we cannot any longer conceive the physical systems, either at the microscopic, mesoscopic, or macroscopic levels, as isolated in the mechanical vacuum, such as in the classical representation:

> The vacuum becomes a bridge that connects all objects among them. No isolated body can exist, and the fundamental physical actor is no longer the atom, but the field, namely the atom space distributions variable with time. Atoms become the "quanta" of this matter field, in the same way as the photons are the quanta of the electromagnetic field ([110], p. 1876).

For this discovery, eliminating the notion of the "inert isolated bodies" in the mechanical vacuum of the Newtonian mechanics at the fundamental level, Walter Nernst is a chemist who is one of the founders of the modern quantum physics.

To complete this historical reconstruction, we must recall that Umezawa's "thermal" interpretation of QFT as fundamental physics of biological and neural systems cannot give a suitable memory mechanism to brains as dissipative systems because in this approach, each new memory "script" overwrites the precedent one. This limit is overcome by Giuseppe Vitiello's further formalization of the principle of *QV-foliation* based on the theoretical—and experimental, e.g., in neuroscience—evidence that *different, stable phase coherences of the quantum fields* at the same "balanced" (0-energy summation) ground state with *T > 0 can coexist* in dissipative quantum systems, without interfering with each other.

Moreover, they can be *locally* ordered over each other, being univocally *addressable* ("labeled") because they are univocally and "dynamically" *indexed* to become an effective tool used by nature for "constructing" *complex systems* and effective *memory sub-systems* in biological and neural dissipative systems (see Sections 4.6 and 4.7). The illustration of Vitiello's approach and his group to QFT is covered in the rest of this section (see the more comprehensive synthesis of this approach in [9]). Afterward, we offer a categorical formalization of the QV-foliation using the powerful colimit operators in Section 4.8.

### 4.3. The Bogoliubov Transform

The Haag theorem showed us that QV at its *ground state* is compatible with infinitely many unitarily inequivalent representations of the canonical commutation relations (CCRs), for bosons, and the canonical anti-commutation relations (CARs) for fermions and then of the relative Hilbert spaces.

The fundamental mathematical modeling in QFT of the process of creation/annihilation from QV of bosons and fermions as quanta of their relative fields was offered by the Russian physicist and mathematician Nikolay Bogoliubov in 1958 [111]. He demonstrated that, given a pair of CCRs (mathematically defined on a hyperbolic function basis) for a pair of creation/annihilation operators for a boson on the Hilbert space, i.e., $\left[\hat{a},\hat{a}^{\dagger}\right] = 1$, and another pair of operators for another boson, i.e., $\left[\hat{b},\hat{b}^{\dagger}\right]$, there exists a transformation (the Bogoliubov transformation) mapping the former pair into the latter.

The same holds for the CARs (i.e., defined on a circular function basis) between pairs of fermionic creation/annihilation operators, i.e., $\left\{\hat{a},\hat{a}^{\dagger}\right\} = 0$, $\left\{\hat{a},\hat{a}^{\dagger}\right\} = 1$. In other terms, Bogoliubov demonstrated that there exists an *isomorphism*, either of the *CCR algebras* for bosons, or of the *CAR algebras* for fermions, by which different "degenerate states" of the QV

at the ground state $|0\rangle$ are modeled, as previewed by the Haag theorem and corresponding to a process of creation/annihilation of bosons or fermions, respectively.

What is fundamental for our aims is that the application of the Bogoliubov transform implies a *phase shift* of the respective fields. In other terms, differently from QM (and QFT in its SQ formalization), the Bogoliubov transform allows QFT to calculate over *phase transitions* that per se imply he consideration of QFT systems as *open systems* because of the passing through of different phases. In this sense, Becchi spoke about "an alternative statistics" as to the SQ one that, as we know, can model per se QFT systems in only one phase.

### 4.4. The Goldstone Theorem

The "dynamic richness" of QV and its "degenerate states" is the result, in fundamental physics, of another momentous theorem that historically played a fundamental role in the construction of SM: *the Goldstone theorem.* Indeed, this theorem, together with the original idea of P. Higgs about the existence of the so-called *Higgs field* and its quantum, the *Higgs' boson*, led to the explanation of the *electro-weak symmetry breaking* that is a fundamental ingredient of SM [112–114]. Effectively, the Goldstone theorem demonstrates the existence of infinitely many *spontaneous symmetry breakings* (SSBs) of the QV ground state $|0\rangle$, corresponding to the spontaneous instaurations (i.e., without any energy waste) of *long-range correlations* among quantum fields at their ground state, and then corresponding to as many *phase coherence domains* among them (see Figure 6).

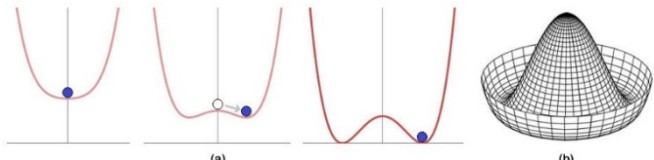

**Figure 6.** Intuitive representation of the Goldstone theorem. (**a**) The principle of SSB. (**a**) Left: the ground state of the system (minimum of the energy potential function) is not at 0-energy equilibrium. (**a**) Center: This induces the possibility that, despite the overall symmetry being conserved, the system (blue ball) can "choose" between different (two, in the example) states, each "locally" breaking the global symmetry. (**a**) Right: Indeed, the system stays in one of the two available ones at the same ground state to determine an SSB. (**b**) Effectively, at the QV ground state, there are infinitely many possible SSBs to determine the classical "sombrero hat" energy potential diagram of the Goldstone theorem.

Therefore, the infinitely (not denumerable) many QV conditions (or "QV degenerate states") compatible with its ground state ultimately exist because of the instauration of these *long-range correlations* among quantum fields at their ground state. Moreover, the SSB principle holds not only in the relativistic (microscopic) domain of QFT but also applies to non-relativistic many-body systems in condensed matter physics of chemical and biological systems (neural systems included), and even at the cosmological scale.

In other words, this discovery of the dynamically generated *long-range correlations* among quantum fields deeply changed the fundamental physics (QFT) of the elementary particles (at the microscopic level) and the condensed matter physics (at the macroscopic level). Indeed, these correlations are mediated by the *Nambu–Goldstone bosons* (NG-bosons) as quanta not of the energy exchanges (interactions) such as the gauge-bosons of the SM (see Figure 5) but as quanta of the *long-range correlation waves* among quantum fields propagating in the vacuum to be associated to the different *coherent modes of oscillating in phase* of the quantum fields [114,115], i.e., the phenomenon of the so-called *quantum entanglement* [21].

Effectively, the short note in which Higgs synthetically expressed his revolutionary idea [113] was published in 1964, two years after the usage made of it by Salam, Goldstone, and Weinberg to solve the electro-weak SSB in [114]. Indeed, Higgs in his note refers to

this paper. However, Higgs illustrated the core of his idea in a lecture given at Princeton in 1961, at which both Weinberg and Goldstone were present. The "Higgs field" hypothesis, indeed, solved the fundamental conundrum of the electro-weak unification. Specifically, it solved the problem of how the *Z-W bosons,* the gauge-bosons of the weak force, can acquire their tremendous mass (of the order of the mass of an iron atom!), so violating the Goldstone theorem for which the gauge-bosons must be massless, just like the *photons* and the *gluons*—the gauge-bosons of the electromagnetic and the strong forces—are (see Figure 5).

Indeed, the Goldstone theorem is also strictly related to Yoichiro Nambu's application of the SSB principle in 1960–61 to model the *chiral symmetry breaking* in *quantum chromodynamics* [115–117]. Specifically, in the QFT of the "strong force" with its three different charges ("colors"), through which the quarks interact. What is amazing for non-physicists such as us is that the largest part (more than 99%) of the mass of protons, and then of atoms, and finally of our bodies depends on the chiral symmetry breaking by which protons (and neutrons) can acquire the largest part of their masses by the "strong interactions" of the quarks constituting them. This is despite "gluons"—that is, the gauge-bosons of the strong force field, see Figure 5—being per se massless such as photons. This sends us back again to the role of the "Higgs field" [113], applied this time not to the electro-weak interaction symmetry breaking but to the electro-strong interaction "chiral" symmetry breaking (see [107] for its intuitive illustration). For this fundamental work, Nambu was awarded the Nobel Prize in Physics in 2008. In this way, SSBs of the Goldstone theorem and the Higgs field play an essential role in the construction of the "local gauge theories" of the three fundamental quantum (strong, electromagnetic, weak) forces of SM.

The quanta of the long-range correlation waves are named *Nambu–Goldstone (NG) bosons.* Now, the NG-bosons are with mass even though always very small (if the symmetry is not perfect in finite spaces), or *without mass at all* (if the symmetry is perfect, in the abstract infinite volume of functional analysis). The lesser the inertia (mass) of the correlation quantum, the greater the distance on which it can propagate, and hence the distance on which the correlation (and the ordering relation they determine) constitutes itself (see Note 21).

To sum up, the main novelty introduced by the Goldstone theorem in this QFT picture is that each of the QV degenerate states constitutes an SSB of QV at its ground state. Each SSB, in turn, corresponds to the "spontaneous" instauration of *long-range correlations* among force fields in QV. Therefore, they can display *collective behaviors* that make their treatment in terms of *individuals* meaningless. This implies a deep *ontological paradigm shift* in modern fundamental physics.

This shift can be summarized as follows:

1.   Firstly, all this means that each massive or non-massive "elementary particle" of the SM, both fermions (quarks, neutrinos, and electrons) and gauge-bosons (gluons, photons, *Z-W* bosons), and the Higgs-boson, are considered in QFT as *quanta* of their respective fields (see Figure 5). This ontological stance is consistent with the passage to the mathematical formalism of the so-called *string theory*, where a particle is not represented by a "point" and its motion as an "unidimensional trajectory" in the state space, but it is represented as a *vibrating string* and its motion as a *bidimensional brane*, where the intensity of the string vibration is proportional to the energy of the associated field.

2.   Secondly, in QFT, an uncertainty relation holds and then a *particle-wave duality*, similar to Heisenberg's one of QM relating the *statistical* uncertainty between the momentum and position of particles (see Equation (1), above). Effectively, in QFT, in the light of the Goldstone theorem, the uncertainty and then the particle-wave duality concerns the number of the field quanta *n,* with respect to the field phase $\phi$, namely:

$$\Delta n \Delta \varphi \geq K \qquad (4)$$

where *K* is a quantization constant related to the type of long-range correlation involved. If ($\Delta n = 0$), $\phi$ is undefined, so that it makes sense to neglect the waveform aspect in favor of the individual, particle-like behavior. On the contrary, if ($\Delta \phi = 0$), *n* is undefined because an extremely high number of quanta are oscillating together according to a well-defined phase, i.e., within a given phase coherence domain. In this case, it would be nonsensical to describe the phenomenon in terms of individual particle behavior since the *collective modes* of the field prevail. We already presented a condensed matter physics example of this phenomenon, that is, the ferromagnetic phase (see Figure 4).

3.　Thirdly, another fundamental unifying notion, not only with respect to quantum physics but also with respect to quantum biology and quantum computing as far as both are based on QFT, is the notion of the *NG-bosons*. Of course, they are not "gauge bosons", quanta of energy, such as the bosons of the four fundamental forces of the SM (see Figure 5), since they are quanta of the coherent modes of being in phase of the quantum fields. Therefore, they appear in all the equations of QFT related to the instauration of long-range correlations among quantum fields. In this way, it is a further consequence of the Goldstone theorem that any long-range phase coherence among quantum fields related to SSB of QV at its ground state has its "fingerprint" in the unique *countable* value $\mathcal{N}$ of a *given condensate of NG-bosons*. Now, despite "these correlation quanta" being real particles, observable with the same techniques (diffusion, scattering, etc.) of the other particles, nevertheless, because their mass is in any case negligible (or even null), *their condensation does not imply a change in the energy state of the system.* This means that, if the symmetric state is a possible ground state (a minimum energy state or a degenerate "vacuum" of a QFT system), the coherent state, after the symmetry breakdown, also remains in *a state of minimal energy* to be *stable* in time. In the macroscopic terms of classical kinematics, it is representable as a *stable (chaotic) attractor* of the overall dynamics [118]. Or, more properly, in the formalism of QFT, this phenomenon is the core of the principle of *foliation of QV* at its ground state for different values $\mathcal{N}$ of a given condensate of NG-bosons, as a "robust principle of 'construction' and 'memory' used by nature to generate ever more complex systems [9,34], as we already recalled and will explain formally using the powerful colimit operation of the CT mathematics (see Section 4.8).

4.　Fourthly, the Goldstone theorem and the SSB principle related to the instauration of long-range correlation applies both to the relativistic QFT of atomic and subatomic physics at the *microscopic level*, and it applies to many-body physics of condensed states of matter at the *macroscopic level*. This constitutes a fundamental *analogy*—to use Nambu's words [116,117]—between these two levels of matter organization. Indeed, the long-range correlations, related to the instauration of phase coherence domains among the involved matter fields and their quanta, all imply a *dynamic re-definition of the metrics* characterizing the system dynamics and its properties. In this sense, the macroscopic phenomena of condensed matter physics related to system phase transitions have their own proper explanation at the *microscopic* quantum level [9].

To sum up, if any phase transition in physics is characterized by an *order parameter* that is the physical magnitude—generally, a statistical density distribution, whose sudden change at the phase boundary characterizes the phase transition [22]—in dissipative QFT, a given condensate of NG-bosons plays the role of a *dynamic control parameter*. Changing its numerical value $\mathcal{N}$ means that the quantum fields can be subject to different dynamic regimes, with different collective properties, and hence with different collective behaviors and functions, *at the same ground state* of the quantum fields.

From the standpoint of theoretical physics, the demonstration of what we intend by saying that condensates of NG-bosons act as a "control parameter" of the phase transition among different phase coherence domains in the same QV-ground state at a temperature *T* > 0 can be found in ([9], pp. 166–169). In the equations explained in these pages, indeed, it is demonstrated that the simple emission-absorption of "a few of" (a finite number of) NG-bosons can also induce a phase transition, *without any energy waste*! Ontologically, they

are "quanta of *form*" (ordering relations) and not "quanta of *matter*" (mass-energy) such as fermions and gauge-bosons.

In this way, NG-bosons in condensed matter physics acquire different names, according to the different topological phases of matter they control.

For instance, in solid state physics (mechanics), the NG-bosons are named *phonons*, as far as they are quanta of the collective coherent modes of mechanical (elastic) oscillations (vibrations) of the molecules. In this case, indeed, the symmetry breaking concerns the *Galilean spherical symmetry* in the propagation of the mechanical vibrational motion of molecules, according to which these vibrations propagate *casually* (i.e., satisfying a spherical symmetry) in whichever direction of the 3D-space. The breaking of such a symmetry determines either their *longitudinal* coherent propagation, corresponding macroscopically to the phase transition to the "liquid state" of the collective behavior of the moles of some material (e.g., the liquid state of a water flow), or the *longitudinal and transverse* coherent propagation, corresponding to the "solid state". In this latter situation, in the case of a rigid crystalline lattice of oscillating atoms/molecules, their coherent oscillation modes determine the regular distribution according to a periodic law of the particles in the lattice to determine dynamically the regular geometric structure of a crystal (e.g., the beautiful geometries of the icy state of water in a snowflake).

Another example is the phase transition to the *magnetic phase* of some metals, which we already presented intuitively in Figure 4, where NG-bosons are named *magnons*. In this case, indeed, the broken symmetry is the rotational symmetry of the *magnetic* dipole of the electrons, and the macroscopic phenomenon of "magnetization" consists of the correlation among all (most) electron spins, so that they "choose", among all the directions, the one correct for the magnetization vector.

Finally, in biological matter and water, in which only the biological molecules are active (this is the deep reason why more than 80% of our bodies is made of water, and more than 90% of our molecules are water), the NG-bosons are named *polarons*. Indeed, the broken symmetry in this case is the rotational symmetry of the *electrical* dipoles characterizing water and organic molecules. The coherent modes of propagation of dipole currents are indeed the dynamic "secret" by which distant bio-molecules "can feel each other" in the cell inner-outer "watery" environment to constitute the physical basis of the ordered collective modes, which ultimately consist of both the complex structures of biological matter and the ordered sequences of chemical reactions of a biological function [23].

To conclude, please note again this ontological consequence for fundamental physics, which we already introduced descriptively at the end of Section 2.5. It is similar to if the matter dynamics by long-range correlations and the relative *phase coherences of the quantum fields* corresponding to as many SSBs of QV defined a *new property and/or function and/or predicate of matter*: "being liquid", "being solid", "being magnet", "being organic", etc. In a proper sense, which will emerge clearly when we discuss more deeply the common *coalgebraic semantics over NWF-sets*, both of QFT in physics and f Kripke's coalgebraic modal semantics in logic, it is similar to if a physical phase coherence constitutes the *logical/mathematical domain* of a given function/predicate.

This physical process can only be properly formalized in the (ante-predicative) CT metalanguage. In it, the constitution of the domain/codomain of objects for a given morphism (function/predicate) depends on the primitive of the $\mathrm{dom}(\cdot)/\mathrm{cod}(\cdot)$ mappings and not on the set-elementhood principle of the (predicative) ST metalanguage, taking the $\in$-membership predicate as a primitive (see Note 1, and Section 2.1).

*4.5. The Non-Commutative Coalgebraic Modeling of QFT Dissipative Systems*

Having given a descriptive survey of some main notions of QFT, let us deepen the understanding of some aspects of the mathematical formalism involved in this subsection. In the quantum physics mathematical formalism, the *Hopf algebras* play an essential role in performing mathematical computations on a lattice of quantum numbers **k** associated with given quantum variables (e.g., the energy *E* or the momentum *J* of particles).

Mathematically, a Hopf algebra $H$ is a "bi-algebra" because it is characterized by two types of operations, i.e., *coproducts* of the coalgebra: $H \rightarrow H \times H$, and *products* of the algebra: $H \times H \rightarrow H$. Both products and coproducts *are commuting*, and defined over a *field K* with a *K*-linear *map S*: $H \rightarrow H$, or *antipode,* sending dually commuting *coproducts* over commuting *products*, and *counits ε* over *units η*, and vice versa, so that the diagram of Figure 7 commutes.

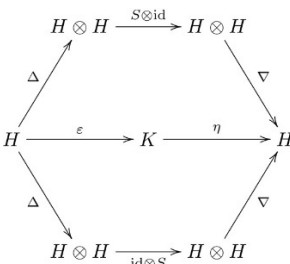

**Figure 7.** Commuting diagram of a Hopf bialgebra.

The role of a Hopf bialgebra in QM calculations over a lattice of quantum numbers emerges immediately when we consider that the "algebraic half" $H \times H \rightarrow H$ of the Hopf bialgebra applies when we must calculate, for instance, the energy $E$ of a single particle; whereas the "coalgebraic half" $H \rightarrow H \times H$ applies when we have to calculate the total energy of two particles superposed in the same quantum state $E = E_1 + E_2$ (coproducts are effectively disjoint sums, as we know). In this case, the *commutativity* of the coproducts also makes sense because the total energy of the quantum state does not change by interchanging between themselves the two particles.

However, in dissipative QFT or, more generally, when we deal with open systems, the total energy $E$ depends on the system and its thermal bath *balanced* contributions, which evidently cannot be treated on the same algebraic basis. This means that the commutativity of the coproduct terms for calculating the total energy cannot hold any longer. Indeed, the terms of the coproducts represent here, necessarily in a *non-interchangeable* and then in a *non-commutative* way, the system state energy value and the associated thermal bath energy value. In these cases, the proper algebraic tool is the *q-deformed Hopf coalgebras*, with *non-commutative coproducts*, strictly related to the Bogoliubov transform (Section 4.3), because the $q$ deforming ("squeezing") parameter is a *thermal parameter* corresponding to the inverse of the $θ$ "mixing angle" of the Bogoliubov transform [9,34].

Therefore, the non-commutative coproducts for a system state $a$ and for its correspondent thermal-bath state $\tilde{a}$ are: $\Delta a_q = a \times q + q^{-1} \times a \equiv a \times q + q^{-1} \times \tilde{a}$. The $q$-deformed Hopf coalgebra mapping $H \rightarrow H \times \widetilde{H}$ thus describes the *doubling* of the degrees of freedom system-thermal bath, i.e., $a \rightarrow \{a, \tilde{a}\}$, and for the phase space $\mathcal{F} \rightarrow \mathcal{F} \times \widetilde{\mathcal{F}}$, with the operators $a$ and $\tilde{a}$, respectively [9,34]. The "toy-model" of the quantum doubling system-thermal bath that is able to make it immediately intuitive is given by *two damped quantum harmonic oscillators* [37], which is the dissipative counterpart of the standard "two coupled quantum harmonic oscillators" for representing the quantum superposition in the standard non-dissipative case of QM (see Section 3.3). The dissipative model therefore plays an essential role in modeling Gerard 'tHooft's approach to quantum gravity that explains quantization in terms of "information dissipation" in a deterministic system [119]. What is worth emphasizing, anyway, is that by such a doubling, the composite system recovers the essential *unitarity condition* of the Hamiltonian: indeed, it is globally a *closed system*. Consequently, we can use the "doubling of the Hilbert spaces" $\mathcal{H} \rightarrow \mathcal{H} \times \widetilde{\mathcal{H}}$ to recover the *canonical* or *Hamiltonian representation* of a dissipative quantum system.

We recall, indeed, that the *canonical* representation of whichever dynamic system, both in classical and quantum physics, is in terms of its *Hamiltonian function.* It gives the total energy of the dynamic system concerned because it is an extension of the *Lagrange equation* to the study of the evolution in time of a dynamic system. Since the Hamiltonian canonically represents the dynamic system as being energetically closed, this implies that if

we want to write the Hamiltonian of a dissipative system, we must also include in it the thermal bath degrees of freedom to grant that it is "globally" energetically closed.

To sum up, from the *algebra doubling* intrinsic to the coalgebraic representation of the balanced state of an open system, we can naturally pass to the canonical, Hamiltonian, representation of the open system dynamics by extending the algebra doubling to its Hilbert space, i.e., $\mathcal{H} \to \mathcal{H} \times \widetilde{\mathcal{H}}$.

Therefore, because we are interested in the phenomena of QFT in condensed matter physics where the electromagnetic force field and hence photon condensates are involved, we can limit ourselves to the Bogoliubov transform for CCRs of bosons and not CARs of fermions (Section 4.3). In the CCR case, in the "doubled" Hilbert space, what are commuting are not, of course, the doubled (system/thermal bath) states but the associated operators $A(\theta), \widetilde{A}(\theta)$, according to the following relations [34]:

$$A(\theta) = A \cosh\theta - \widetilde{A}^{\dagger}\sinh\theta \tag{5}$$

$$\widetilde{A}(\theta) = \widetilde{A} \cosh\theta - A^{\dagger}\sinh\theta \tag{6}$$

Consequently, CCRs are:

$$\left[A(\theta), A(\theta)^{\dagger}\right] = 1, \ \left[\widetilde{A}(\theta), \widetilde{A}(\theta)^{\dagger}\right] = 1 \tag{7}$$

All other commutators are equal to zero. Equations (5) and (6) are nothing but the *Bogoliubov transformations* for the $\left\{A, \widetilde{A}\right\}$ couple, evidently applied in a *reversed* way between the system (5) and its thermal bath (6), to signify the *energy balance* for a given temperature $T > 0$ (= boundary condition) of the QV ground state, characterizing any *phase transition* of a dissipative QFT system. They, I repeat, are concerning phase coherence domains, including the system and its thermal bath states, i.e., they are "entangled" in only one phase coherence domain to constitute only one dissipative system.

Indeed, the "reversal of the arrows" has an immediate physical significance in the correspondent *reversal of the energy arrow* characterizing the energy balance in any dissipative system in *far-from-equilibrium conditions*. This allows us to define the powerful principle of the "QV foliation" as an *ordered family*—effectively, a "(sub)category" in CT formalization (see Section 4.8)—of "unitarily inequivalent representations" of QV $|0\rangle$ at its ground state, such as many pairs of doubled Hilbert spaces univocally indexed in $\theta$ (and/or in $q$, in the case of the correspondent category of the $q$-deformed Hopf coalgebras). Indeed, there exists a fundamental relationship between the $q$-deformation (or "squeezing") *thermal* parameter of Hopf coalgebras in the calculations over a specific set $\mathbf{k}$ of quantum numbers and the "$\theta$-mixing angle" of the Bogoliubov transform, i.e., the $\theta$-set dynamically *labeling* the different "vacua" (QV-foliation) is defined as: $\{\theta_{\mathbf{k}} = \ln q_{\mathbf{k}}, \ \forall \mathbf{k}\}$.

Quoting directly from [34], let us introduce the QV "splitting" $|0\rangle \equiv |0\rangle \times |0\rangle$ as denoting the vacuum "annihilated" by the Bogoliubov operators $A$ and $\widetilde{A}$ : $A|0\rangle = 0 = \widetilde{A}|0\rangle$, respectively. This means that $|0$ is not annihilated by $A(\theta)$ and $\widetilde{A}(\theta)$ of Equations (5) and (6). On the contrary, the vacuum annihilated by the Bogoliubov operators $A$, $\hat{A}$ is:

$$|0(\theta)\rangle_{\mathcal{N}} = e^{i\sum_{\mathbf{k}} \theta_k G_k}|0\rangle = \prod_{\mathbf{k}} \frac{1}{\cosh\theta_k} \exp\left(\tanh\theta_k A^{\dagger}\widetilde{A}^{\dagger}\right)|0\rangle. \tag{8}$$

where the subscript $\mathbf{k}$ refers to a set of quantum numbers for quantum variables; $G_{\mathbf{k}} \equiv -i\left(A_{\mathbf{k}}^{\dagger}\widetilde{A}_{\mathbf{k}}^{\dagger} - A_{\mathbf{k}}\widetilde{A}_{\mathbf{k}}\right)$ is the generator of Bogoliubov transformations of Equations (5) and (6) for a whole category of *doubled Hilbert spaces*, effectively, a sub-category for the Bogoliubov functor of the category **Hilb**, as we see in Section 4.8; the subscript N refers to a given *unique* value (its "fingerprint"!) of the condensate of NG-bosons characterizing an annihilated QV state (i.e., in our case, a phase coherence of electromagnetic fields emerging from SSB of QV, corresponding to a dissipative quantum system); and $\theta$ denotes, as we know, the set $\{\theta_{\mathbf{k}}, \forall \mathbf{k}\}$, and $_{\mathcal{N}}\langle 0(\theta)|0(\theta)\rangle_{\mathcal{N}} = 1$.

Finally, it is possible to demonstrate that in the infinite volume limit $V \to \infty$, the phase space splits *into infinitely many inequivalent representations*, as the Haag theorem previews. In dissipative QFT, however, each of them is *dynamically labeled* by a specific $\theta$—set $\theta_{\mathbf{k}} = \ln_{q\mathbf{k}}, \forall \mathbf{k}$. This is exactly what we intend by the notion of *QV-foliation* in QFT, as a powerful dynamic tool of "construction" used by nature of *complex* systems in physics and *memory* in biology and neuroscience, given that all biological systems are dissipative systems.

*4.6. The QV-Foliation and the Principle of Doubling of the Degrees of Freedom as a Solution of the Complexity Issue in Fundamental Physics*

The *QV-foliation* emphasizes the advantages of modeling QFT systems as dissipative systems, the main principles of which we illustrate in this section. This advantage can be synthesized in the following statement:

> *Because of the principle of the "doubling of the degrees of freedom" (DDF), the canonical Hamiltonian, i.e., the finite number of degrees of freedom for a faithful representation of the system, must not be supposed any longer such as in the SQ modeling of QFT (Section 3.3) but (thermo)dynamically justified in far-from-equilibrium conditions of many-body physics.*

Any dissipative system indeed must satisfy the energy balance condition corresponding to the *ground state* of the overall dynamics. This means that a ground state is a *0-sum total energy condition $E_{tot}$* between the system energy $E_{sys}$ and the environment (thermal bath) energy $E_{env}$, i.e., $E_{tot} = E_{sys} - E_{env} = 0$. However, the same ground state $E_{tot} = 0$—and this is the deep significance of the Goldstone theorem—is compatible with different *orderings* of the concerned "order parameter", and then with different *structural conditions* of the balanced system, related to different *NG-boson condensates* N or *long-range correlations* for each N, including in the dissipative case both the system and its thermal bath, and corresponding in QFT to as many *degenerate states* of the QV "coexisting without interferences" at its ground state.

The notion of *QV-foliation* is therefore that "an ordered hierarchy" of different QV degenerate states indexed in N can coexist without interfering with each other in the same ground state of the balanced system. This is the basis at the level of QFT as the fundamental physics of condensed matter systems of the notion of *system complexity*, I. Prigogine's notion of *dissipative structures* is included [110,120]. A complex system can indeed be intuitively described as a system characterized by a hierarchy of different "emerging" levels of structural organization, each with its own "order parameter" [121]. How they are ordered and how they emerge using only one framework of reference at the level of fundamental physics remains unexplained *without referring to the notion of QV-foliation in QFT.*

In a word, as D. K. Morr stated on the *Science* journal [122] to introduce two further successes of QFT in explaining high-temperature superconductivity, and whose reports are published in the same issue of the journal [123,124], QFT of the dissipative systems is the only theoretical tool "for lifting the fog of complexity" both at level of fundamental physics and fundamental biology.

Moreover, because the principle of the "infinite QV-foliation" also implies per se an infinite foliation of the representational Hilbert spaces as previewed in the Haag theorem—and happens not casually for the infinite inclusion chains of NWF-set—in thermal QFT, it is possible "to exorcise", at least in a partial but significative way (see Section 4.8), such a "nightmare" for mathematicians, physicists, and computer scientists without any reference to an extrinsic "observer", i.e., the supposed knowledge of the system Hamiltonian.

In fact, the *minimum free energy* can be used here as a *dynamic choice criterion* of admissible states of the doubled Hilbert space, differently from quantum thermodynamics based on statistical mechanics [125], where the stability is studied at *equilibrium* in the so-called *asymptotic condition* of the perturbative techniques so that only open systems "near-to-equilibrium" can be studied by such a formalism [126].

On the contrary, the possibility in QFT of using the minimum free-energy function as a *dynamic selection criterion of physically admissible states* makes the notion of the *doubling of the degrees of freedom* (DDF) effective.

A fundamental consequence of the DDF principle is that, in finite temperature QFT, a fundamental statistical tool consists of computing the thermal average expectations of some observable, say $\mathcal{O}$. This requires the computation of "traces", with the trace of a square matrix **A**, $\langle n \times m \rangle$, which is denoted tr(**A**), being defined as the sum of elements on the main diagonal (from the upper left to the lower right) of **A**. Typically ([34], p. 45):

one deals with matrix elements of the type $\mathcal{O}_{nm} = \langle n|\mathcal{O}|m\rangle$, with orthonormal states $\langle n|m\rangle = \delta_{nm}$ in the Fock space and $H|n\rangle = E_n$.

Here, $H|n\rangle$ is the statistical expectation value with respect to the state $n$ and $E_n$ is the associated energy value. In this way,

the trace $\sum \mathcal{O}_{nm}$ is obtained by multiplying the matrix elements by $\delta_{nm}$ and summing over $n$ and $m$.

Now, in the standard SQ approach to QFT, the supposition that we already know the Hamiltonian of the system (Section 3.3) consists in introducing $\delta_{nm}$

as an external (to the operator algebra) computational tool, which essentially amounts in picking up "by hand" the diagonal elements of the matrix and summing them. One may instead represent the in terms of the doubled tilde-states $|n\rangle$, $\langle \widetilde{n}|\widetilde{m}\rangle = \delta_{nm}$ with $\widetilde{H}|\widetilde{n}\rangle = E_n|\widetilde{n}\rangle$ and $\widetilde{H}\omega\widetilde{A}^\dagger\widetilde{A}$ ([34], p. 45).

Then, using the notation $|n,\widetilde{n}\rangle = |n\rangle \times |\widetilde{n}\rangle$, since $\mathcal{O}$ operates only on the non-tilde states, we have:

$$\langle n,\widetilde{n}|\mathcal{O}|m,\widetilde{m}\rangle = \langle n|\mathcal{O}|m\rangle\langle\widetilde{n}|\widetilde{m}\rangle = \langle n|\mathcal{O}|m\rangle\delta_{nm} = \langle n|\mathcal{O}|n\rangle \qquad (9)$$

In other terms, the DDF principle in QFT dissipative systems, requiring the doubling of the Fock state spaces, means that the introduction of $\delta_{nm}$ is *intrinsic* to the operator algebra because the determination of the orthonormal basis of the Fock space and then of the representation doubled Hilbert space is intrinsic to the dynamics of the system. Indeed, we are dealing with Pauli matrices, where

in the states $|m,\widetilde{m},\rangle$ $m$ is an integer number. We thus have a Kronecker delta in Equation (9) and not a Dirac delta ([34], p. 45).

Moreover, what is also relevant for our aims is that to each set of degrees of freedom $\{A\}$ and its "entangled doubled" $\left\{\widetilde{A}\right\}$ corresponds a *unique integer number* $\mathcal{N}$, i.e., $\mathcal{N}_A, \mathcal{N}_{\widetilde{A}}$, so that $|\mathcal{N}|$ *identifies univocally*, i.e., it *dynamically labels*, a given *phase coherence domain* of the quantum fields, including the system and its environment. This depends on the fact that in the QFT mathematical formalism, as we know, the number N is a numeric value expressing the NG-bosons condensate value, on which a phase coherence domain *directly depends* such as on its "control parameter". In an appropriate *set theoretic interpretation*, because for each "phase coherence domain" $x$, $|\mathcal{N}|$ effectively *identifies univocally* such a domain, it corresponds to an "identity function $Id_x$" that, in a "finitary" coalgebraic logical calculus, corresponds to the *predicate satisfied by such a domain because it univocally identifies it*. We discuss more extensively in Section 4.8 this fundamental *dynamic* indexing principle using the powerful operation of *colimit* as a comma category of indexed functors, which in our case is the comma category of the Bogoliubov functors for the generator $G$ of all the Bogoliubov transform (see Equation (8), Section 4.8, and overall Appendix A).

Finally, the DDF principle justifies, as discussed elsewhere [34], the interpretation of the *maximal entropy* in a QFT "doubled" system as *a semantic measure of information*, i.e., as a statistical measure of *local truth* in the CT coalgebraic BAO logic, as we see in Section 5.2. This is strictly related to the notion of *doubled qubit* or *semantic qubit* characterizing the quantum computations in these classes of dissipative quantum systems. Indeed, in the QFT "composed Hilbert space", including the thermal bath degrees of

freedom, $\widetilde{A}$, i.e., $\mathcal{H}_{A,\widetilde{A}} = \mathcal{H}_A \otimes \widetilde{\mathcal{H}}_{\widetilde{A}}$, to calculate the static and dynamic entropy associated with the time evolution generated by the free energy, i.e. $|\phi(t)\rangle, |\psi(t)\rangle$, of the qubit mixed states $|\phi\rangle, |\psi\rangle$, one needs to double the states by introducing the tilde states $\big|\widetilde{0}\big\rangle$ and $\big|\widetilde{1}\big\rangle$, relative to the thermal bath, i.e., $|0\rangle \rightarrow |0\rangle \otimes \big|\widetilde{0}\big\rangle$, and $|1\rangle \rightarrow |1\rangle \otimes \big|\widetilde{1}\big\rangle$. This means that such a QFT version of a qubit effectively implements the CNOT (controlled NOT) logical gate, which flips the state of the qubit, conditional on a *dynamic* control of an effective input matching [34,127,128]. Because it is demonstrated that it is possible to recursively calculate in such a computational architecture the *Fibonacci series,* this architecture is a particular type of (topological) quantum implementation of a "golden machine" (see [129] for a different model of this type of machine related to the "non-commutative anions" for the Hall-effect), i.e., it is a "universal computer" (see [34,129] for further details with the necessary bibliography).

*4.7. The Cognitive Relevance of the QFT Modeling of Non-Linear Brain Dynamics*

The main interest of Walter Freeman (1927–2016) during the many decades of active research with his group in the neuroscience lab of the University of California at Berkeley was the explanation of the neural basis of *intentional behavior* in humans and animals. More specifically, it was the explanation for the *dynamic integration* of the "mosaic" of the different brain modules of the cortex, each performing different sensory or motor functions. Effectively, this aim was pursued by Freeman using for the electroencephalogram (EEG) and electrocorticogram (ECoG) computational signal analysis, not the Fourier transform but the *Hilbert transform.* Indeed, while the former decomposes the signal into its frequency components, the Hilbert transform decomposes the signal into the analytical amplitude $A(t)$ and its analytic phase $\varphi(t)$. In this way, he discovered the empirical evidence of the massive presence of patterns of AM phase-locked oscillations in the *background activity* of the brain. This is registered by EEG, but often it is filtered as "noise" by neurophysiologists exclusively interested in studying the neuron (arrays) spike activity and their synaptic circuitry [130].

On the contrary, these *long-range correlation patterns* have their neural medium in the cortex *neuropil*, that is, "the dense felt-work of axons, dendrites, cell bodies, glia and capillaries forming a superficial continuum 1–3 mm in thickness over the entire extent of each cerebral hemisphere" in mammals ([39], p.95). These correlation patterns are intermittently present in resting and/or awake subjects, and the same subject actively engaged in cognitive tasks requiring a goal-directed interaction with the environment. These "wave packets" extend almost instantaneously over coherence domains covering much of the hemisphere in rabbits and cats, and regions of linear size of about 19 cm in human brains [131–134].

During the largest part of his research life, Freeman tried to find a suitable physical explanation of this amazing phenomenon, as I can personally testify during the many meetings I had with him during the 1990s because we were both interested at that time in studying the overall chaotic dynamics of natural (him) and artificial (myself) neural networking. I recall, therefore, when he enthusiastically announced to me, during one of our last personal meetings in 2008, that he solved the problem. Indeed, Vitiello demonstrated mathematically that the trajectories in the phase space between different phase coherence domains, along which a dissipative quantum system moves at the *microscopic* level, display a chaotic character at the *macroscopic* level [118]. Effectively, since 2002, a fruitful collaboration has started between Freeman and Vitiello, leading to a series of common publications till the year of Freeman's death in 2016.

The final result is that Freeman's discoveries constitute at the moment the more extended experimental confirmation at the level of condensed matter physics of the main notions of QFT for the "dissipative brain" model (see [39,40] for a synthesis). This excited my interest in searching for the most suitable *intensional* logic to associate with this modeling of the *intentional* behavior in brain dynamics conceived as a natural computational system

(see Section 1.1). Additionally, this is surely the *coalgebraic semantics of Kripke relational logic* in the CT framework, given also the non-casual but essential role that coalgebras and coproducts generally play in QFT computations (see Section 4.5).

> To conclude, in this "holistic" framework of intentional behaviors, the concept of the boson carrier and the boson condensate does more; it enables an orderly and inclusive description of the phase transition that includes all levels of the macroscopic, mesoscopic, and microscopic organization of the cerebral patterns that mediate the integration of the animal with its environment, down to and including the electric dipoles of all the myriad proteins, amino acid transmitters, ions, and water molecules that comprise the quantum system. This hierarchical system extending from atoms to the whole brain and outwardly into the engagement of the subject with the environment in the action-perception cycle is the essential basis for the ontogenetic emergence and maintenance of meaning through successful interaction and its knowledge base within the brain. By repeated trial-and-error, each brain constructs within itself an understanding of its surroundings, which constitutes its knowledge of its own world that we describe as its *double*. It is an *active* mirror [24] because the environment impacts onto the self independently and reactively. The relations that the self and its surround construct by their interactions constitute the meanings of the flows of information that are exchanged during the interactions ([39], p. 108).

To intuitively understand what all this means and specifically the notion of brain–environment "entanglement" or "active mirroring" in QFT, as the cognitive counterpart of the DDF principle, it is noted, for example, that as the elementary evidence humans receive visual stimulations from the optical part of the electromagnetic spectrum while owls receive from the infrared part (see Figure 8). Intuitively, but consistently in the light of the DDF principle, this is because the cone and the rod cells of the human retina are oscillating in phase—so to constitute "only one phase coherence domain"—with the optical (visible light) part of the electromagnetic spectrum in the environment while the owl retina cells are in phase with the infrared part of the spectrum.

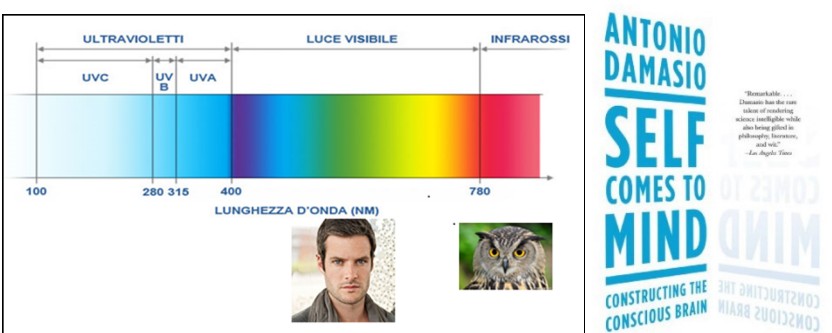

**Figure 8.** Intuitive representation of the "active mirroring" brain–environment in cognitive neuroscience, consistent with Damasio's biological foundation of intentionality on the physical homeostasis principle—as suggested metaphorically in the promotional picture of Damasio's book on the right—that holds in biology, from the epigenetic principle of cell specialization, till brains and beyond (see text).

All this is also consistent with Antonio Damasio's proposal of finding the biological foundation of *intentionality* in the *homeostatic non-linear mechanisms* characterizing all living systems as dissipative systems. Living systems are endowed, differently from non-living ones, with complex non-linear *self-regulation* processes [135]. The complex self-organizing "intentional" structure of living dissipative systems "entangled" with their environments thus ranges from the elementary level of the epigenetic mechanisms in cell specialization to the formation of tissues, organs, nervous systems, brains (see [120]), and even beyond to model the social behavior ("the society of brains") of animals [41,136]. The reader can

usually refer to [137] to synthetically deepen the relationship between Damasio's and QFT approaches to the foundations of intentionality in biological and neural systems.

Finally, this physical foundation of the intentional behavior in biological and neural systems is consistent with the so-called interpretation of the *extended mind,* locating it not in the brain but in the physical (material and informational) intercourse between brains and their environments, given that a *living* brain is a *dissipative* brain [138,139]. This intercourse effectively ends only with the death of the animals when they finish being a dissipative system, stable in far-from-equilibrium conditions, and they are in equilibrium with their environment.

*4.8. Categorical Definition of the QFT Systems as Initial (Terminal) Objects of Comma Categories (Universal Morphisms)*

This final subsection is devoted to discussing the fruitfulness of modeling the dissipative QFT and its non-commutative coalgebraic formalism in the framework of CT. Indeed, in the light of what is discussed in this section, it is evident that the *q*-deformed Hopf coalgebras form a category for the Bogoliubov functor $\mathcal{B}$ ([34], pp.47–48), i.e., **qHCoalg($\mathcal{B}$).** Moreover, because of the existence of a common generator of all Bogoliubov transforms $G_{\mathbf{k}} \equiv -i\left( A_{\mathbf{k}}^{\dagger}\widetilde{A}_{\mathbf{k}}^{\dagger} - A_{\mathbf{k}}\widetilde{A}_{\mathbf{k}} \right)$ that are part of the Hamiltonian of the system, which is strictly related to the NG-boson condensate value $\mathcal{N}$ univocally indexing each phase coherence domain or degenerate state of QV $|0(\theta)\rangle_{\mathcal{N}}$ (see Equation (8)), the consequent principle of the QV-foliation can be properly formalized in CT by the powerful *colimit operation* (i.e., "direct limit operation" in mathematical analysis) applied to the category of the QFT coproducts for the Bogoliubov functor. Indeed, we have all the "ingredients" satisfying the categorical definition of QFT systems as objects that are *universal morphisms,* specifically, as initial (colimits) or terminal (limits) objects of *comma categories* (see Appendix A). Here, a comma category is a category $\mathcal{C}$ constituted by morphisms relating objects belonging to other two categories $\mathcal{A}$ and $\mathcal{B}$ having $\mathcal{C}$ as their common functorial target, i.e., $\mathcal{A} \xrightarrow{F} \mathcal{C} \xleftarrow{G} \mathcal{B}$ (see Definition A1).

It is worth emphasizing here the physical, mathematical, and then ontological relevance of this categorical construction, by which whichever object $c$ in a category $\mathcal{C}$ is a functor, in turn, mapping from the category **1** with only one object ($*$) (a "limit point") and its identity morphism $1_*$ to the category $\mathcal{C}$. In this way, $c$ can always be represented as the initial/terminal object of a "triadic" comma category of morphisms (functors): $\mathbf{1} \xrightarrow{c} \mathcal{C} \xleftarrow{F} \mathcal{B}$. This diagrammatic construction leads to the universal formalization of the "constructive power" of nature of new systems by the general mechanism of SSB, where the new generated system corresponds to a "phase coherence domain" of the quantum fields at their ground state (=quantum vacuum), and then to the one-object category **1** of the comma category construction, i.e., mathematically, they are limit points of the *compact, Hausdorff, totally disconnected* topological spaces of the $\sigma$-algebras on which the probability spaces of quantum systems in physics and the "Stone spaces" of the RTBA in logic are defined (see Section 5.1.1 and Appendix B). The possibility of defining an ordered hierarchy of different phase coherence domains by the QFT notion of the "QV-foliation" can then obtain its proper diagrammatic universal formalization by the colimit operation using as the third category $\mathcal{B}$ of the comma category a particular small category of indices $j \in \mathcal{J}$ (see Appendix A), where in the QFT case, the set of indices corresponds to the set $\{\mathcal{N}\}$ of QFT coproducts, with each identifying a phase coherence domain of the QV, i.e., $\{j\} \equiv \{\mathcal{N}\}, \forall(\mathcal{N})$.

Indeed, the fundamental QFT construction of the "QV-foliation" as the fundamental dynamic tool used by nature for constructing "complex systems" in the physical reality and "memories" in the biological systems, and all the related QFT constructions discussed in this section, receive their proper "diagrammatic" *universal definition* in the CT metalanguage. Thus, we can obtain *the diagrammatic universal definition* of classes of QFT systems as initial (colimits) and terminal (limits) objects of indexed comma categories, which are, respectively, categories of "cocones" and "cones" of morphisms over a functor $F$ (see Appendix A).

Indeed, following the classic Adamek's-Rossicky's book [140] in the characterization of *different classes of small categories* and then *their limits-colimits* according to the different cardinalities, $\aleph_0, \omega_1, \omega_2, \ldots$, of their objects (see Appendix A), one can usefully apply these distinctions to the QFT construction of the category of the *q*-deformed Hopf coalgebras for the Bogoliubov functor **qHCoalg($\mathcal{B}$)** and, by the principle of the "algebra doubling", to the category of the doubled Hilbert spaces **DHilb($\mathcal{B}$),** as a subcategory of the category **Hilb.** In this way,

1.  We can formalize the general "particle-wave" duality principle in QFT of Equation (4) in Section 4.4, for which new objects (systems) "emerge" in nature as new phase coherence domains of the quantum fields at their ground state such as many SSBs of QV, using the categorical constructions of the initial/terminal objects in comma categories. Indeed, we can start from a given category $\mathcal{B}$ of quantum fields mapping its objects (quanta) and morphisms (fields) over a category $\mathcal{C}$ via the functor *F: B → C*. In turn, the SSB mechanism of the phase coherence among quantum fields can be formalized as a functor from the category **1** with one object ($*$) and its identity (reflexive) morphism $1_*$ as a functor $c$ to $C$ to obtain the structure of the comma category $1 \xrightarrow{c} \mathcal{C} \xleftarrow{F} \mathcal{B}$. In this way, we can define any new "emerging" object (system) in nature by the "universal morphism" (commuting diagram) from $c$ to $F$ as the *initial* objects (= colimits) in the comma category $(c \downarrow F)$ or dually as *terminal* objects (limits) in the comma category $(F \downarrow c)$ (see Definition A1 and Definition A2 with their explanations in Appendix A). Finally, it is worth emphasizing for the logical applications of this construction that set-theoretically this interpretation of objects as (self-containing) phase coherence domains of the quantum fields requires the *set self-membership* property characterizing the coalgebraic category of the NWF-sets (see Sections 2.5 and 5).

2.  Because of the category **qHCoalg($\mathcal{B}$)**, and the existence of only one generator $G$ of all the Bogoliubov transforms—strictly related to the NG-boson condensate value $\mathcal{N}$ indexing "dynamically" each phase coherence domain or degenerate state of QV $|0(\theta)\rangle_{\mathcal{N}}$ (see Equation (8))—we can categorically formalize the fundamental QFT construction of the "QV-foliation" by the colimit construction. Indeed, also in the QFT case, we can substitute the category $\mathcal{B}$ of Definition in Appendix A with the small category of indices $\mathcal{J}$, where the set of indices $\{j\} \in \mathcal{J}$ corresponds here to the set $\{\mathcal{N}\}$ of the NG-boson condensates univocally associated with the QFT non-commutative coproducts, i.e., $\{j\} \equiv \{\mathcal{N}\}$, $\forall \mathcal{N}$, with each $\mathcal{N}$ identifying a phase coherence domain (or "degenerate state") of the QV, i.e. $|0(\theta)\rangle_{\mathcal{N}}$. The "universal morphism" is therefore in terms of the initial objects of comma categories $(F \downarrow \Delta)$, which are *colimits* of "cocones of morphisms" according to Definition A3 in Appendix A.

3.  This depends on the fact that in QFT, there also exists a constant diagram: $\Delta_c : \mathcal{J} \to \mathcal{C}$, mapping every object in $\mathcal{J}$ to $c$ and every morphism in $\mathcal{J}$ to $1_c$. We can therefore define the *diagonal functor* $\Delta : \mathcal{C} \to \mathcal{C}^{\mathcal{J}}$ as the functor assigning to each QFT object $c$ of C the diagram (constant functor) $\Delta_c$ and to each morphism (Bogoliubov transform) $f : c \to c'$ in C the natural transformation $\Delta f = \Delta c \to \Delta c'$. Because $\Delta c, \Delta c'$ are constant functors, $\Delta f$ is just the morphism $f : c \to c'$ for every object in $\mathcal{J}$.

4.  Because the "diagonal functor" $\Delta : \mathcal{C} \to \mathcal{C}^{\mathcal{J}}$ is, in QFT, a categorical generalization of the *diagonal form* of the $n \times m$ Pauli's matrices of the dissipative QFT calculations on integer numbers (see Section 4.6 and Equation (9) with the relative comments), one could propose the following hypothesis requiring further studies and developments: namely, that the category **DHilb($\mathcal{B}$)** constitutes a *locally finitely presentable sub-category* of the category **Hilb**, with ordered objects (coproducts) characterized by infinitely many *countable $\aleph_0$-directed colimits* (see [140], pp. 2–3) because it satisfies the fundamental condition of having only one generator $G$ of all the Bogoliubov transforms (see [37], p. 18).

Indeed, recalling here again the remarks of Adamek and Rosicki ([140], pp. 70; pp. 101–105), the category of Hilbert spaces **Hilb** is not "locally presentable", and even less it is "locally *finitely* presentable". **Hilb** is closed, indeed, under a particular class of colimits (the $\omega_1$-*directed* colimits [25] in the category of Banach spaces **Ban**) [26]; however, therefore, it is not *cocomplete* (i.e., it does not satisfy the powerful "cocompleteness theorem", see Theorem A1 in Appendix A). Specifically, all the colimits are not in **Hilb** but in **Ban**, as the GNS-construction demonstrated (see Section 3.2 and Note 19 in it), so that **Hilb** is a full subcategory of **Ban**. A condition that the authors emphasize depends on the fact that the category **Hilb**, such as the strictly related category **Hopf** of Hopf bialgebras, differently from the category **Ban,** is made of objects that are *self-dual* [27].

In other terms, we know that in the operator algebra approach to QM and QFT, different families and then subcategories of the category **Hilb** exist. For instance, in the light of the GNS-construction, infinite subcategories exist, with each composed (by a family) of *finitely many unitarily equivalent* Hilbert spaces for the Gelfand functor, one for each *closed* quantum system, because it satisfies the Stone–Von Neumann theorem (see Section 3.1). In the light of our precedent discussion, however, the sub-category of the *doubled* Hilbert spaces *DHilb* for the Bogoliubov functor $\mathcal{B}$ also exists, i.e., **Hilb** $\hookrightarrow$ **DHilb($\mathcal{B}$).** It is composed of a *denumerable infinite number of inequivalent pairs* of *doubled* Hilbert spaces $\mathcal{H}, \widetilde{\mathcal{H}}$, with each pair being *dually unitarily equivalent* because it represents a *dissipative* quantum system in a balanced state.

Indeed, each pair, despite being defined on different non-commutative algebraic "footings", is *dually unitarily equivalent* for the contravariant application of the Bogoliubov transform, i.e., $\mathcal{H} \cong \widetilde{\mathcal{H}}$. Namely, despite each $\mathcal{H}$ of the pair at the level of its *inner structure* being, of course, self-dual and then belonging to the category **Hilb,** nevertheless, the two finite Hilbert spaces of each pair, generated by the contravariant application of the Bogoliubov transform, *share the same metrics* (effectively, they constitute one "$\sigma$-algebra" of the related probability space; see Section 5.1.1 and Appendix B) because they form only one dissipative system, and then they are *dually unitarily equivalent.* Therefore, the category **DHilb($\mathcal{B}$)** is constituted by infinitely many *inequivalent* (pairs of) finite Hilbert spaces, with *each dually equivalent pair* (i.e., satisfying the *coequalizer* condition for coproducts (see Definition A5 in Appendix A)) representing a different dissipative quantum system.

Moreover, because only one *generator* of all the Bogoliubov transforms characterizing the **DHilb($\mathcal{B}$)** indexed subcategory for the Bogoliubov functor exists in the dissipative QFT equations, it satisfies the fundamental condition for which this subcategory can be defined as a *locally finitely presentable category*, characterized by infinitely many *countable $\aleph_0$-directed colimits* (see [140], pp. 2–3). Therefore, the hypothesis, which, I repeat, needs to be properly mathematically developed and demonstrated, is that, differently from the category **Hilb,** its sub-category **DHilb($\mathcal{B}$)** is *cocomplete* (i.e., it satisfies the fundamental "cocompleteness theorem" (see Theorem A1 in Appendix A) because it is $\aleph_0$-*locally presentable*. Namely, despite it is constituted by infinitely many inequivalent pairs of doubled Hilbert spaces, whichever subset of them of any dimension is finitely denumerable, because a finite number of NG-bosons $|\mathcal{N}|$ is *always* sufficient for indexing each of these subsets (see Appendix A for further details).

The non-irrelevant consequence is that despite this subcategory being made of infinitely many inequivalent representations of CCRs and CARs modeled by the DDF principle for dissipative QFT systems, and then the pairs of *DHilb* spaces in the subcategory represent the outcomes of as many *phase transitions* of an overall non-linear dynamics in far from equilibrium conditions, nevertheless, the cocompleteness theorem grants the *compactness* of the underlying topological space. Namely, it contains all its limit points (colimits), with each representing a different phase coherent domain of the quantum field dynamics at their (balanced) ground state. Or, better, each represents a different class (cocone of morphisms) of (dissipative) quantum systems generated by nature through the universal mechanism of SSBs. For the very same reason, the trajectories in the phase space

between different phase coherent domains are (macroscopically) *chaotic trajectories* because of the compactness of the space in which they are defined (see [34]).

To sum up, the QFT conundrum of the infinitely many unitarily inequivalent representations (Hilbert spaces) of CCRs (and CARs) related to the Haag theorem has an at least partial (?) solution in the hypothesis of the thermal interpretation of QFT, for which any quantum system is a dissipative system, so that the DDF principle holds because of the Bogoliubov natural transformation.

Finally, from the *logical* standpoint, another character (see [34], p. 44) of this subcategory of Hilbert spaces is that their topologies satisfy the condition of being *Chu spaces* [141], in which NWF-sets (see [34], p. 44; [142], and Section 2.5) and then "rooted trees" of Kripke structures (models) can be defined [73] (see Section 5.2.3). This opens the way to the core of the "relational natural realism" and then the NR-formal ontology based on the principle that the semantics of the descriptive propositional formulas of the ontological language, formalized in a modal Boolean algebra, are *validated directly by homomorphism* onto the (co)algebraic mathematical structure of the physical systems they describe and/or to which they refer. Of course, this also gives a categorical positive answer to Quine's logical issue about the same possibility of a naturalistic formal ontology, with which we started this work (see Section 1.2).

## 5. From Coalgebras in Physics (QFT) to Coalgebras in Kripke's Relational Semantics

To formalize all these "descriptive" statements of the NR-formal ontology, it is necessary to define therefore which is the proper algebraic (Boolean) modal logic of this ontology in a categorical setting, with evident consequences for a formalized philosophy of nature and science. The main steps of this construction, corresponding to as many subsections of this section, are the following:

1.  An examination of the Boolean logic in the light of *Stone's momentous RTBA* and the consequent *Tarski's and Jónsson's construction of BAO's*, extending the operator algebra formalism from physics to logic. The core of both constructions is, indeed, the notion of *field of sets*, that is, the subalgebra of the power-set of a given set. Indeed, this subalgebra is a *σ-algebra*, that is, an algebra defined on a *measure space* and specifically on a *probability space* characterized by a finite number of degrees of freedom (the orthonormal basis of the associated Hilbert space, for instance) on which the statistical expectations are calculated. This opens the way for an extension of the operator algebra formalism over a topological complex algebra (algebra-subalgebras structures), from the statistical and quantum physics to the so-called "topological approach" to Boolean logic, properly to BAO logic.
2.  A formal explanation of the algebraic formalization of the relational notion of *meaning function* in CT logic, in which the semantics of the Boolean propositional logic is validated by its homomorphism with a complex (co)algebra over a topological space.
3.  Its extension to Goldblatt's and then to Kripke's *coalgebraic modal relational semantics* of BAOs in their categorical setting, which is the proper logic of the NR-formal ontology.

### 5.1. From Stone's Representational Theorem for Boolean Algebras to Tarski's Theory of the Boolean Algebras with Operators

### 5.1.1. Definition of "Field of Sets"

After Stone's RTBA [36], when we speak today in mathematical logic of an *algebra of sets,* we are speaking of a *Boolean algebra* (BA) and/or a *field of sets* [143]. Indeed, the more synthetic way for expressing the main statement of RTBA is: "RTBA states that every BA is *isomorphic* with a certain *field of sets*". Following [144], we can give the following definition of a "field of sets":

**Definition 1.** (*Definition of field of sets*). *A field of sets* $(X, \mathcal{F})$ *is a pair consisting of a set X and a collection (family) $\mathcal{F}$ of subsets F of X called "algebra over X", which contains the empty set as an*

*element and is closed under the set-theoretic operations of taking complements, finite union, and finite intersection, i.e., such that:*

- $F \in \mathcal{F} \Rightarrow X \backslash A \in \mathcal{F}$ (*closed under complementation*).
- $F, G \in \mathcal{F} \Rightarrow F \cup G \in \mathcal{F}$ (*closed under union*).
- $F, G \in \mathcal{F} \Rightarrow F \cap G \in \mathcal{F}$ (*closed under intersection*).

In other words, $\mathcal{F}$ constitutes a *subalgebra* of the power-set atomic BA of *X*. Elements of *X* are called *points* while the elements of $\mathcal{F}$ are called *complexes* (i.e., subalgebras of the complex structure algebra-subalgebras we are considering here) and/or the *admissible subsets* of *X*.

**Definition 1.** *(Definition of a σ-field of sets). A particular field of sets* $(X, \mathcal{F})$ *is called the σ-field of sets—and then the algebra $\mathcal{F}$ is called a σ-algebra—if one or both the two equivalent further conditions are given:*

- $\cup_{i=1}^{\infty} F_i := F_1 \cup F_2 \cup \ldots \in \mathcal{F} \mid \forall F_i \in \mathcal{F}$ (*closure under countable unions*).
- $\cap_{i=1}^{\infty} F_i : +F_1 \cap F_2 \cap \ldots \in \mathcal{F} \mid \forall F_i \in \mathcal{F}$ (*closure under countable intersections*).

*Because of its measurable character, a σ-field of sets is defined as "normal".*

Therefore, given that with "representation theorem" we intend to prove that every abstract structure with certain properties is isomorphic with another structure, the precedent definitions help us to understand the core of Stone's RTBA, even though its formal proof is outside the limits of this work (see for this [36]).

5.1.2. Stone Representation Theory of Boolean Algebras

To understand this step, it is necessary for philosophers unacquainted with this notion to recall what is an *ultrafilter* in set-theory defined by the power-set of a given set, and its relevance for set-theoretic logic, and Boolean logic. A power-set always includes the empty set because it is given by the combinations of all the subsets of a given set, so that if the generic *finite* set *X* has cardinality *n*, its power-set will have cardinality $2^n$ because of the inclusion of both the empty set as its minimal element and the same set *X* as its maximal element so that it satisfies a *reflexive* relation (beside *transitivity* and *anti-symmetry*) and any power-set is a partially ordered set or *poset*.

We already know from the previous discussion that *abstractly*, a BA is isomorphic with the power-set of a given set and more precisely with a field of set. Now, when we think of some *concrete* realization of a BA, such as, for instance, in any logical application of a BA, all the subsets of the power-set must satisfy the *further condition* of representing some *property*, say *p* expressed by a propositional formula *φ* of the logical calculus. This means that we must include some *filtering* condition over the power-set, first, the condition of *excluding the empty set*, given that every subset must contain *p* and this condition cannot be evidently satisfied by the empty set.

In other terms, this means that we need an *atomic* BA, where the minimal element is an *atom a*. Whereas, in *order theory,* an element *a* of a poset with the least element 0 (e.g., a power-set) is an *atom* if $0 < a$ and there is no *x*, such that $0 < x < a$. Therefore, in Boolean logic, every Boolean term corresponds to a propositional formula *φ* of propositional logic expressing a given property *p*. In this translation between Boolean algebra and propositional logic, Boolean variables *x, y*... become propositional variables (or *atoms*) *p, q, . . .* , Boolean terms such as $x \vee y$ become complex propositional formulas $p \vee q$, **0** becomes *false* or $\bot$, and **1** becomes *true* or $\top$. In this way, "an *atomic* BA is isomorphic to an *ultrafilter* of partially ordered sets, in our case a power-set" (see Figure 9).

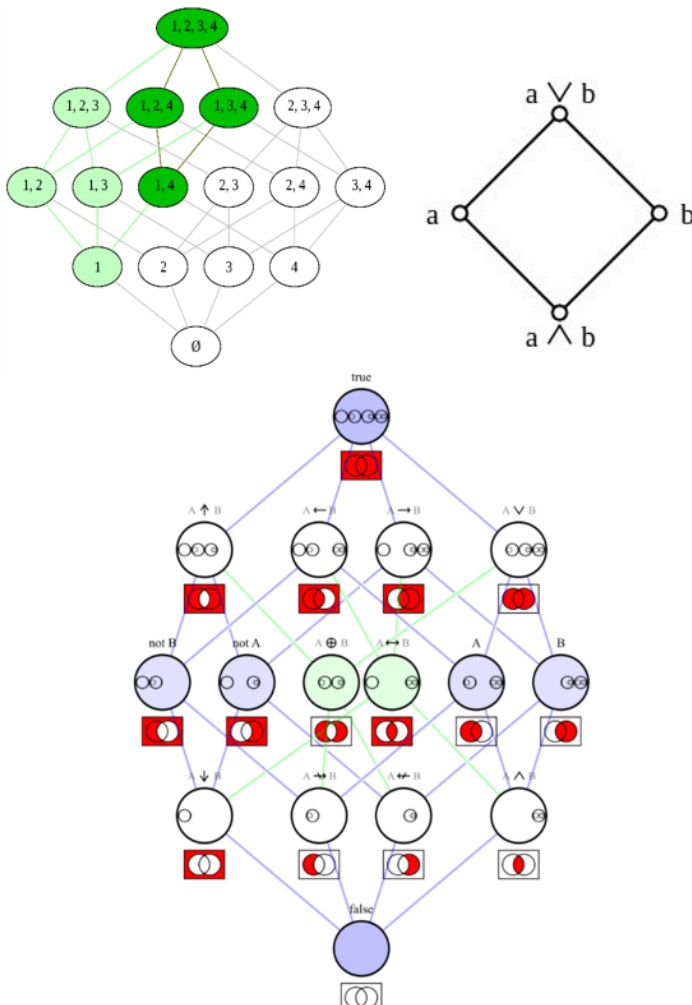

**Figure 9.** (**Top-left**): Representation of a *principal ultrafilter* (all the subsets colored in green), that is, the *maximal proper filter* of the power-set of the 4-element set $S$ $\{1, 2, 3, 4\}$ ordered by inclusion in a Hasse diagram. It has as the *principal element* or *upper set* ($\uparrow$) the *singleton* or 1-element set ($\uparrow \{1\}$). The subsets colored in dark green also constitute a *proper filter*, which is also a *principal filter*, because it has a principal element, i.e., ($\uparrow \{1, 4\}$). However, it is not an ultrafilter because only by adding the subsets coloured in light green, can we obtain the maximal proper filter of this power-set with the principal element ($\uparrow \{1\}$). (**Top-right**): Schematic representation of the *meet* (*last lower bound*) ($a \wedge b$) and the *join* (*least upper bound*) ($a \vee b$) of a Boolean logical lattice, where $a$ and $b$ are two propositional constants. (**Bottom**): *Complete* Hasse and Venn diagrams of all the propositional connectives representable on a Boolean logical lattice, having as *meet* (bottom-element $\bot$) the *always false* proposition ($0 := a \wedge \neg a$) and as *join* (top-element $\top$) the *always true* proposition ($1 := a \vee \neg a$).

To understand this statement, let us explain briefly what an ultrafilter is in mathematical *order* theory. An ultrafilter is the *maximal filter* of a given set, that is, the maximal partially ordered subset of the power-set of a given set, with the empty set excluded.

More generally, if $P$ is a set partially ordered by $\leq$ (and the power-set is always partially ordered), then:

- A subset $F \subseteq P$ is called a *proper filter* on $P$ if:
  - $F$ is nonempty (i.e., with the exclusion of the empty set $\varnothing$);
  - For every $x, y \in F$ $x \leq x$ and $x \leq y$ hold; and
  - For every $x \in F$ and $y \in P$, $x \leq y$ implies that $y$ is in $F$ too.
- A proper subset $U$ of $P$ is an *ultrafilter* on $P$ if:

○   *U* is a filter on *P;* and
○   There is no proper filter *F* that properly extends *U*.

Filters and ultrafilters are called *principal* if they contain a *least* element, i.e., they are of the form $F_a = \{x : a \leq x\}$ for some but not all elements *a* of the poset *P* (*a* is an atom). In this case, *a* is called the *principal element* of the ultrafilter. Therefore, all ultrafilters that are not principal are called *free.*

For ultrafilters defined on the power-set of a given set *S*, $\mathcal{P}(S)$, a principal ultrafilter consists, therefore, of all subsets of *S* containing a given element $p \in S$. This is fundamental in Boolean logic for understanding why the semantics of a formula of the propositional calculus interpreted equationally on a Boolean equation logic is defined onto the principal ultrafilter of the power-set of some set.

For instance, if we take (see Figure 9 (top-left)) the lattice *P* representing the power-set of the set *S* $\{1, 2, 3, 4\}$, ordered by inclusion in a Hasse diagram $(\mathcal{P}(S), \subseteq)$, the colored subsets represent the maximal filter starting from the *principal element p* (= *least element* or the "meet" of the corresponding Boolean lattice), which in our case is $\{1\}$. In this way, we can say that all the elements of the filter are *upward closed*, with respect to *p*: ↑*p*, i.e., they are *upsets* because they satisfy the condition: $(\uparrow p := \{\forall x \text{ in } F \mid x \geq p\})$, i.e., they are opsets (finite *intersections* or set products) closed by other opsets (finite intersections).

Moreover, if we observe the subsets of *S* colored in dark green in Figure 9 (top-left), we see immediately that these also constitute a *principal filter* with the principal element $(\uparrow \{1, 4\})$. However, it is not the *maximal proper filter* on *S* without the addition of the other light-green-colored subsets, which therefore is an *ultrafilter* with a principal element $(\uparrow \{1\})$.

Finally, in order theory, the *dual* of a filter *F* is called the *ideal I,* and the dual of an ultrafilter *U* is the *prime* or *maximal ideal.* Generally, we can obtain an ideal *I* from a filter *F* and a prime ideal from un ultrafilter *U* simply by *inverting the ordering relations* in *F* (*U*), i.e., $x \leq y$ with $x \geq y$, and then, in the corresponding Boolean logic, by substituting products (finite intersections) ∧ with disjunctions (finite unions) ∨. In this way, we are satisfying the duality (∧, ∨) of the fundamental *De Morgan logical laws* [28], i.e., we are satisfying in Boolean logic the so-called *De Morgan duality.* In the case of a principal ideal *I*, the principal element *p* is the *last element*, in our case $\{1, 2, 3, 4\}$. In this way, again dually, as to filters, we can say that all the elements of the ideal are *downward closed* with respect to *p*: ↓ *p*, i.e., they are *downsets* because they all satisfy the condition: $(\downarrow p : +\{\forall x \text{ in } I \mid x \leq p\})$, i.e., they are downsets (*disjoint unions* or set coproducts) closed by other downsets (unions).

However, to *properly* pass to posets that are BAs, the ultrafilters must be characterized by containing exactly its Boolean complement ¬*a* for each element *a* (atom) of BA to satisfy the BA signature of the proper operations defining BA, i.e., (∧,∨,¬) (see Figure 9 (top left and right)). Therefore, if *P* is BA and *F* is a proper filter, the following statements are equivalent ([145], p.186):

1.   (*F* ↑) is an ultrafilter on P.
2.   (*F* ↓) is a prime ideal on P.
3.   For each a ∈ P, either a ∈ P, or ¬ a ∈ P.

In the light of this discussion, it is possible to understand at least intuitively the notion of *Stone representation,* which is the core of his RTBA. Indeed, every *finite* BA can be represented as a power-set of its *atoms,* with each element of BA corresponding to the set of atoms below it, i.e., the join of which is the element. This power-set representation can be constructed for any *complete* atomic BA, where "complete BA" means BA in which every subset has a "supremum" or *least upper bound* [29].

In the light of this discussion and that in Section 5.1.1, Stone's RTBA stating the *isomorphism* and then the *Stone duality* between a BA *B* and a topological field of sets also becomes easily understandable. Specifically, this is a field of sets constituting the basis of the *Stone space* associated to BA *B*, i.e., *S(B)*. The *points* in *S(B)* are indeed the *ultrafilters* on

*B,* or equivalently the *homomorphisms* from *B* to the two-element BA. Conversely, given any topological space $(X, \mathcal{T})$, its subsets that are clopen form BA.

Therefore, on the basis of this and of Appendix B, we can understand why a simpler version of Stone's RTBA states that every BA *B* is isomorphic to the algebra of clopen subsets of its Stone space $S(B)$. The isomorphism indeed sends an element $b \in B$ to the set of all ultrafilters that contain *b,* as the previous illustration of the notion of the topological field of sets showed.

In the framework of CT, the Stone RTBA states that a *dual equivalence* for the contravariant application of the Stone functor $\mathcal{S}$ between the category of BAs **Bool** and the category of Stone spaces **Stone** exists, i.e., $\textbf{Bool}(\mathcal{S}) \simeq \textbf{Stone}(\mathcal{S})^{\textbf{op}}$. This depends on the fact that in addition to the (invertible) homomorphisms between BA *B* and its Stone space $S(B)$ making them isomorphic, i.e., $B \cong S(B)$ (= "Stone duality"), each homomorphism (monotone function) from a BA *A* and a BA *B* in the category **Bool** corresponds naturally (by a natural transformation) to an homomorphism (continuous function) going in the opposite direction from the Stone space $S(B)$ to the Stone space $S(A)$. This theorem is thus a special case of the more general *Stone duality* between topological spaces and partially ordered sets.

For our ontological aims, what is relevant is indeed the extension of Stone RTBA to *measure spaces* that are the topological spaces used in physics. As we have just said, Stone RTBA by the construction of the fields of sets and the ultrafilters over them holds for arbitrary set unions and intersections, i.e., it does not require *complete* atomic Bas, and this gives the theorem its full and powerful generality. However, if an algebra over a set is closed over countable unions and intersections, it is a *σ-algebra* (see Section 5.1.1) and the corresponding field of sets is a *measurable space,* and its complexes are *measurable sets* [30]. Now, a theorem demonstrated independently by L. H. Loomis [146] and R. Sikorski [143], the *Loomis–Sikorski theorem,* grants the Stone-duality between *σ-complete* BAs, $\mathbb{BA}^{\sigma}$ (or "abstract *σ*-algebras") and *measurable spaces*.

We recall, indeed, that in applied mathematical sciences, and physics, a *measure space* $(X, \mathcal{F}, \mu)$ is a measurable space and $\mu$ is a measure defined on it. If $\mu$ is a *probability measure,* we have a *probability space,* and its underlying measurable space is a *sample space.* The points of this space are *samples* representing the outcomes of measuring operations while the measurable sets (complexes) are called *events* and represent the outcomes of physical processes to which we want to assign probabilities. In physics and specifically in quantum physics, we work on measure and probability spaces derived from significant algebraic structures such as the inner products of Hilbert spaces and the topological groups with the associated topological spaces and complexes. This is confirmed by the fact that the topological spaces of Stone's RTBA and Hilbert space complexes (Hilbert spaces with their C*-subalgebras) are the same (see [45], pp. 805–833, and Appendix B).

5.1.3. Jónsson–Tarski Theory of the "Boolean Algebras with Operators"

In this framework, the Jónsson–Tarski representation theorems for the *Boolean algebras with operators* (BAO), effectively extending Stone's RTBA to the operator algebra approach [4,5], acquire a fundamental relevance. To understand the theoretical core of these theorems, we must introduce the notion of representation of *interior algebras* by field of sets *preorders* [31]. Indeed, a *preorder field* is a triple $(X, \leq, \mathcal{F})$, where $(X, \leq)$ is a preordered set and $(X, \mathcal{F})$ is a field of sets.

The preorder fields play an important role in the representation theory of interior algebras, given that an *algebraic* field of sets can be defined only on a (topological) *complex algebra* (algebra-subalgebras structure) or *algebra of complexes* [32] $\mathbb{A}^{+}$, so that $x \leq y$ if $S \in \mathbb{A}^{+}$, $(y \in \mathbb{A}^{+} \Rightarrow x \in \mathbb{A}^{+})$ for every complex. To pass to BAOs, we must consider structures $(X, (R_i)_I, \mathcal{F})$, where $(X, (R_i)_I)$ is a *relational structure* $\mathbb{S}$, i.e., a set with an indexed family of relations defined on it, and $(X, \mathcal{F})$ is a field of sets. Therefore, the *complex algebra* $\mathbb{A}^{+}$ determined by the field of sets $\textbf{X} = (X, (R_i)_I, \mathcal{F})$ on a relational structure is BAO:

$$\mathbb{BA}(\textbf{X}) := (\mathcal{F}, X, \cap, \cup, \top, \bot, (f_i)_I)$$

where for all $i \in I$, if $R_i$ is a relation of arity $n + 1$, then $f_i$ is an operator of arity $n$ and $\forall(S_1, \ldots, S_n \in \mathcal{F}) \, f_i(S_1, \ldots, S_n) \, \{x \in X : \exists(x_1 \in S_1, \ldots, x_n \in S_n), R_i(x_1, \ldots, x_n, x)\}$.

Such a construction can be generalized to fields of sets on arbitrary algebraic structures, where both operators and relations as operators can be viewed as a special case of relations. If $\mathcal{F}$ is the whole power set of $X$, i.e., $2^X$, then $\mathbb{BA}(\mathbf{X})$ is called a *full complex algebra* or *power set algebra* $\mathbb{P}(S)$. Therefore, Jónsson and Tarski [4,5] demonstrated that every BAO can be represented as a field of sets on a *relational structure* $\mathbb{S}$ in the sense that it is *isomorphic* to the complex algebra corresponding to the field (see [147,148] for such a reconstruction). Particularly, in such a way, Jónsson and Tarski extended Stone's RTBA to a particular type of Boolean algebras, the "$\sigma$-complete Boolean algebras" $\mathbb{BA}^\sigma$, and then, in the light of the Loomis–Sikorski theorem introduced in Section 0, they extended the RTBA to the *measurable* topological spaces [148]. This formally exemplifies the extension in CT logic of the (topological) "operator algebra" formalism from physics to logic and vice versa via the category of BAOs, **BAO.**

Finally, an important extension of this representation theory for the category **BAO** that is fundamental in its application to *Kripke relational semantics* (see Section 5.2) consists in the extension of this representation theory at the level of *morphisms*. Specifically, an *algebraic homomorphism* between two Boolean algebras $\mathbb{BA}_1 \to \mathbb{BA}_2$ *dually* induces a certain type of structure-preserving *reversed* mapping between the correspondent topological structures $\mathbf{X}_{\mathbb{BA}_2} \to \mathbf{X}_{\mathbb{BA}_1}$ called *bounded morphism*. This means the extension of the categorical dual equivalence between Stone spaces and BAs via the contravariant application of the Stone functor $\mathcal{S}$, $\mathbf{Stone}(\mathcal{S}) \simeq \mathbf{Bool}(\mathcal{S})^{\mathbf{op}}$, to the categorical duality between BAOs and the relational structures on Stone spaces, i.e., $\mathbf{Stone}^+(\Omega) \simeq \mathbf{BAO}(\Omega)^*$. In both cases, indeed, these dual equivalences depend on the respective dual functors, generally rewritten as $(\cdot)^* = \Omega = (\cdot)$ [147].

### 5.1.4. The Meaning Function in Relational Semantics

As discussed, in their seminal papers on BAOs, Jónsson and Tarski introduced the notion of *relational semantics* and then the notion of an *algebraic interpretation* of the set-theoretic notion of *meaning function* that we discuss here in its categorical formalization for the category **BAO**, essentially following Yde Venema's exposition in ([6], pp. 331–426).

The meaning function is the arrow-theoretic version of the set-theoretic semantics of the propositional calculus, for which a propositional function (e.g., $(p \wedge q)$) is evaluated as true/false on the operations over correspondent sets, *P, Q*, that is, on the propositional function *extension*. So, in our example, $(p \wedge q)$ is *true* if and only if the intersection of the correspondent sets holds, i.e., $(P \cap Q)$.

Correspondingly, the *meaning function* $[\cdot]$ maps a propositional formula $\phi$ to its extension $[\varphi]$; *that makes $\phi$ true.* One can then impose an *algebraic structure* $\mathbb{S}$ on the formulas of the propositional calculus, i.e., the so-called $\tau$-*formula algebra* $\mathbb{F} \, \phi_\tau$, where $\{\tau\}$ is the set of propositional connectives or propositional predicates («and», «or», «not», etc.), by which substitutions are *completely determined by their values on the variables*. Namely, for any function $\sigma$ assigning a formula to a variable, the substitution by $\sigma$ is the unique extension $\widetilde{\sigma}$ of $\sigma$ to an *endomorphism* on $\mathbb{F} \, \phi_\tau$. Therefore, given an arbitrary algebra $\mathbb{A}$ of type $Bool_\tau$, any assignment, mapping variables to elements of the carrier-set of $\mathbb{A}$, has a unique extension $\widetilde{\sigma}$, which is a *unique homomorphism* from $\mathbb{F} \, \phi_\tau$ to $\mathbb{A}$: $\mathbb{F} \, \phi_\tau \to \mathbb{A}$.

For instance, in the well-known case that A is the «two-valued Boolean algebra», 2BA, its carrier is given as the set $2 = \{0,1\}$ while the classical *truth tables* give the interpretation (semantics) of the Boolean symbols/functions. Namely, given a *valuation* $V: X \to 2$ of truth values to propositional variables, we can simply arithmetically *compute* the truth value $\widetilde{V}(\varphi)$ of any complex propositional formula $\phi \, (p_1, \ldots, p_n)$, using the unique homomorphism $\widetilde{V} : \mathbb{F}\phi_\tau \to 2\mathbb{BA}$ *extending* the assignment *V*. This formalizes the usual statement of Boolean logic, according to which we can "extend" the valuation functions of propositional calculus to the homomorphic binary "arithmetical" operations of a Boolean algebra.

To generalize the precedent example to whichever relational semantics of propositional logic [6], pp. 337–342, to complete the transition from a set-theoretic to an arrow-theoretic relational semantics, we refer to the already introduced notion of *complex algebras* $\mathbb{A}^+$. We can therefore redefine this notion, descriptively introduced in the precedent subsection, in the CT logic formalism [6], p. 339:

**Definition 1.** *(Complex algebra of a $\tau$-frame). Given an n + 1-ary relation R on a set S, define the n-ary map $\langle R \rangle$ on the power set of S by:*

$$\langle R \rangle (a_1, \ldots, a_n) := \{s \in S | Rs_1 \ldots s_n \text{ for some } s_1 \ldots s_n \text{ with } s_1 \in a_1, \forall i\}$$

*The complex algebra $\mathbb{S}^+$ of a $\tau$-frame $\mathbb{S}$ is obtained by expanding the power set algebra $\mathbb{P}(S)$ with operations $\langle R_{\nabla} \rangle$ for each connective $\nabla$. Specifically:*

$$\mathbb{S}^+ := \langle \mathcal{P}(S), S, \varnothing, \approx_S \cap, \cup \{\langle R_{\nabla} \rangle | \nabla \in \tau\} \rangle. \tag{10}$$

*where $\approx_S$. denotes all the equivalences in S. All this means that, from the perspective of complex algebra, a valuation is nothing but an assignment of variables of $\mathbb{S}^+$. Much more significantly, all this means that given a valuation V on a frame $\mathbb{S}$ , it is possible to prove by induction that:*

$$\mathbb{S}, V, s \Vdash \varphi \Leftrightarrow s \in \widetilde{V}(\varphi) \tag{11}$$

*where $\widetilde{V} : \mathbb{F}ma_{\tau} \to \mathbb{S}^+$ is the unique homomorphism extending V.*

Consequently, as Venema emphasizes, "what equivalence (11) reveals is that, in a slogan, *meaning is homomorphism*" ([6] p. 338). In other words:

**Definition 1.** *(Meaning function). Let V be some valuation on a $\tau$-frame $\mathbb{S}$. Then, the meaning function is the unique homomorphism $\widetilde{V} : \mathbb{F}ma_{\tau} \to \mathbb{S}^+$ that extends V to $\widetilde{V}$.*

From this, the following theorem holds:

**Theorem 1.** *Because of* Definition 13., *let $\phi^{\approx}$ denote the equation $\phi \approx \top$, where $\approx$ is the algebraic equality symbol and $\top$ stays for "true" (e.g., "1" in a two-valued Boolean logic). Then, we have that for any $\tau$-frame $\mathbb{S}$, and any $\tau$-formula $\phi, \psi, \ldots$, the following validations hold:*

$$\mathbb{S} \Vdash \varphi \Leftrightarrow \mathbb{S}^+ \vDash \varphi^{\approx}; \text{ and } \mathbb{S} \Vdash \varphi \leftrightarrow \psi \Leftrightarrow \mathbb{S}^+ \vDash \varphi \approx \psi. \tag{12}$$

*Specifically, the validity of a formula in the frame $\mathbb{S}$ corresponds to that of an equation in the complex algebra $\mathbb{S}^+$ of $\mathbb{S}$ , and vice versa.*

In this case, the metalogical biconditionals $\Leftrightarrow$ in (12) express the core of the notion of *algebraization of a propositional logic*, that is, of its translation into an *equational logic* by which substitutions are *completely determined by their numerical values on variables*.

Finally, it is worth emphasizing that the *validation direction* of the homomorphism constituting the algebraic construction of the meaning function is from the formula structure $\mathbb{F}$ to its extension $\mathbb{F}'$, i.e., $(\mathbb{F} \vDash \varphi) \to (\mathbb{F}' \vDash \varphi)$. However, and this is very interesting for our aims, when *a selection criterion of admissible sets* is present, e.g., an *ultrafilter Uf*, the validation direction is obviously *reversed*, i.e., $(\mathbb{F} \vDash \varphi) \leftarrow \left(Uf(\mathbb{F}') \vDash \varphi\right)$. This is the case of the partially ordered sets of Stone's RTBA. However, the usage of ultrafilters in standard set theory is limited by their *non-finitary* character because of the "ultrafilter lemma" in **ZF(C)** (see [149], pp. 57–68), as we discuss (Section 5.2.1) [33], which indeed implies *second-order* semantics. This limitation can be "locally" avoided if we define the (coalgebraic) topologies of the Stone theorem over NWF-sets such as in Kripke's relational semantics, as we see immediately.

*5.2. The Coalgebraic Relational Semantics of Kripke Models in Modal Logic*

5.2.1. From Goldblatt's to Kripke's Development of a Modal Relational Semantics

To properly understand Kripke's modal relational semantics, we must previously understand the extension of RTBA categorical duality to the category of *modal Boolean algebras* because of the fundamental "Goldblatt–Thomason theorem". It is not possible nor necessary to illustrate here the formal details of this demonstration for which we refer to ([6], pp. 354–356). For us, it is sufficient to recall that this theorem demonstrated the dual equivalence between the category of the *descriptive general frames DGFs*, which are the frames of Definition 12. for a set $\{\tau\}$ of *modal* propositional connectives, and the category of *modal* BAOs, i.e., $\mathbf{DGF}_\tau\ (\mathbf{\Omega}) \simeq \mathbf{BAO}_\tau\ (\mathbf{\Omega^*})$ In this way, we are effectively extending to the modal logic $\mathrm{ML}_\tau$ the Stone categorical dual equivalence $\mathbf{Stone(\Omega)} \simeq \mathbf{Bool(\Omega^*)}$ for the Stone functor $\mathcal{S}^* := (\cdot)_* \circ \Omega \circ (\cdot)^*$, rewritten as the dual equivalence between the category of *descriptive fields of sets*, which are the equivalent in set-theory of the Stone spaces in topology, and the category of Boolean algebras [147,148].

The consequent construction of the meaning function can be extended from a *Goldblatt general frame* $\mathfrak{G}$ to a *Kripke frame* $\mathfrak{F} = W, R$, where $W$ is a set of "possible worlds" and $R$ is an "accessibility relation" between pairs of them (see Section 5.2.3), and its extension F'= $\langle W', R'\rangle$. This, indeed, is effectively a *general frame* $\mathfrak{G} = \langle \mathfrak{F}, \mathbb{W}\rangle$ extending F with a $\tau$-algebra of the $W$ power-set $\mathcal{P}(W)$, closed under Boolean operations and modal operators $\{R_\alpha\}_{\alpha \in \tau}$. W thus plays the same role as F of a complex algebra S$^+$ as S in Definition 12., so we can denote it with G$^+$. Therefore, also in the case of Kripke frames F, the truth preservation direction is normally from a frame F to its extension F' = G$^+$, that is, $(\mathrm{F} \models \phi) \to (\mathrm{F'} \models \phi)$.

To pass the Goldblatt–Thomasson theorem, we must consider the ultrafilter extensions of $\mathfrak{G}$, $Uf(\mathfrak{G})$, denoted as *descriptive general frames*, with the fundamental difference as to the simple general frames $\mathfrak{G}$, for which, in this case, just as in the Stone duality of the Stone RTBA, the truth preservation direction is reversed, that is, it goes from $\mathfrak{F}'$ to $\mathfrak{F}$, that is, $(Uf(\mathfrak{F}) \models \varphi) \to (\mathfrak{F} \models \varphi)$. This reversal of the validation direction depends on the fact that ultrafilters act, here as everywhere in logic and mathematics, as a *second-order selection criterion of admissible sets on the power-set of a given set*. This depends on the so-called "ultrafilter lemma" for which any proper filter is the intersection of *all* ultrafilters containing it, requiring that free ultrafilters for existence must be defined on *infinite* sets (see [149], pp. 57–68). Therefore, the ultrafilters lemma supposes in **ZF** the "axiom of choice" (hence **ZFC**), or the "Zorn's lemma", as was the case of Stone's demonstration of RTBA.

What is typical of Kripke's modal relational semantics is that it also allows a *first-order selection criterion of admissible sets* related to the possibility of defining the truth valuation function $V(p)$ of Kripke models/structures on some *restriction W'* over the whole set of possible worlds $\{W\}$, i.e., $W' = \restriction \{W\} := (W' \subseteq W)$, that is, exclusively for all the worlds accessible by a given world.

If *ontologically* these restrictions over the possible states of the world give us back *the logic of the physical causality principle* (light-cone) (see Section 1.2 and Figure 1) this formally immediately leads us to Kripke structures defined onto a coalgebra (disjoint sums) of NWF-sets, for which only set *partial orderings* are allowed, defined on Stone spaces. This is, indeed, the original intuition underlying the seminal work of S. Abramsky in 1988 [150], who proposed for the first time the possibility of using the so-called *Vietoris construction* on a coalgebra of NWF-sets over Stone spaces. Indeed, this allows us to use Aczel's powerful construction of the *final coalgebra theorem*, justifying the duality between a final coalgebra and an initial Boolean algebra, using the *Vietoris functor* V/V* as *dual functors*. The core of this extension is the *isomorphism* between the category of descriptive general frames **DGF** of Goldblatt's theorem and the category of coalgebras on Stone spaces or *Stone coalgebras, SCoalg*, using the Vietoris functor V, **SCoalg(V)** as its endofunctor, that is, $\mathbf{DGF}_\tau \cong \mathbf{SCoalg(V)}$. In this way, from Goldblatt's dual equivalence $\mathbf{GDF}_\tau\ (\mathbf{\Omega}) \simeq \mathbf{BAO}_\tau\ (\mathbf{\Omega^*})$, we can derive immediately the other one: $\mathbf{SCoalg(\Omega)} \simeq \mathbf{MBA(\Omega^*)}$ for the Vietoris

functor: $\mathcal{V}^* := (\cdot)_* \circ \Omega \circ (\cdot)^*$, between the categories of the Stone coalgebras and the modal Boolean algebras [6]. We return in Section 5.2.3 to the logical and ontological relevance of such a construction.

5.2.2. Bisimulation and Co-Induction in Coalgebraic Logic

As a further step, let us introduce the notion of *bisimulation,* as an arrow-theoretic counterpart of the notion of algebraic *congruence* effectively defined on coalgebras. This notion was defined for the first time by Aczel with respect to NWF-sets in [82], even though in TCS, this notion today has a wider application for defining the *behavioral equivalence* between computational systems (automata) in any algebraic fashion (see [6], p. 388):

**Definition 1.** *(Bisimulation (symbol: $\leftrightarrows$)). Let $\mathbb{S} = \langle S, \sigma \rangle$ and $\mathbb{S} = \langle S', \sigma' \rangle$ be two systems for the set functor $\Omega$. A relation $B \subseteq S \times S'$ is called bisimulation between $\mathbb{S}$ and $\mathbb{S}'$ if we can endow it with a coalgebra map $\beta = B \to \Omega B$ in such a way that the two projections $\pi: B \to S$ and $\pi': B' \to S'$ are homomorphisms from $\langle B, \beta \rangle$ to $\mathbb{S}$ and $\mathbb{S}'$, respectively:*

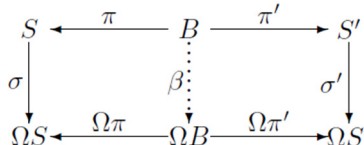

*If a bisimulation B with states (s, s') ä B exists, we say that s and s' are bisimilar, i.e., sÆ s'.*

Now, let us suppose two deterministic systems formally defined as pointed $2 \times \mathcal{I}^{\mathcal{C}}$-coalgebras, $\mathbb{S}0$, $\mathbb{S}1$, where $\mathcal{I}$ is the set of the identity relations $Id_s$ univocally indexing each $s$ state of the system. In case the functor $\Omega$ admits a final coalgebra $\mathcal{Z}$, we can formalize the notion of *observational equivalence*, with an evident relevance for quantum physics, by expressing the equivalence between the state $s_0$ in coalgebra $S_0$ and the state $s_1$ in coalgebra $S_1$, as $!_{\mathbb{S}_0}(s_0) =!_{\mathbb{S}_1}(s_1)$. More generally (see [6], p.389):

**Definition 1.** *(Definition of observational equivalence). Let $\mathbb{S} = \langle \mathbb{S}, \sigma \rangle$ and $\mathbb{S}' = \langle \mathbb{S}', \sigma' \rangle$ be two systems for the set functor $\Omega$. The states $s \in S$ and $s' \in S'$ are observationally equivalent, i.e., S, s $\equiv_\Omega S'$, s', if an $\Omega$-system $\mathbb{X}\langle X, \xi \rangle$ exists and homomorphisms f: $S \to X$ and f': $S' \to X'$, such that f (s) = f' (s'). In case $\Omega$ admits a final coalgebra Z, then S, s $\equiv_\Omega S'$, s' iff $!_S$ (s) = $!_{S'}$ (s').*

This is precisely the case of the NWF-sets [11] for which the powerful "final coalgebra theorem" holds [82] (see Section 2.5).

However, this is also the case of the physical category of QFT coalgebras related to the category of the non-commutative coproducts of Hopf algebras for the Bogoliubov functor **qHCoalg(**$\mathcal{B}$**)**, which also satisfy the final coalgebra theorem, and a categorical indexing by a diagonal functor $\mathcal{C}^{\mathcal{J}}$, as we explained at length in Section 4.5 and Appendix A. This conversely means that NWF-sets are those on which the mathematical formalism of dissipative QFT can be naturally defined! Not casually, indeed, in NWF-sets, the set self-membership and then infinite chains of set inclusions are allowed, just as in dissipative QFT systems, the phase coherences of quantum fields are related to the infinitely many SSBs of QV (see also the discussion of the cocomplete category of **DHilb(**$\mathcal{B}$**)** and its coalgebraic internal structure in Section 4.8).

Afterward, Lawrence C. Paulson demonstrated, in the style of Peter Aczel for NWF-sets, a final coalgebra theorem for **ZF** set theory, even though it was significantly limited to *denumerable sets* [151], showing the generality for both standard and non-standard set theories of the categorical duality between initial algebras and final coalgebras, and their usefulness in TCS [152], in quantum physics and then in quantum computing [34].

The theoretical relevance of these CT notions for a *finitary* constructive mathematics and for functional programming in TCS but also for a *dynamic* approach to the evolutionary

cosmology in fundamental physics is that, in addition to the (algebraic) *inductive* and *recursive* methods for the set definition and proof, we now have the (coalgebraic) *co-inductive* and *co-recursive* methods for the set definition and proof. That is, for the "set unfolding" from a common root in NWF-set trees of *different posets* along different and reciprocally irreducible branching (edges of the set tree) from the common root (see Section 5.2.3). Intuitively, "the crucial feature here is that processes need not be *bottom-up,* inductive, but it can instead be *top-down* co-inductive streams of events". More precisely (see [152,153], p. 46)—and waiting for a more explicit formalization of this duality between induction/coinduction using Kripke's relational logic, given in Section 5.2.3—we have:

**Definition 1.** *(Sets inductively/co-inductively defined by F). For a complete lattice L whose points are sets, and for an endofunction F, we have the sets:*

$$F_{ind} := \cap\{x|F(x) \leq x\}$$
$$F_{coind} := \cup\{x|x \leq F(x)\}$$

Specifically, the meet of the pre-fixed points and the join of the post-fixed points, i.e., the least and the greatest fixed-points if *F* is monotone, are, respectively, the sets *inductively* defined by *F* and the sets *co-inductively* defined by *F*.

Therefore, the following rules hold:

**Definition 1.** *(Induction and co-induction as proof principles). In the hypothesis of Definition 16., we have:*

$$if \ F(x) \leq x \ then \ F_{ind} \leq x \ (\text{induction as a method of proof})$$
$$if \ x \leq F(x) \ then \ x \leq F_{coind} \ (\text{co} - \text{induction as a method of proof})$$

In the light of the two precedent definitions, it is easy to grasp the main idea underlying the usage in TCS of coalgebra structures of NWF-sets to model *concurrent computations* [34], which was evident since the preface of Aczel's book (see [11], p. xiv), and as it was finally synthesized in J. M. Rutten's construction of the *Universal Coalgebra* as a "general theory of systems" (see [81], especially pp. 69–70). In this framework, the semantics of BAOs, either in functional programming, or in propositional logic, is given directly *by the physical states* of the system interpreted as a "labeled state-transition system" (LTS) and *coalgebraically* modeled [81]. This means conceiving the coalgebraic co-recursive computations (co-inductively defined: ↓), and the algebraic recursive computations (inductively defined: ↑) as two *concurrent computations* (see Note 34), respectively, for a final coalgebra (defined on a Stone space) and for an initial Boolean algebra to give a logical/computational counterpart in TCS of the categorical duality **SCoalg(Ω) ≃ MBAO(Ω\*)**. When the two computations "match" with each other, we have a *Boolean algebra with operators* (BAO), that is, a Boolean algebra whose signature ($\neg$, $\wedge$, $\vee$, $\top$, $\bot$) is constituted by *operators* acting on an algebra of complexes [4,5] and whose "top" and "bottom" ($\top$/$\bot$) operators indicate that computations are coinductively and inductively lower and upper bounded on a *finitary* basis.

This depends on the powerful notion of the *functorial bounded morphism* between Kripke models $\mathfrak{M} \overset{\rightrightarrows}{\leftarrow} \mathfrak{M}'$, as we discuss in Section 5.2.3 (see also the related powerful construction of the "infinite-state black-box machine" $\mathbb{M}$ in TCS and its application to the problem of "data streaming" in AI *machine learning* in [81,154]).

To provide an intuitive illustration of the notion of the concurrent coalgebraic/algebraic computations, see Figure 10 in which the coalgebra (up) "functorially mirrors" its structure onto the algebraic structure (down) to give the latter a *selection criterion* of admissible sets, in a word, its *finitary semantics,* as we discuss.

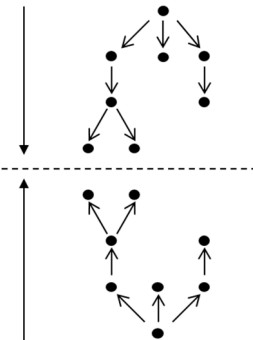

**Figure 10.** Intuitive representation of the principle of the *coalgebraic/coinductive* (final: upper part of figure) *algebraic/inductive* (initial: lower part of figure) concurrent computations, where the coalgebra functorially *mirrors* its structure over the algebra, and for which, in both directions, "the *finitary* is the limit of the successions of finites" (Abramsky).

However, we have already discussed the relevance of the coalgebraic construction of coinduction not only in TCS and logic but primarily in mathematical analysis and mathematical physics because of the strict categorical connection between the notion of "direct limits" in mathematical analysis and colimits in CT (see [71,72], and Section 4.8 with Appendix A).

5.2.3. The Relevance of Kripke's Modal Relational Semantics for an Ontology of Quantum Physics

Referring to [6,43] for a more complete formal discussion about Kripke's modal relational semantics, let us sketch its main elements and its relevance for a formal ontology and then for a *formal philosophy of nature* based on QFT as the fundamental physics.

As a starting point, let us recall some fundamental axioms of ML in Lewis' axiomatization that we introduced in Section 1.1 while for a complete treatment of this axiomatic approach to ML, we refer to classic handbooks such as [14]. On this regard, we recall here the so-called *normal modal system* **K**, based on the *necessitation axiom* **N**:

$$((X \vdash \alpha) \to (\Box X \vdash \Box \alpha))$$

where $X$ is a set of propositions, $\alpha$ is a propositional meta-symbol, and $\Box$ is the necessity operator of ML. This axiom is sufficient for validating in modal propositional logic, ML, all the logical laws (tautologies) in their modal form, starting from the modal *modus ponens* for propositional formulas:

$$\Box(\vdash \varphi \wedge \vdash (\varphi \to \psi) \to (\vdash \psi))$$

Now, this looks like a standard second-order semantics of first-order formulas, where $\vdash \varphi$ means "$\varphi$ is provable" and $\Box$ behaves like $\forall$.

Therefore, on the basis of Definition 13. and Theorem 1. about the "algebraic" meaning function for BAOs and its extension to modal BAOs through the Goldblatt–Thomasson theorem (Section 5.2.1), we can state this other fundamental theorem, necessary for defining the *minimal* modal algebra for the normal modal system **K** (for the theorem complete statement and relative proof (see [6], pp. 340):

**Theorem 2.** *(Minimal modal algebra for the normal modal system* **K***). Let* $\Gamma$ *be a set of* $\tau$*-modal formulas. Then, the class of atomic Boolean algebras with operators* $BAO_\tau$ *(*$\Gamma$*), which validates the set of equations* $\Gamma^\approx := \{\gamma \approx \top \,|\, \gamma \in \Gamma\}$*, algebraizes* **K**$_\tau$ $\Gamma$*. In particular, the class of modal algebras MA(*$\Gamma$*) algebraizes* **K** $\Gamma$*.*

Let us apply the previous definitions and theorems to Kripke's relational semantics by reducing the polyadic frames $\mathbb{S}$ to the dyadic *Kripke frames* $\mathfrak{F} = \langle W, R \rangle$, constituted by "a universe" (and/or "a world") $W$, which is a set of "possible worlds" (and/or of "possible states of a world") $\{w_1, w_2, \ldots, w_n\}$, and by $R$ as a dyadic "accessibility" relation between a

pair of *w* defined on the Cartesian product of all possible worlds: $W \times W$, i.e., $(R \subseteq W \times W)$ (see Figure 11).

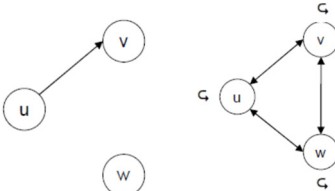

**Figure 11.** Representation of the Kripke $\mathfrak{F}$ with $W = \{u, v, w\}$ and with (**Left**) $(R \subseteq u, v)$, where *v* is accessed by *u* and not vice versa while *w* is irrelated. (**Right**) with $(R \subseteq W \times W)$ to define an equivalence class of possible worlds, or *universe* because all satisfy the reflexive, symmetric, and transitive accessibility relations among them. In parentheses, the right graph is a representation of a **S5 (KT45)** modal system, as we see.

Therefore, if we take as the basic semantic entity of relational semantics a *Kripke model* $\mathfrak{W} = \left( W, \{R_\nabla\}_{\nabla \in \{\tau\}}, V(p) \right)$, where $\nabla$ is a propositional connective of the connective set $\{\tau\}$ of the propositional calculus, and $V(p)$ is an evaluation function of a proposition *p* defined on *W*. The model $\mathfrak{M}$ is effectively a *relational structure*, denoted as a *Kripke structure*. We have, indeed, a domain of quantification *W*, a collection of dyadic "accessibility" relations *R* over this domain, and a collection of unary (1/0) valuation relations $V(p)$ for each propositional symbol $p \in \mathrm{PROP}$. This means that properly, it is not necessary to speak about Kripke models using modal languages: "they provide us with everything needed to interpret classical languages too" [42], p. 10. In other terms, for our aims, we can also apply Kripke's model theory to the *mathematical languages of physics* and not only to the modal languages of philosophy (see [148]).

Moreover, in philosophical logic and formal philosophy, Kripke's relational semantics was developed for formalizing modal logics (ML) because of the "stipulatory character" of Kripke's "possible worlds" notion and their accessibility relations. Namely, in this sense, they are "possible", as far as satisfying the logical rules of a given (modal) language. This implies that the notion of "possible worlds" are applicable to whichever intensional logic interpretation (either alethic, or epistemic, or deontic (see Section 1.1) of the MC), and not only to the ontological (alethic) one, differently, for instance, from Leibniz' "possible world" semantics, whose meaning is essentially cosmological and then "alethic". Of course, in our ontological and then alethic application, the dyadic accessibility relations *R's* of the Kripke structures are interpreted as *causal relations* defined on the physical causal light-cone (see Figure 1), given that the causal event satisfies the notion of *terminal object* in the (sub-)category of its effects (see Section 1.2 and Definition 7.). This, again, emphasizes the coalgebraic nature of the causal light-cone in Kripke's relational semantics.

Finally, what makes Kripke semantics so attractive in logic is the possibility of satisfying, because of the Van Benthem's "correspondence theorem" [155,156], both a "second-order relational semantics" on Kripke *frames*, in which we quantify over all the *valuation functions* ($\forall V$) for the proposition validity over all the possible worlds, and a "first-order relational semantics" on Kripke *structures*. In it, the validity holds for a given state of the world, and for all the other ones accessible from this state. In this way, we can always quantify at the *first-order*, either over *all* states ($\forall x,y$) of a given universe *W*, e.g., when we state something about our universe as "generated" from the big-bang causal event at its origins, or over *some* states of the universe causally accessible from another state ($\exists x,y$), that is, over a partition of the universe ($W' \subseteq W$). In both cases, however (universal and particular quantification), we are referring to the first-order notion of *local* truth of the type: *it is locally the case* (F. W. Lawvere) (see Chapter 14 of [7], pp. 359–437). For instance, an example is when we speak about the radiating electromagnetic force that is true in our expanding universe only after the stage in which "cosmological microwave background radiation" (CMBR) originated, about 300,000 years after the big bang. That is, to speak

about the electromagnetic radiation, and then of Maxwell's electromagnetism laws, is true only for a *partition* of possible universe states, from a stage of the universe evolution onward. "Cosmogony is the legislator of physics", and Kripke's relational semantics is its logic!

This logical evidence is the core of Patrick Van Benthem's celebrated *correspondence theory* between modal relational semantics and a variable-free fragment of first-order logic [155–157]. Indeed, coming back to Lewis' axioms in ML (see Section 1.1), in addition to the "necessitation axiom" **N** defining the "normal modal system" **K,** all the other axioms of modal systems (**T, D, 4, 5, . . .**) used for axiomatically extending **K** can be defined via *first-order formulas*.

In this light, the result of Theorem 2. associated with Definition 13. allowed logicians and computer scientists to think at axiomatic extensions of **K** (e.g., **KT, KT4, KT5, . . .** modal systems) not as giving rise to new systems of the modal calculus but as different *theories* over the minimal system **K,** "just as a first-order theory (e.g., of linear orders) is constructed over first-order validities" [42], p. 35.

This means that we can answer the fundamental question about the types of Kripke frames $\mathfrak{F} = \langle W, R \rangle$ that are able *to validate* the different modal axioms extending **K**. Because of Definition 13., whereas we interpret the generic frame S of the CT relational semantics (see Section 5.1.4) as a Kripke frame $\mathfrak{F}$ simply by limiting $R$ to only dyadic (accessibility) relations, if the property defining a particular class of Kripke frames is *a first-order relation over sets*, it is possible to answer the precedent question. In fact, it is enough to interpret the set **Γ** of *modal formulas* extending **K** as the set of *modal axiomatic formulas extending* **K.** In a word, in Kripke modal semantics, we are faced with a particular solution of the famous "Löwenheim -Skolem paradox" [158] in set-theoretic semantics.

According to this paradox, indeed, despite all the axioms of **ZF** set-theoretic semantics being expressible through first-order formulas, nevertheless, their justifications need a higher-order logic. In modal semantics, however, because a modal formula evaluation typically distinguishes between "actual" and "possible" states (worlds) satisfying it, we can have different semantic levels because of the two double dichotomies: 1) *Kripke frames*: $\mathfrak{F} = \langle W, R \rangle$, versus *Kripke structures (models)*: $\mathfrak{M} = \mathfrak{F}, V$; and 2) *total*, versus *local* truths that we introduce now and justify as follows (see [43], p. 252).

Indeed, given that the *basic* semantic notion in Kripke modal logic is the *truth of a formula at a world-state* $\langle w \rangle$ *in a Kripke structure (model),* this notion is *local* and of a *first-order* nature. Therefore, the passage from structure semantics to frame semantics and then from *local* to *total* truth depends on looking or not at *all the valuations over a frame*. Namely:

1. By an abstraction through a universal *second-order* quantification over all *valuations* on *propositional formulas*: $\forall V(V(\phi))$.

2. Or, on the contrary, as we anticipated some paragraphs before but we now discuss in a more formal way, we can pass in Kripke's model theory to a first-order semantics of a propositional formula $\phi$ in a *total* and/or in a *local* way if we do not quantify over all the valuation functions (second-order), but if we quantify at the *first-order* over *all* (respectively *some*) possible worlds. Namely:

   a. Either, over *all* the possible world-states $w$ of the universe $W$ for which a given formula $\varphi$ is true, i.e., $\forall w \in W(V(\varphi_w))$: *total* truth.

   b. Or, over a *restriction of possible world states* for which a given formula $\phi$ is true: *local* truth. Specifically, $\forall w' \in W' | W' \subseteq W(V(\varphi_{w'})$. Namely, if $R \subseteq W \times W$ is any binary relation over $W$, and $W' \subseteq W$, we write $R \restriction W'$ for the *restriction* of $R$ to $W'$, i.e., $R \restriction W' = R \cap (W' \times W')$. Similarly, for a valuation $V$ on $W'$, $V | \restriction W'$ stands for the restriction of the evaluation function to the formula defined on the partition $W'$.

This means that in Kripke modal relational semantics, we can validate a formula $\phi$ over a given world-state $w$ and on the sub-set of all the possible world-states accessed by $w$, which is evidently the logic of the light-cone causality in fundamental physics.

For the sake of brevity, we cannot develop a full formal treatment of the CT notion of meaning function here, applied to Kripke's modal semantics for which, among a wide literature, we essentially refer to two fundamental chapters of the monumental *Handbook of Modal Logic* (see [6], and especially [43]). We can then synthesize the main theoretical passages of their treatment as follows.

The first step is to extend **K,** by introducing the notion of Kripke frames *validity* that in turn is a *modal* extension of the frame validity in relational semantics (see Definition 13.). Namely [43], p. 253:

**Definition 18.** *(Kripke frame validity). Let φ (p₁, ... , pₙ) be a modal formula consisting of the atomic proposition symbols p₁, ... , pₙ, with the associated monadic predicates for each atomic proposition, P₁, ... , Pₙ. φ is locally valid on a frame at a state (point, node, world) w ∈ W, if for each valuation V of its proposition symbols, φ is satisfied in the resulting model at w, i.e., M, w ⊨ φ, for each M over F, so that we write F, w ⊨ φ. Specifically, φ is valid in the pointed frame (F, w). Consequently, we say that φ is valid in F, denoted as F ⊨ φ, if F,w φ for every w ∈ W. Finally, φ is valid, denoted as ⊨ φ, if F ⊨ φ for every frame F.*

To sum up, we can say that a first-order modal formula *φ* defines *a class of frames* F if it is valid on every frame in F and falsified on any frame that is not in F. Then, we can define several classes of frames, as far as satisfying basic modal formulas, in the classical axiomatic approach to ML [14]. There are effectively as many axiom schemes for modal systems, extending the normal system **K.**

In this introductory exposition, for sake of clarity, for each (Lewis') axiom and the correspondent FO relation in a Kripke's frame, we make its significance for the philosophical logic explicit, according to Van Benthem's *correspondence theory* ([42], pp. 36). Of course, this makes explicit the subdivision into "three ages" (axiomatic, Kripkean, algebraic) of the modern history of ML during the XX cent., as we anticipated in Section 1.1, quoting [28]. In the present expositions, we limit the application of the correspondence theory only to some modal axioms: **T, D, 4, 5(E)** and their definitions that are more relevant for an ontology of physics.

**Proposition 1.** *(Definition of some classes of Kripke frames validating Lewis' modal axioms).*

- $\Box p \to p$ ($\equiv_{\text{def}}$ **T**) *defines (is validated by) the class of frames, which consists of isolated reflexive points/worlds such that* $\forall x,y$ *(Rxy ↔ x = y).*

  ○ *[The meaning of axiom* **T** *(from "truth") is evident: it is the axiom scheme of all alethic logics in modal formal logics and ontologies. It says, indeed, that if a proposition p is true in all possible worlds, it is evidently true also in the actual one]. For example, if the Galilean law of falling bodies is true in all possible physical worlds, it is also true in ours.]*

- $\Box p \to \Diamond p$ ($\equiv_{\text{def}}$ **D**) *defines the class of frames where the frame relation R is "serial":* $\forall x \exists y$ *(Rxy).*

  ○ *[The meaning of axiom* **D** *(from "deontic") is evident too. It says that if p is necessary, it is possible as a necessary condition. It is therefore the axiom scheme of all "deontic logics". Nobody, indeed, can be morally or legally obliged to something that is impossible for him/her: the possibility of satisfying a moral oughtness is a necessary condition of the validity of a moral obligation ("impossibilia nemo tenetur", in Latin). In the difference between axiom* **T** *and axiom* **D***, the core of the famous "Hume principle" of not confusing alethic and deontic necessity is hidden, the "world of facts" and the "world of values"].*

- $\Box p \to \Box\Box p$ ($\equiv_{\text{def}}$ **4**) *defines the class of frames, where the frame relation R is "transitive":* $\forall x,y,z$ *((Rxy ∧ Ryz) → Rxz).*

  ○ *[The meaning of the axiom* **4** *is, indeed, the "transitivity of necessitation's". It is typically the axiom of the modal formalization of the "scientific necessity" according to*

> *distinct levels of necessitation (ordered natural laws) against any naïve reductionism].*
> *For example, the laws of physics are necessary also in chemistry, even though they are*
> *not sufficient for justifying all the chemical phenomena, etc.]*

- $\Diamond p \rightarrow \Box \Diamond p$ ($\equiv_{\text{def}}$ **5** *or* **E**) *defines the class of frames, where the frame relation R is "Euclidean",*
  *sometime denoted as a "weak transitivity": $\forall x,y,z$ ((Rxy $\wedge$ Rxz $\rightarrow$ Ryz)).*

  - *[The axiom **5** (in Lewis' enumeration) or **E** (from "Euclidean") is in some sense the*
    *axiom of the formalization of "metaphysics", because it states that if something is*
    *possible, it is "necessarily possible". In this sense, it formalizes the notion of "faculty"*
    *as a "power" that necessarily pertains to something/somebody because it characterizes*
    *its/her/his "nature" or "essence"]. For example, think of the faculty or the "necessary*
    *possibility" of thinking or of freely deciding as characterizing each human person, as*
    *an irreducible subject of rights and duties in society.]*

- *(…)*

As we know, by properly combining modal axioms, we obtain different systems of the modal calculus, each constituting the common "syntax" of different philosophical theories (i.e., of different "semantics"). Therefore, we can say, for instance, that Lewis' axiomatic modal system **S4**, i.e., **KT4,** is defined over a frame-class simultaneously satisfying the reflexive and transitive accessibility relations. The axiomatic system **S5**, i.e., **KT45,** is a **KT4** defined over the frame class that also satisfies the Euclidean relation characterizing the axiom **5**. On the contrary, with respect to **S5**, the **deontic S5** system, i.e., **KD45** that substitutes the axiom **T** with **D** as is necessary in deontic contexts, is defined over Kripke frames that satisfy the serial relation instead of the reflexive one, and so on.

Let us discuss some exemplifying applications in philosophical logic (modal semantics) of two modal systems: the **KT45 (S5)** and **KD45**, which is also defined as a **generated S5** system, for the reason we explain below, which has applications not only in formal ethics (deontic logic: **deontic S5**) but also in formal epistemology and formal ontology. Indeed, the **S5** modal system is the more powerful one because in a proper sense, it includes all the other ones [14].

Not casually, therefore, **S5** is universally recognized as the "modal syntax" of whichever *metaphysical theory* because it effectively represents a complete *universe of possible worlds W* as constituting only one equivalence class of entities that all "might exist" according to the very same defining axioms of a given metaphysical theory. Let us think, for instance, of a physical theory satisfying a TOE that would reveal in this way its "metaphysical" nature. In Figure 11, the right graph is a representation of the **S5** system for an oversimplified universe of only three worlds.

Now, if we observe Figure 12, the left graph represents a **KD45** system for a simplified four-world universe, justifying its definition as a **generated S5** system, with the inaccessible world $u$ as the "generator" of an equivalence class of the other three worlds because all are accessible to $u$ but not vice versa. The right part of the same figure represents for the sake of simplicity the calculus of relations generating an equivalence sub-class of two worlds for an oversimplified **KD45** system of a three-world universe, with the world $u$ as the generator, according to the following relational steps: ($\forall u,v,w$) (($uRv \wedge uRw) \rightarrow vRw$) > [transitive rule: **4**]; ($\forall u,v,w$) (($uRv \wedge uRw) \rightarrow (vRw \vee wRv$)) [Euclidean rule: **5**]; ($\forall u,v,w$) (($uRv \wedge uRw) \rightarrow (vRv \vee wRw$)) [serial rule: **D**]. To sum up, we *can generate a transitive-symmetric-reflexive (equivalence)* relation for the world sub-class $\{v, w\}$ starting from the asymmetric accessibility relation of $\{u\}$ with $\{v, w\}$ by a suitable "composition of morphisms" (accessibility relations $R$) satisfying the **KD45** axioms, i.e., ($\forall u,v,w$): (($uRv \wedge uRw)) \rightarrow (vRw \wedge wRv \wedge vRv \wedge wRw$) (see Figure 12 (Right)).

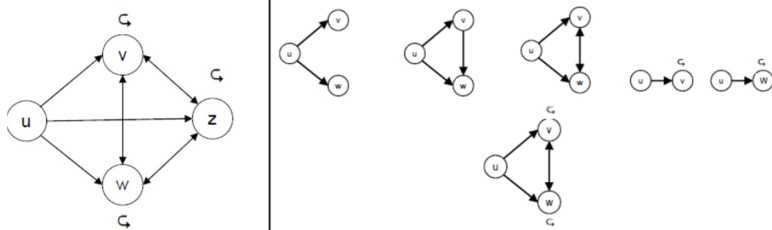

**Figure 12.** (**Left**): Representation of a KD45 modal system for a four-state world *W*, where the frame sub-class $\{u, w, z\}$ constitutes an equivalence class generated by only one inaccessible world-state $\{u\}$. KD45 constitutes a "secondary" or **generated S5** system. (**Right**): Oversimplified calculus of relations underlying a three-state world *W* for a **generated S5** system, where the accessibility relations from an inaccessible world-state $\{u\}$ generates the equivalence sub-class $\{v, w\}$ (see text).

When we integrate this simple relation calculus with the powerful construction of the *generated rooted trees* of Kripke structures over a coalgebra of NWF-sets, we can understand why a *nested structure* of **KD45** systems is the proper logic of the NR-formal ontology as I summarize in the general conclusions (Section 6), and as I anticipated in a semiformal way in [30], and, finally, as proposed for the first time by Francesco Panizzoli in [159].

As a further step, what about the *extensions* of Kripke frame definability and Boolean validity in the framework of the above Definition 13. and Theorem 1. concerning the "meaning function" $[\![\phi]\!]$? For this, let us extend the notion introduced in Section 5.1.3 of "bounded morphism" for a topology of *Kripke structures (models)* $\mathfrak{M}$. Specifically, these wre Kripke frames $\mathfrak{F}$, endowed with *valuation functions V*, over the atomic propositional symbols *p* of molecular modal formulas $\phi$ associated at each state/point *w* ä *W* (see [43], pp. 258–259). Let us, therefore, define the fundamental notion of "bounded morphism" for Kripke models that we introduced in discussing in the Section 5.1.3. Jónsson–Tarski Theory of the "Boolean Algebras with Operators".

**Definition 20.** *(Bounded morphism between Kripke structures).*
Let $\mathfrak{W} = \left(W, \{R_\nabla\}_{\nabla \in \tau}, V(\varphi)\right)$ *and* $\mathfrak{W}' = \left(W', \{R'_\nabla\}_{\nabla \in \tau}, V'(\varphi)\right)$ *be two Kripke structures, where* $\nabla$ *is a modal connective of the set* $\tau$ *of modal connectives, and* $\varphi$ *is the meta-symbol for propositional variables p. A morphism* $\rho : W \to W'$ *is a bounded morphism from* $\mathfrak{M}$ *to* $\mathfrak{M}'$ *if its graph is a bisimulation (see Definition 14.) from* $\mathfrak{M}$ *to* $\mathfrak{M}'$, *denoted as* $\mathfrak{W} \overset{\rightrightarrows}{\to} \mathfrak{W}'$. *Bounded morphisms between frames are similarly defined.*

**Remark 2.** *It is important to emphasize the "truth preservation" property of the bounded morphisms. Specifically, if* $\rho : \mathfrak{W} \overset{\rightrightarrows}{\to} \mathfrak{W}'$ *is a bounded morphism and* $\varphi \in ML_\tau$, *then* $\forall u \in dom(F)$: $\mathfrak{M}, u \models \varphi$ *if* $\mathfrak{M}', \rho(u) \models \varphi$. *Therefore, if* $\mathfrak{F}, u \models \varphi$, *then* $\mathfrak{F} p(u) \models \varphi$.

**Remark 3.** *If* $\rho$ *is defined onto Kripke structures, then* $\mathfrak{M}'$ *is a "bounded morphic image" of* $\mathfrak{M}$ *(the same holds for frames). Therefore, for each* $u \in W$, *a bounded morphism* $\rho$ *"uniquely singles out" a bisimilar state* $\rho(w)$ *in W'.*

**Remark 4.** *Each model* $\mathfrak{M}' = \mathfrak{F}', V'$ *over frame* $\mathfrak{F}'$ *can be "pulled back" to give a model* $\mathfrak{M} = \mathfrak{F}, V$ *over the frame* $\mathfrak{F}$ *via* $V(p) := \rho^{-1}[V'(p)] = \{w \in \text{dom}(\mathfrak{F}) \mid \rho(w) \in V'(p)\}$. *This turns* $\rho$ *into a "reversed bounded morphism" from* $\mathfrak{M}'$ *to* $\mathfrak{M}$, *i.e.,* $\mathfrak{W} \overset{\leftleftarrows}{\leftarrow} \mathfrak{W}'$. *Nevertheless, not every model* $\mathfrak{M}$ *over* $\mathfrak{F}$ *can be obtained by such a construction.*

Indeed, it holds only for Kripke models (structures) $\mathfrak{M}'$ endowed with a *selection criterion* of the admissible sets on which the semantics of $\mathfrak{M}$ can be defined, as we see immediately. Now, what about the truth preservation in the case that $\mathfrak{F}'$ is interpreted as an *extension* of $\mathfrak{F}$ in the sense of S and S' of Definition 13.? Effectively, this application of the relation semantics notion of "bounded morphism" (already introduced as BAOs in

Section 5.1.3) to Kripke models/structure is based, as the same used symbol ($\overrightarrow{\rightarrow}$) emphasizes very well, on the notion of "bisimulation" of Definition 14. (symbol: $\leftrightarrows$), where the "long arrow" in $\overrightarrow{\rightarrow}$ stays for an endofunctor in the category of Kripke models.

Indeed, we know that the first usage of the bisimulation notion was by Aczel to justify the "set unfolding" of subsets from the common root of NWF-sets that acquires all its logical relevance in the light of the functorial dual equivalence **SCoalg(Ω) ≃ MBA(Ω>\*)**, and all its *ontological relevance* when we recall that BAOs are defined on topological measurable (probability) spaces ($\sigma$-algebras: see Note 30) such as the Hilbert spaces. In this way, for what we discussed in Section 4.8, the same dual equivalence holds if we substitute the category **SCoalg(V)** for the Vietoris endofunctor, with the category of **qHCoalg($\mathcal{B}$)** for the Bogoliubov endofunctor at the coalgebraic footing of the category of the "doubled Hilbert spaces" **DHilb($\mathcal{B}$)** and its cocompleteness property (see Section 4.8 and Appendix A). Indeed, because the algebraic footing of the "doubled Hilbert spaces" consists in the non-commutative $q$-deformed Hopf coproducts, as explained in Section 4.5 and Section 4.6, this means that the dual equivalence **SCoalg(Ω) ≃ MBA(Ω\*)** for the Vietoris functor $\mathcal{V}^* := (\cdot)_* \circ \Omega \circ (\cdot)^*$ can be rewritten as **qHCoalg(Ω) ≃ MBA(Ω\*)** for the Bogoliubov functor: $\mathcal{B}^* := (\cdot)_* \circ \Omega \circ (\cdot)^*$.

Indeed, as the Vietoris functor $\mathcal{V}$ in ML endows the system with a first-order criterion of the admissible sets, making the $\mathcal{V}^*BA$ dually associated with ("induced by") its $\mathcal{V}SCoalg$ naturally "modal" as explained below, the same also holds for the Bogoliubov functor $\mathcal{B}$ with respect to the category of the $q$-deformed Hopf coalgebras in QFT. Indeed, the category of **qHCoalg($\mathcal{B}$)** is also endowed with a (dynamic) selection criterion of the admissible sets naturally making $\mathcal{B}^*BA$ dually associated with ("induced by") its $\mathcal{B}qHCoalg$ "modal". A further advantage is that the $q$-deformed Hopf coproducts and then the associated $q$-deformed Hopf Coalgebras are dynamically indexed via the construction of the $\aleph_0$-directed (i.e., denumerable locally presentable) comma category of the associated colimits, as explained in Section 4.8 and Appendix A.

The indexing value $\mathcal{N}$ defined on positive integers and univocally denoting a given condensate of NG-bosons, characterizing (or dynamically labeling) a given phase-coherence domain of a degenerate QV state ("QV-foliation") in a dissipative QFT system "balanced" with its environment, therefore confirms itself, also from the logical and computational standpoints, as a *finitary computable* "dynamic memory address" for biological systems and for natural and artificial neural systems (see Section 4.7). On this regard, see [127], and more recently [154] for an application addressed to the "data streaming" problem in machine learning. Finally, it is worth emphasizing that the non-commutative character of the $q$-deformed Hopf coproducts implies the non-commutative character of the associated "skew" Boolean algebras (see Note 6 and [160]), enhancing the "discriminant" semantic power of this computational architecture based on the "double qubit" computational principle (see above Section 4.6).

Therefore, coming back to Kripke's model theory, it is not casual that in logic and TCS, one of the most important constructions related to the bounded morphism equivalence between Kripke models, and with the pullback of evaluation functions in them, is the *unfolding* or *tree unravelling* of a Kripke structure modeled on a coalgebra of NWF-sets according to the categorical duality **SCoalg(Ω) ≃ MBA(Ω\*)**. To understand this, see the powerful notion of *generated and rooted sub-structures (sub-frames)* of a Kripke structure (frame) ([43] pp. 259–261).

For the moment, and for the sake of simplicity, we can say that the rooted graphs of Figure 10 can also be interpreted as representing such a construction of *generated rooted trees* of point-set Kripke structures, where each point represents a world-state $w \in W$. Now, the core of the dual equivalence **SCoalg(Ω) ≃ MBA(Ω\*)** for the contravariant application of the Vietoris functor $\mathcal{V}^* := (\cdot)_* \circ \Omega \circ (\cdot)^*$ is that the Vietoris endofunctor in the category of the coalgebras of NWF-sets **SCoalg($\mathcal{V}$)** acts as an equivalent of the power-set functor (see [6], p. 418). Therefore, when we apply the Vietoris construction to the rooted trees of Kripke structures defined on NWF-sets, this allows us to define the relation of *converse*

*membership* (*comembership*) $\ni$ in the category **SCoalg($\mathcal{V}$)** as functorially dual to the relation of *membership* $\in$ in the equivalent category **MBA($\mathcal{V}$*)** (see [6], p. 393 for a formal exposition). In other terms, the Vietoris construction in this case acts as a first-order (*local*) selection criterion of admissible sets for the Boolean logical calculations.

  For our aims, apart from the formalisms, it is very important to understand the logical meaning of this construction from a philosophical standpoint. As we anticipated many times, what characterizes the set "unfolding" of subsets from the root of an NWF-set tree is that because of the "anti-foundation axiom" (see Section 2.5), the set-subsets relationship along the edges of the set trees can be justified along reciprocally irreducible, infinitely many *arbitrary* "paths". This means that we cannot use the usual set-subsets inclusion relationship $\subseteq$ such as in **ZF** that supposes the set total-ordering to signify the membership to the root-set of the "disjoint unions (coproducts $\cup$))" of subsets along different unfolding paths or of the "intersections (products $\cap$)" of unfolded subsets along different paths but the modal notions, respectively, of "possible comembership" $\ni$ and "necessary comembership" $[\ni]$ ([6], p. 393). Here, the angular and square parentheses stay for the modal operators $\Diamond$ and $\Box$, respectively, but *relativized* to some partitions of the whole set of possible world states, and so justify FOL of *local* truths.

  In this way, we can express this modal comembership of subsets to a set by saying that in the coalgebraic Kripke rooted structures on NWF-sets—constituting the extensions validating in the reversed direction $(\mathbb{F} \vDash \varphi) \leftarrow (f\mathbb{F}') \vDash \varphi)$ the correspondent modal Boolean propositional formulas because the extensions are endowed with a first-order *principal filter condition f* (see Section 5.1 and above)—the superset (i.e., the common root of the NWF-set tree) *admits* or better *generates* (not includes) its subsets. Just as we are acquainted to say, when we speak about "natural kinds" in a naturalistic ontology, that a *genus* (e.g., "mammalians") *admits* (not includes!) its several different *species* ("horses", "dolphins", "elephants", etc.), given that the genus–species relationship cannot satisfy any total ordering condition. Which ordering relation $\leq$, indeed, could be defined between different species and between subsets belonging to different branches of the unfolding process of NWF-set trees (see Figure 8)?

  On the other hand, from the ontological standpoint, the notion of *generated* rooted trees of Kripke structures rigorously formalizes the notion of the *local* truth of a formula, evaluated at a current (actual) state of the world, and preserved (and carried) along the edges of accessibility (=causal) relations to other states. This justifies the usage of a "relativized" (indexed) universal quantifier and the related necessity modal operators. Both characters make this coalgebraic modal logic a "guarded fragment of FOL" (see [43], p. 323), particularly suitable for modeling an evolutionary cosmology (and biology) based on QFT as fundamental physics, where the "cosmogony is the legislator of nature". Therefore, let us discuss briefly, to conclude this section, the logical and ontological relevance of these constructions for a formalized philosophy of nature.

  In the light of what we have discussed till now, it is easily understandable why, since its first appearance during the 1960s [15,16], Kripke's possible world semantics seemed more adequate for logically representing the contemporary evolutionary cosmology than the classical Tarski model theory, whose truth condition is for *only one* state of affairs, i.e., for only one "actual world" [17]. Particularly, the relational semantics of possible (states of) world(s), where a formula is evaluated at the *current (=actual) state* of the world and preserved along the edges of the accessibility relations to other possible states, seemed immediately consistent with the causality principle of the *light-cone* of special relativity theory [20].

  Based on what we demonstrated in this contribution, we therefore have to interpret the accessibility relations of the rooted trees of Kripke models/structures on a coalgebra of NWF-sets as *causal relations* according to the light-cone causality principle in fundamental physics.

  In this way, the modal distinction suggested by Kripke in *Naming and Necessity*—and acclaimed as one of the most relevant contributions to the analytic philosophy movement of

the XX cent. (see Section 1.1)—between "*epistemic* (logic) necessity" and "*ontic* (causal) necessity" and then between *logical classes* (class-subclasses logical structures) and *natural kinds* (genus-species, intended as causally generated structures), where the latter ones are validating the former ones, has its proper formalization in the "reversed validation" between Kripke coalgebraic and algebraic models. This holds in the framework of the categorical dual equivalence **qHCoalg(Ω) ≃ MBA(Ω\*)**, which is the counterpart in the ontology of fundamental physics of the dual equivalence in modal logic **SCoalg(Ω) ≃ MBA(Ω\*)**.

This "local" validation of the (Kripke) models of the modal descriptive philosophical language over the (Kripke) models of the mathematical language of physics and then of natural sciences is therefore the semantic core of the *NR formal ontology* we propose in this contribution as a formalized philosophy of nature. This ontology is, at the same time, our categorical version of the approach of the so-called *ontic structural realism* in the philosophy (ontology and epistemology) of quantum physics while at the same time solving the debate between its "causal" and "statistical" justification [1,2,161]. The DDF principle in QFT, indeed, constitutes a dynamic (causal) selection criterion of the admissible sets, validating both the statistical and the logical representations of the QFT system at hand.

Historically, this ontology, as we discuss in Section 6.2, is a re-proposal, by the functorial *dual equivalence* coalgebra/algebra, of the Aristotelian categorical duality between the *ontic necessity*, i.e., the *cause–effect entailment* ("an effect without a cause is impossible") that means the converse implication between a cause $\alpha$ and effect $\beta$: $\Box((\alpha \Leftarrow \beta)) := (\neg \diamond (\beta \wedge \neg\alpha))$, where the cause $\alpha$ is the "necessary condition", and the dually correspondent *logical necessity*, the *premise-conclusion entailment* that means the direct implication between premise $\alpha^*$ and consequence $\beta^*$ ("it is impossible that a premise is true, and the consequence is false", that is $\Box((\alpha^* \Rightarrow \beta^*)) := (\neg \diamond (\alpha^* \wedge \neg\beta^*))$, where the cause $\alpha^*$ is represented as a "sufficient" condition made "sound" by the dually equivalent causal entailment.

In a word, coming back to our precedent biological example of the *logical truth* of the statement: "horses are mammals", the logical membership ($\in$) between the sub-class of horses and the class of mammals is functorially induced in the NR formal ontology by the *ontic truth* of the dual statement: "the genus of mammals causally admits ($\ni$) the species of horses" from a given step $m \leq n$ onward of the universe evolution. This happens for other species of mammals but following different generation paths from the common root of a shared progenitor. In synthesis:

$$\Box_{\forall n \geq m} \left( \underbrace{horse^* \in mammalian^*}_{\text{Modal Boolean Algebra}(\Omega^*)} \quad \underset{\underbrace{\Omega^*/\Omega}_{\text{Bounded Morphism}}}{\overset{\rightleftarrows}{\longleftarrow}} \quad \underbrace{horse \ni mammalian}_{\text{Stone Co-Algebra}(\Omega)} \right)$$

(13)

where the symbol $\overset{\rightleftarrows}{\longleftarrow}$ stays for a functorially induced "*onto/logical* dual equivalence" between modal statements, respectively, in a coalgebraic (*ontic, causal*) and algebraic (*logical, representational*) formalization. Effectively, this is a *reversed bounded morphism* $\mathfrak{W} \overset{\rightleftarrows}{\longleftarrow} \mathfrak{W}'$ between a physical coalgebraic Kripke model $\mathfrak{M}' = \langle W', R', V' \rangle$ and its logical algebraic dual homomorphic image $\mathfrak{M} = \langle W, R, V \rangle$, with *local* evaluations $V'$, defined on a world-state $w'_m \in W'$ and preserved along the states $w'_{n \geq m}$ *causally* accessible to it. This exemplifies the fundamental notion of *local truth* in modal relational semantics illustrated above, which therefore has its proper syntax in *nested structures/sub-structures* of **KD45** systems (see [43] for an extensive formal treatment of this logic of nested trees of Kripke models/structures).

## 6. Some Final Remarks from a Historical Perspective

### 6.1. The Logic of NR-Formal Ontology as a Formalized and Ecological Philosophy of Nature

I showed in this work the usefulness of using CT as the proper metalanguage of formal philosophy, and specifically of formal ontology and formal epistemology, and therefore for a formalized philosophy of nature and philosophy of science, respectively. Specifically, this is the formal ontology and epistemology of the "natural realism" (NR).

This justifies why I dedicated Section 2 of this work to make philosophers more acquainted with the main notions of CT, in a direct comparison with ST, more known by philosophers of the analytic tradition, with special attention given to the axiomatic modal logic as the proper logic of the philosophical languages.

The main results that I tried to show using this formalism are the possibility of modeling and validating the descriptive statements of an ontology and an epistemology of the natural sciences directly on their mathematical models at the level of their fundamental physics (QFT), in the common framework of the operator algebra formalism extended from physics to logic.

The *physical core* of the NR formal ontology (see Sections 3 and 4) is summarized in the possibility of modeling the powerful construction of the "QV-foliation" for classes of dissipative QFT complex systems sharing the same dynamic (causal) structure, in terms of the categorical *universal* construction of the functorial "comma category" (see Appendix A). In this categorical construction, each class of complex quantum systems is modeled as the one-object category **1**, and then as a "colimit" or the "initial object" of the (cocone of morphisms of the) correspondent comma category $\mathcal{C} := 1 \xrightarrow{c} \mathcal{C} \xleftarrow{F} \mathcal{J}$, so that its complex structure is indexed by the correspondent "diagonal functor" $\mathcal{C}^{\mathcal{J}}$ (see Section 4.8 and Appendix A). In this way, it is possible to also give a physical foundation to the mathematical methodology of the "memory evolutive systems" (MESs) proposed by Ehresmann and Vanbremeersch in another paper of this journal issue as the core of a renewed philosophy of nature to formalize the fundamental notions of *emergence, complexity,* and—in the case of biological and neural systems—*memory* [38]. This methodology is extended in our interpretation from an evolutionary biology to an evolutionary cosmology, as required by a formalized philosophy of nature.

Moreover, the theoretical core of the extension of QFT to dissipative quantum systems, consisting of the DDF principle between the system and its thermal bath, means the proposal of a fundamental physics that is *intrinsically ecological.* Indeed, it is compliant with a generalized vision of physical systems that can no longer be conceived as *closed* systems. Additionally, this can be performed without renouncing the mathematical apparatus of statistical mechanics, as it is synthesized in a nustshell by the possibility of recovering the Hamiltonian canonical representation of dynamic systems simply by inserting the environment (thermal bath) degrees of freedom. Simply, we must consider from a new local perspective the perturbative methods of statistical mechanics as a useful abstract tool for modeling dynamic classical and quantum systems that does not capture, however, the intrinsically dissipative character of the "many-body physics".

On the other hand—without charging the physical research and practice of excessive social and cultural responsibilities—it is straightforwardly evident that a mathematical physical formalism conceiving in a generalized way physical systems as closed systems has contributed significantly to the development of a modern technology that is disrespectful of the environment. In this framework, indeed, the environment is naturally considered as an "irrelevant boundary condition", only good for being indefinitely energetically despoiled. This approach, indeed, is compliant with a modern culture that is no longer aware of the intrinsic *relational nature* of whichever physical non-living and living entity, the human individuals included. For example, to pass to social contexts the ideologies of the *individualist liberalism* or the *collectivist communism* characterizing the Modern Age, both forget the essential character of the human individual as a *person,* i.e., as an *individual-in-relation* with her natural and social environment, starting from her biological and cognitive faculties. Both ideologies, indeed, forget at the level of the social, economic, and political modern sciences the fundamental ethical consequence of a personalistic anthropology. Namely, the "common good" that society ought to pursue is the *personal flourishing* of human individuals and groups (see [162]).

Of course, such an "ecological" approach to fundamental physics and its ontology is perfectly compliant with the NR formal ontology as a categorical version of the *relational*

*natural realism* (see Section 1.4 and Figure 2), here proposed as a formalized philosophy of nature and sciences.

Indeed, the *mathematical and logical core* of the NR formal ontology is the possibility, in the framework of the CT formalism, of using Kripke's coalgebraic "many world" relational semantics, both on the *physical* or *ontic side*, and on the *logical side*. On the ontic side, this is because it is compliant with the causal light-cone in fundamental physics for the mathematical modeling of QFT systems. On the *logical side* of the modal logic—effectively a modal BAO—is the descriptive language of ontology. In both cases, indeed, Kripke's model theory can express all its mathematical and logical representational power if models are defined onto a coalgebra of generated rooted trees of NWF-sets, validating the correspondent (modal Boolean) algebraic models of the descriptive language of an ontology. This depends on the fact that the former ones are endowed with a selection criterion of the admissible sets, the DDF principle in QFT, on which the semantics of the ontology can be truthfully defined. The NWF-set theory, indeed, via its "anti-foundation" axiom, makes a set capable of "containing" itself $\{\{\cdot\}\}$, so violating **ZF** set theory "total ordering", and then allowing an indefinite number of *arbitrary* "partial orderings", along different unfolding paths sharing the same root.

This is the deep reason for which the "generated rooted-trees" of Kripke structures and models defined onto a (topological) coalgebra of NWF-sets (see Section 5.2.3 and [43]) can constitute the proper representation in mathematical logic of the QFT process of the "QV-foliation", indexed by the comma-category of colimits applied to the (non-commutative) Hopf coproducts of QFT, such as many "unitarily inequivalent representations" of QV. The related *q*-deformed Hopf coproducts (coalgebras) constitute, indeed, only one category for the Bogoliubov endofunctor, as discussed in Sections 4.8 and 5.2.3 of this paper. For this reason, I showed that by the powerful CT logic construction of the (functorially reversed) "bounded morphism" between Kripke's coalgebraic models in the mathematical physics, and Kripke's algebraic models in the modal languages of the natural philosophy, the statements of the latter are validated onto the statements of the former.

All this holds in the logical framework of the dual equivalence **SCoalg(Ω) ≃ MBA(Ω\*)** between the category of coalgebras on NWF-sets defined onto Stone spaces and the category of modal BAOs for the contravariant application of the Vietoris functor $\mathcal{V}^* := (\cdot)_* \circ \Omega \circ (\cdot)^*$ (see [6]). In this contribution, we extended this logic to the *non-commutative case* of the dual equivalence, **qHCoalg(Ω) ≃ MBA(Ω\*),** between the category of the *q*-deformed Hopf coalgebras on NWF-sets defined onto the Chu spaces of QFT and the category of modal BAOs of the ontological descriptive languages for the contravariant application of the Bogoliubov functor $\mathcal{B}^* := (\cdot)_* \circ \circ (\cdot)^*$.

Finally, this *onto*-logic semantics of the NR-formal ontology is a significant realization of what we anticipated as being the core of the CT logic. Namely, the fact that a statement $\alpha$ over a category $\mathcal{C}$ is true if and only if the opposite statement $\alpha^{\mathrm{op}}$ in the dually equivalent category $\mathcal{D}^{\mathrm{op}}$ is true (see Section 2.4).

### 6.2. The NR-Formal Ontology from a Historical Perspective

From a historical standpoint, this particular version of the *relational natural realism* (see Section 1.4) is consistent with the anti-Platonic solution of the long-standing issue of the *realism of universals* in the history of logic and epistemology, having its ancestor in the Aristotelian naturalism. We emphasized in Section 2.1 that the categorical diagrammatic formalization of the Aristotelian syllogism, as Peirce first emphasized in the Modern Age [163], is more adequate than the Leibnizian extensional interpretation that, as such, is unable to formalize the modal syllogistic forms [164]. The intrinsic "relational character" of the Aristotelian logic is confirmed by the *dual character* of the epistemology and ontology of the Aristotelian modal naturalism. Indeed, the dual core of the Aristotelian naturalism is synthesized in the famous Aristotelian motto according to which, epistemologically, "what is *last* in being (the cause known starting from its effects), is *first* in reasoning (it becomes the *sound* premise of the demonstrative reasoning) and vice versa" [35].

Of this dual relationship, during the Middle Age, Thomas Aquinas gave a justification in terms of the modal propositional logic, well known by him [36], in his commentary on a passage of the Aristotelian Second Book of *Physics* (II, 199b,34ss.). In this passage, Aristotle used this reversal of the arrows between *real (causal)* and *logical* relations to distinguish between the *logical* necessity of the mathematical demonstrations, i.e., the *logical entailment* "premise → conclusion"), and the *physical* necessity of a causal process, the *causal entailment* "effect → cause", since from the same cause several different effects can derive.

In his commentary on this passage, Aquinas, after having correctly explained the two fundamental logical laws (tautologies) in the hypothetical reasoning of the *modus ponens* and the *modus tollens,* and the relative "paradox of the consequent", for which the truth of the consequence cannot grant the truth of the premise, since "sometimes true consequences can be derived by false premises", to logically justify the Aristotelian "reversal of the arrows" between the logical and the causal entailments, Aquinas adds:

> However, in those things that happen *because of something* (*propter aliquid*), either by technology (*secundum artem*), or by nature (*secundum naturam*) [37], this reversal holds, because if some final state [effect] is or will be, *it is necessary that something before this final state, or will have been or is* [cause]. (. . . ) Therefore, the similarity is from both sides, even though with *an inversion of the relation between the two ones* (*quamvis e converso se videatur habere*). ([86] *In Libros Physicorum,* II, lect.15, n.5).

What Aquinas in this passage is implicitly affirming is the categorical duality, i.e., the reversal of the similarity (of the homomorphism) between the modal *logical entailment* "it is impossible that the premise is true, and the consequence is false": $\neg \diamond (\alpha^* \wedge \neg\beta^*)$, and the modal *causal (ontic) entailment* "it is impossible the effect without the cause" according to the physical causality principle, i.e., $\neg \diamond (\beta \wedge \neg\alpha)$. Synthetically, using the (functorially reversed) bounded morphism symbolism of CT for this universally valid *local* truth:

$$\Box_n(\alpha* \to \beta*) \xleftarrow[\Omega^*/\Omega]{\rightleftarrows} (\alpha \leftarrow \beta) \tag{14}$$

In this way, the well-known Aristotelian logical duality between the *causal syllogism* or *"quia"* (literally, "because") *syllogism* (from effect to cause, on the right side of the functorial dual equivalence above) can be easily understood. It gives the *demonstrative* or *"propter quid"* (literally, "because of which") *syllogism* (from premise to consequence on the left side) its *sound* (true) premise. It eliminates the possibility of the premise being false and the consequence true such as in the hypothetical syllogism without a causal entailment semantically validating it. In this way, which can be consistently formalized *only* in the functorial framework of CT logic, Aristotle and Aquinas obtained what C. I. Lewis searched for in vain with his theory of the purely logical "modal implication", as the correct metaphysical logic, as Quine's criticism of Lewis emphasized very well (see Section 1.2). For this reason, Aquinas, commenting on the Aristotelian teaching in the First Book of his *Posterior Analytics*, defined the demonstrative syllogism, based on *causally sound* premises, as the "because-of-which" (*propter quid*)" syllogism (see [86], *Expositio libri Posteriorum Analyticorum*, I, lect. 23, nn. 6–7; see also [87], *Summa Theologiae,* I, 2, 2).

In this sense, indeed, we must also interpret Aristotle's and Aquinas' epistemological theory of the *ontological truth* expressed in the motto of "the adequacy of the intellect and the thing" (*adaequatio intellectus et rei*) in the sense of a *universally valid local truth* in propositional logic (see the relativized modal operators (and/or quantifiers) in the bounded morphism of the modal formula (14)) [38]. This is based, indeed, on the *conformity (conformitas)*—in CT language, the *homomorphism up to isomorphism*—between the causal structure genus-species (or species-individual) of a natural kind, on the physical or *ontic* side, and the reversed logical structure or logical composition subject-predicate (the membership relation subclass/class) of a truthful sentence referring to the former, on the *logical side*.

The "circular conformity" (isomorphism via a natural transformation) of the adequation relationship is indeed, a homomorphism from the ontic side to the logical side (intellect ← thing) in the causal constitution ("formal causality" as a "homomorphic mapping") of a

*sound* predicative sentence as far as dually "mirroring" a causal generative process (natural kind) to which it refers. Afterward, it goes in the opposite direction from the logical side to the physical side (intellect → thing) to justify the *predictive* power of a sound premise in logic and mathematics of natural sciences, and so complete the semantic reference relationship (see [86], *Quaestiones De Veritate*, I, 1–4, for such a reconstruction).

This double reversed homomorphism—or isomorphism, effectively a natural transformation between the functors of the correspondent categories justifying a reversed *bounded morphism* between Kripke models in the CT modal coalgebraic semantics—therefore constitutes the *composite circular* notion of truth as *adequation*. This is epistemologically interpreted by Aquinas as an *intentional cognitive notion* (see the reference to the "appetitive faculty" or "emotional will" in the next quotation), straightforwardly synthesized by him in the following passage:

> The movement of the cognitive faculty terminates into the mind: it is therefore necessary that the known be in the knowing according to the knowing modality; on the other hand, the movement of the appetitive faculty terminates into the thing. Therefore, this is the sort of circle in the acts of mind that the Philosopher affirms in his III Book of *De Anima*. According to it, the thing that is outside mind moves the intellect, then the intellectualized thing (*res intellecta*) moves the appetitive faculty, and this directs itself toward the thing for reaching that from which the cognitive movement started (see [86], *Quaestiones Disputatae De Veritate*, I, 2co.).

Walter Freeman was therefore right in vindicating the continuity between his "action-perception cycle" in the QFT foundation of intentionality in cognitive neuroscience (see above Section 4.7) and Aquinas' theory of intentionality [165].

Unfortunately, this Aristotelian logical and epistemological theory of the ontological truth and its ML version given by Aquinas strictly depended on Aristotle's physics being abandoned, starting from the XIV-XVI centuries. Together with it, both the notion of *natural kinds* (genus-species) of the causal generative processes in nature and, more generally, the same modal logic were abandoned during all the Modern Age until their re-proposal in XX cent.

During the XVI cent. the duality between the causal and the logical entailments transformed itself—particularly in the work of the more known "Aristotelian" logician at that time, Jacopo Zabarella (1533–1589) —into the confused and confusing relationship between the *resolution* (inductive) and the *composition* (deductive) methods in logic, by which the duality between the "causal" *quia* (from the effect to the cause) and the demonstrative "logical" *propter quid* syllogisms, just recalled, is inconsistently interpreted without having the capability of distinguishing between the *real* and *logical* categorical modalities. Specifically, Zabarella interpreted the former as the *induction* of the premise (cause) from its consequence (effect) and the latter as the *deduction* of a consequence (effect) from its premise (cause), confusing the logical and the causal entailments that belong to different modal logic categories.

Now, Zabarella's fatal confusion, emphasizing how severe the abandonment of the modal logic in the XVI cent. was, deeply influenced the same debate between Galilei and the Inquisition at the beginning of the Modern Age. Indeed, as recently noted by Enrico Berti, one of the more recognized contemporary historians of the Aristotelian philosophy:

> It is (...) interesting, though not always remembered, the fact that Galileo also adhered to this (resolution-composition) method, which he learned from his visits as a young man to the Jesuits of the Roman College, who were profoundly influenced by Zabarella. In fact, Galileo also believed that physics, in particular astronomy, was structured like mathematics, that is, that it proceeded first with the resolutive and then with the compositive method, and that way was able to provide "necessary demonstrations", that is demonstrations endowed with necessity, not only from causes to effects, but also from effects to causes. Furthermore,

in logic he always considered himself, as we know, totally Aristotelian, referring to the Aristotelianism of his time, that is, above all, of Zabarella. The novelty that Galileo introduced in *regressus* were the experiments, the "sensible experiences", that is, the so-called experimental method, aimed at assuring the truth of the effects, which is the truth of the conclusions. However, he did not doubt that, once the truth of the conclusions was determined, they would be enough to guarantee the truth of the hypotheses from which they sprung, transforming them in unmitigated principles (...). As we know, Galilei claimed to have found the argument that proved in an absolute necessary way the truth of Copernican theory, and he pinpointed it in the phenomenon of tides, which he explained as a consequence of the earth's movement ([166], pp. 289–290).

In a word, Galilei pretended to follow Zabarella to justify the soundness of the Copernican heliocentric hypothesis by his astronomical observations through his telescope. The inconsistency of the Zabarella method applied to the Galilean epistemology of modern science led Leibniz—much less naïf in logic than the great Italian physicist—to give the only possible consistent interpretation of the "reversal of the arrows" in the extensional logic he used to justify the Aristotelian syllogism. In extensional logic, indeed, the reversal of the arrows premise-consequence is formally consistent in the *tautologies* of the double-implication (i.e., of the logical equivalence, ↔).

This logical elementary evidence led Leibniz to distinguish between the meaningless tautologies of the *a priori analytic judgements*, and the meaningful but contingent empirical *a posteriori synthetic judgements.* In turn, this distinction led Kant to define their synthesis by his notion of the *a priori synthetic judgments,* whose logical inconsistency was at last demonstrated by Quine, as we know [167]. All these "contortions" of the modern ontological conceptualism are an evident consequence of the abandonment of the modal non-extensional logic in the academy that started in the late XV cent. and endured until the actual recovery during XX and XXI centuries.

Effectively, indeed, from Gottfried Leibniz on, the causality relation was interpreted as the *sufficient condition* of the premise-conclusion deductive inference, therefore reducing the causal relation to the logical one, as Kant explicitly demonstrated, making "causality" (differently from Aristotle [39]) one of the logical (predicative) categories of his table in the *Critique of the Pure Reason*. Leibniz even made the "Sufficient Reason" or "Reason-Consequent Principle" together with the "Contradiction Principle" the two pillars of his *Théodicée* (Section 44) and his metaphysics (*Monadologie*, Sections 81–82).

What is worse for modern ontology and metaphysics is that this confusion between the causal and logical entailments led to the acritical generalized extension of the Leibnizian interpretation of "cause" as "sufficient reason" in a demonstrative procedure, also to Aristotle's and Aquinas' metaphysics [40]. In the Aristotelian logic and ontology, on the contrary, the causality is not a categorical (simple) predicate such as in Leibniz and Kant, but it is the result of the composition of three predicative categories of the Aristotelian table: *relation, action, and passion.* Therefore, when we speak about a "primary cause" in the Aristotelian cosmology, we are referring not only to the "heavenly spheres" (=acting principle) but necessarily also to the "primary matter" (=passive principle) of the heartily physics on which the heavenly bodies act.

Moreover, as we have seen before, in the *modal* causal entailment, the cause plays the role of the *necessary condition* (it is categorically on the left side of the reversed causal arrow ← "from the effect to the cause"), thus justifying, evidently, the resolution into some *primary cause* for closing "upwardly" the causal chain, either on the physical plan (Aristotle) or on the metaphysical plan (Aquinas) [30–32]. To say all this more synthetically with the words of Michael Heller in a paper where he proposed the CT logic as the proper logic of formal metaphysics and theology for the construction of an updated "theology of nature", what is lacking in Leibniz's logic and metaphysics is the notion of "categorical duality" [168]. Indeed, in the modal coalgebraic logic of CT, as we have seen, the reversal of the arrows of

the Aristotelian logic and ontology acquires its full intelligibility and soundness, i.e., its categorical, diagrammatic *universality*.

Finally, it is evident that this re-proposal of the categorical duality between the causal and logical entailments can suggest a solution not only to the *ontological* but also to the *epistemological* conundrum from Galilei to Popper concerning the justification of the soundness of the mathematical hypotheses in modeling physical processes and events, according to the hypothetical-deductive method of modern natural sciences, as I anticipated in the introduction of this paper.

**Funding:** This research received no external funding.

**Informed Consent Statement:** Not applicable.

**Data Availability Statement:** Not applicable.

**Conflicts of Interest:** The Author declares no conflict of interest.

## Abbreviations

| | |
|---|---|
| BA | Boolean Algebra |
| BAO | Boolean Algebra with Operators |
| CCR | Canonical Commutation Relation (in QM and QFT) |
| CDM | Cold Dark Matter (Model in GR) |
| CMB | Cosmic Microwave Background (Radiation) |
| CSM | Cosmological Standard Model (in GR) |
| CT | Category Theory |
| DDF | Doubling of the Degrees of Freedom |
| ESR | Epistemic Structural Realism |
| FOL | First Order Logic |
| GR | General Relativity Theory |
| ML | Modal Logic |
| NBG | Von Neumann-Bernays-Gödel (Set Theory of) |
| NG | Nambu-Goldstone (bosons) in QFT |
| NR | Natural Realism (formal ontology) |
| NWF | Non-well-founded sets (Theory of) |
| OSR | Ontic Structural Realism |
| PC | Predicate Calculus |
| QFT | Quantum Field Theory |
| QM | Quantum Mechanics |
| QV | Quantum Vacuum |
| RTBA | (Stone's) Representation Theorem for Boolean Algebras |
| SM | Standard Model (in QFT) |
| SQ | Second Quantization (interpretation of QFT) |
| SR | Special Relativity Theory |
| SSB | Spontaneous Symmetry Breaking (of the QV) |
| ST | Set Theory |
| TCS | Theoretical Computer Science |
| TOE | Theory Of Everything (in Fundamental Physics) |
| Z | Zermelo (Set Theory of) |
| ZF | Zermelo-Fraenkel (Set Theory of) |
| ZFC | ZF with Choice Axiom (Set Theory of) |

## Appendix A. The Categorical Operations of "Limits" and "Colimits"

Landsman, in his short but significant account about the relevance of the CT meta-language for formalizing the main notions of operator algebra in quantum physics and quantum logic (see [45], pp. 805–833), indicates in the categorical operation of *(co)limits* another fundamental contribution of clarification for mathematical and functional analysis in physics (see [45], p. 805). We give in this appendix some fundamental definitions of the

*limits* and *colimits* operations, mainly referring to a recent paper by Kairui Wang, aimed at using limits and colimits operationally [169], even though we also refer to more classical works such as [170], pp. 22–23, [171], pp. 16–18.

The first step is to introduce the notion of *comma category.* Effectively, this is a category whose objects are morphisms, given that "limits" and "colimits", interpreted in CT as "vertices" of "cones" and "cocones" of morphisms that share, respectively, the same "domain" or the same "codomain" of morphisms, are *specific types of comma categories* [41], as we see below.

To approach the notion of the comma category, it is fundamental to understand which problem we want to solve by introducing it. Wang thus explains the *diagrammatic universality* problem that the comma category solves in CT logic and mathematics.

> One type of problem in category theory is the universal mapping problem. Informally, these problems look for a morphism (called the "universal morphism") *that satisfies some desired property*, such that any other morphism satisfies the property "factors through" it in the sense that it is the same universal morphism composed of some other morphism ([169], p. 3).

From this characterization, the relationship between limits and colimits as comma categories clearly emerges, and our problem of a categorical formalization (diagrammatic universality) in formal ontology of the dynamic process of *factorizing through* (limits) or *constructing from* (colimits), respectively, but *universally* some properties/functions/predicates both in physics and logic is identified, as we have seen in the rest of this paper. What is highly significant for our aims is that this is true in formal ontology because it is true in logic and mathematics that *colimits* give a diagrammatic universalization to the fundamental operations of *direct limits* $X = \varinjlim X_i$ in mathematical and functional analysis, and then in physics and logic [42]. Therefore, we can refer to the definition of *comma category* as a "category whose objects are morphisms", following Wang himself ([169], p. 3):

**Definition A1.** *(Comma category). Let A, B and C be three categories, and $F : \mathcal{A} \to \mathcal{C}$ and $G : \mathcal{B} \to \mathcal{C}$ be two functors with the same target category (codomain), i.e., $\mathcal{A} \xrightarrow{F} \mathcal{C} \xleftarrow{G} \mathcal{B}$. The "comma category" $(F \downarrow G)$ has objects that are triples $(\alpha, \beta, f)$, where $\alpha$ is an object in A, $\beta$ is an object in B, and $f : F\alpha \to G\beta$ is a morphism in $\mathcal{C}$ that thus is constituted by morphisms relating objects belonging to the other two categories.*

This allows us to apply to our construction of comma category the notion of "final objects", that is, *initial* and *terminal objects* (see Definition 7.). For this, it is necessary to consider comma categories where one functor of the two considered has as its source the category **1** with only one object $(*)$ and its relative identity morphism $1_*$. This functor simply maps to an object *c* ä Ob(C) and therefore we call this functor *c*. We can then easily understand the notion of *universal morphism* as the initial (respectively, the terminal) object in each of the *dual* comma categories $(c \downarrow F)$ and $(F \downarrow c)$ ([169], pp. 4–5):

**Definition A2.** *(Universal morphism as the initial (terminal) object of a comma category). Let c be an object in a category C and let F: B → C be a functor. If we consider c as a functor from category **1** with one object $(*)$ and its identity morphism $1_*$ to C, we define the "universal morphism" (natural transformation) from c to F as the initial object in the comma category $(c \downarrow F)$, and dually, the "universal morphism" from F to c as the terminal object in the comma category $(F \downarrow c)$.*

To pass from this last definition of a comma category $\mathbf{1} \xrightarrow{c} \mathcal{C} \xleftarrow{F} \mathcal{B}$ to the definition of *colimits* as categorical universalization of the notion of *direct limits* —and dually of *limits* as categorical universalization of the notion of *inverse limit*—it is sufficient to recall that direct and inverse limits ultimately consist of a *family of sets indexed by a fixed set* or, equivalently, by a function from the indexing set to a class of sets (see note 42). In the CT generalization, a colimit (and a limit) interpreted as a diagram is therefore a collection of objects and

morphisms, labeled by a fixed "small" *category of indices* $j \in \mathcal{J}$; or, equivalently, it is a *functor* from a fixed small *index category* $\mathcal{J}$ to an arbitrary category [43].

Indeed, one can interpret the category of *J*-shaped diagrams in C as the *functor category* $\mathcal{C}^{\mathcal{J}}$. It means that, for each object *a* in C, a constant diagram exists: $\Delta_a : \mathcal{J} \to \mathcal{C}$, mapping every object *j* in $\mathcal{J}$ to *a* and every morphism in $\mathcal{J}$ to $1_a$. One can therefore define the *diagonal functor* $\Delta : \mathcal{C} \to \mathcal{C}^{\mathcal{J}}$ as the functor assigning the diagram (*constant functor*) $\Delta_a$ to each object *a* of C and $f : a \to b$ in C to each morphism the natural transformation $\Delta f = \Delta_a \to \Delta_b$ in $\mathcal{C}^{\mathcal{J}}$. Moreover, because $\Delta_a, \Delta_b$ are constant diagrams, this construction implies a correspondent natural transformation $\iota$ between the functors $\Delta$ and *F*, therefore involving the morphisms between objects, *i,j* in $\mathcal{J}$.

At this point, we have all the necessary components for understanding the definitions of colimits (and limits) in CT as a diagrammatic generalization of the notions of direct (and inverse) limits in mathematical and functional analysis (see [169], pp. 5–7).

**Definition A3.** *(Categorical notion of colimit). We can connote a functor F from a small category $\mathcal{J}$ to a category C, $F : \mathcal{J} \to \mathcal{C}$, as a diagram $\Delta$ over $\mathcal{J}$ in C. It is evident that F is an object in the functor category $\mathcal{C}^{\mathcal{J}}$ for the diagonal functor $\Delta : \mathcal{C} \to \mathcal{C}^{\mathcal{J}}$. The colimit of a diagram F as an object in C and denoted as Colim F is therefore the universal morphism from $\Delta$ to F.*

Specifically, Colim *F* is the initial object in the comma category $(F \downarrow \Delta)$. It is a natural transformation $\iota : F \to \Delta_{\mathrm{Colim}\ F}$, where Colim *F* is an object in C. Because $\Delta_{\mathrm{Colim}\ F}$ is a constant functor, the naturality of $\iota$ produces the following commutative diagram for every morphism $f : i \to j$ in $\mathcal{J}$:

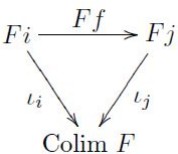

From this derives the further categorical universalization of Colimits as a *cocone of morphisms* (see note 41):

**Definition A4.** *(Categorical notion of a cocone of morphisms). A natural transformation from a diagram $F : \mathcal{J} \to \mathcal{C}$ to a constant functor $c \in Ob(\mathcal{C})$ is denoted as a "cocone" over F.*

**Remark A1.** *From the definition of cocones, the interpretation of the comma category $(F \downarrow \Delta)$ as the category of cocones over diagram F immediately derives. In it, the colimit of F is the initial object. This comma category is therefore denoted as $Cocone_F$.*

The universality problem solved by the comma category $Cocone_F$ is the following [169], p. 6.

Given a diagram $F : \mathcal{J} \to \mathcal{C}$, there is an object Colim *F* with associated morphisms from each $F_j$, where $j \in Ob(\mathcal{J})$ to Colim *F* such that Colim *F* and its associated morphisms $\iota_i$, $\iota_j$ commute with all morphisms *Ff*, where *f* is a morphism in $\mathcal{J}$. This construction is universal in that if an object *Y* and its associated morphisms $v_i$, $v_j$ from each $F_i$, $F_j$ to *Y* also commute with all morphisms *Ff*, then a unique (universal) morphism *h* exists: Colim $F \to Y$, such that for any $f : i \to j$ in $\mathcal{J}$, the following diagram commutes:

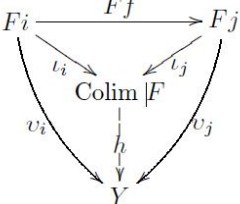

An analogue *dual* construction concerns the categorical operation of *limits* as the comma category $(\Delta \downarrow F)$ and its diagrammatic universalization as the **Cone**$_F$ category of the *inverse limit* operation for different categories of objects and morphisms (see Note 42).

Moreover, to immediately realize the relevance of these very abstract constructions, it is appropriate to recall here that, in the framework of the comma category construction, the notion of *diagonal functor* $\Delta$ is a generalization granting diagrammatic universality in CT to the *diagonal form* of a $n \times m$ matrix. It is thus a notion whose relevance is difficult to exaggerate in mathematics and physics!

Indeed, following the synthesis offered in [140], because all the *small* categories are categories defined not on *proper classes* with *Card* $\geq V$ (*large* categories) but on *sets* with *Card* $< V$, we can distinguish among *different classes of small categories* and then *their limits-colimits* according to the different cardinalities $< V$, $\aleph_0$, $\omega_1, \omega_2$, ... of their objects (sets). We can therefore have *locally finitely presentable categories* with $\aleph_0$-*directed colimits*, or *locally not-finitely $\omega_1$-presentable categories* with $\omega_1$-*directed colimits*, or *locally not-finitely $\omega_2$-presentable categories* with $\omega_2$-*directed colimits*, etc., where the attribute "local" means that the cardinality at issue *does not concern the category* but the objects in it. For instance, "**Set** is locally finitely presentable. In fact (i) every set is a directed colimit of the diagram of all of its finite subsets (ordered by inclusion), and (ii) there exists, up to isomorphism, only a (countable) set of finite sets" while the category of finite sets is not $\aleph_0$-countable and then it is not locally *finitely* presentable (see [140], p. 17). The categories **Pos, Grp, Aut, ...** are also locally finitely presentable while the categories **CPO** (*complete posets* in which every directed set has a join) and **Top** are not finitely presentable.

In Section 4.8, always following [140], we briefly introduced two examples of $\omega_1$-*presentable categories* characterized by $\omega_1$-*directed colimits* that are, respectively, either *locally* presentable (the $\omega_1$- *locally presentable* category **Ban** of Banach spaces) or *not-locally* presentable (the category **Hilb** of Hilbert spaces that is a full sub-category of **Ban** because **Hilb** is not "cocomplete" since its colimits are in **Ban** not in **Hilb**). To them, in the light of the precedent discussion, we added the subcategory of the doubled Hilbert spaces for the Bogoliubov functor, i.e., **Hilb** $\hookrightarrow$ **DHilb($\mathcal{B}$),** which splits into an infinite number of pairs of *finite* Hilbert spaces *pairs* $\mathcal{H}, \widetilde{\mathcal{H}}$, dually equivalent between them. Regarding this subcategory, we proposed the hypothesis, which needs to be further developed and rigorously mathematically justified for its novelty and relevance, that this subcategory of infinitely many doubled Hilbert spaces for the Bogoliubov functor **DHilb($\mathcal{B}$)** is *cocomplete* because its colimits are within it, and it is an $\aleph_0$-*locally finitely presentable* category, because the indexing small category $\mathcal{J}$ of its colimit construction is constituted by the denumerable sets $\{\mathcal{N}\}$ of NG-boson condensates, i.e., each finite subset—no matter how large it is—of the infinitely many pairs of the doubled Hilbert spaces is univocally indexed by a finite number of NG-bosons, i.e., it is $\aleph_0$-*countable*. Therefore, to understand the sense of our hypothesis, it is necessary to understand at least the statement of the fundamental *cocompleteness theorem* that the category **DHilb($\mathcal{B}$)** satisfies while the demonstration of this theorem—and of its dual *completeness* theorem—is outside the limit of the present paper and can be found in [169], pp. 7–9.

To understand the statement of the categorical cocompleteness theorem, only the definition of the categorical notion of the *coequalizer* for pairs of morphisms between coproducts is necessary, which applies in our case to each pair of dually equivalent doubled Hilbert spaces $\left(\mathcal{H}, \widetilde{\mathcal{H}}\right)$ as far as both are indexed by the same finite value of NG-boson condensates $|\mathcal{N}| := \left(\mathcal{N} \equiv \widetilde{\mathcal{N}}\right)$.

**Definition A5.** *(Categorical notion of coequalizer). Given two objects A, B in a category $\mathcal{C}$ with two morphisms f, g: A $\rightarrow$ B the "coequalizer" of f and g is an object denoted Coeq(f, g), and a morphism p : B $\rightarrow$ Coeq(f, g), such that fp = gp, and it is universal in that if Y is an object with morphism v : B $\rightarrow$ Y such that fv = gv, then a unique morphism h : Coeq(f, g) $\rightarrow$ Y exists such that the following diagram commutes:*

$$A \underset{g}{\overset{f}{\rightrightarrows}} B \overset{p}{\to} \mathrm{Coeq}(f,g)$$
$$\searrow_{v} \quad \downarrow h$$
$$Y$$

It is not difficult to see that $Coeq(f,g)$ is nothing but the colimit of *F*, that is, a diagram *F* in the precedent definition of the comma category of $Cocone_F$.

The notion of *coequalizer* allows us to understand the notion of *cocomplete category,* that is, a category where colimits over diagrams with a small source category *exist.* Specifically, it satisfies the cocompleteness theorem (see [169], p. 9):

**Theorem A1.** *(Cocompleteness Theorem). A category $\mathcal{C}$ is cocomplete if the coproduct of any set of objects in $\mathcal{C}$ exists and the coequalizer between any two morphisms with the same source and target exists.*

The statement of the theorem (and much more of its proof in [169], p. 9) emphasizes, in our application of the theorem to the category **DHilb($\mathcal{B}$),** its strict formal relationship with the category of coproducts **qHCoalg($\mathcal{B}$)** that we justified from a physical standpoint in Sections 4.6 and 4.8. Additionally, this indirectly supports the consistency of our hypothesis. To understand the physical and mathematical relevance of this result, see [71], in which it was recently demonstrated that the internal coalgebras of a locally presentable cocomplete category satisfy the *compactness* of the topological space on which they are defined because they contain all their limit points (colimits). This is not an irrelevant result in our case, given that by the **DHilb($\mathcal{B}$)** category, we model the *phase transitions* among quantum fields in non-linear (even chaotic) systems in far-from-equilibrium conditions.

Finally, the discussion in this Appendix and this whole paper reinforces, in a proper sense, what V. Pratt and D. Pavlović said about the relevance of the category of coalgebras for mathematical analysis and the same notion of continuum in mathematics as far as it is coinductively defined and, consequently, for logic in its duality with the category of algebras (see [72]):

> It is reasonable to ask why the continuum should be defined coinductively rather than inductively. It seems to us that the coinductive nature of the continuum is a consequence of our computing not with the reals themselves but inductively with rationals as approximations to reals. The computational status of the reals is not as *elements* but as *predicates* on the rationals, specifically the Dedekind cuts in the rationals. But elements and predicates are dual notions: whereas elements of a set *X* can be understood as functions *to X*, predicates on *X* are functions *from X* ([72], p. 119).

Effectively, this synthetic statement summarizes what I said in this paper, vindicating the ante-predicative nature of the CT logical analysis as far as it does not consider the membership relation as a primitive such as ST. Indeed, what is the notion of colimit as a cocone of morphisms just illustrated if not a sort of "structural" *relational anatomy* of the constitution of a class of elements as the "domain" of a given predicate?

**Appendix B. The Core of Stone's Representation Theorem for Boolean Algebras**

In the light of what we have already said in Sections 5.1.1 and 5.1.2 and pictorially represented in Figure 9, we know that every finite BA can be represented as the power-set of its set of atoms. Each element of BA corresponds to the set if the atoms below it, that is, the join of which is an element. As we know, this power-set representation can be constructed in general for any *complete* atomic *BA.*

The core of *Stone representation* is that we can generalize the precedent construction for *non-complete and atomic* BAs using fields of sets instead of power-sets (see Section 5.1).

Indeed, the atoms of a finite BA correspond to its ultrafilters and an atom is below a given element of BA if and only if it is included in the ultrafilter corresponding to that atom. This leads to the possibility of constructing a representation of BA by taking the set of its ultrafilters and defining complexes by associating each element of BA with the set of ultrafilters containing that element.

Equivalently, as we know, we can consider the set of *homomorphisms* onto the two-element BA and constructing complexes (and then fields of sets) by associating each element of the BA with the set of such homomorphisms mapping it to the top element. All this leads us to the so-called *Stone representation of BA as a certain field of sets,* that is, the fields of sets just illustrated, satisfying some topological requirements we discuss immediately as follows.

Indeed, the final step in understanding Stone's RTBA stating the *duality* between a given BA *A* and its corresponding *Boolean topological space* or, more synthetically, its *Stone space S(A)*, consists of extending the previous construction to *topological* fields of sets. In fact, we can consider the complexes of a field of a set representing BA as a *basis* for generating a given topology $\mathcal{T}$. Indeed, in mathematics, the basis of a topology $\mathcal{T}$ for a *topological space* $(X, \mathcal{T})$ is a family *B* of open subsets of *X* such that every open set of the topology can be considered as the *union* of some members of *B* (see [172], p. 30). Now, the fields of sets, the complexes of which can be the basis of a Stone topological space associated (isomorphic) with a given BA *B*, i.e., *S(B)*, must satisfy the following two conditions (see [173], pp. 69–76):

1. The field of sets must be *separative,* i.e., for every pair of distinct points, there is a complex containing one point and not the other.
2. The field of set must be *compact*, i.e., for every proper filter over *X*, the intersection of all the complexes contained in the filter is non-empty.

Therefore, by denoting with $\mathbf{X} = (X, \mathcal{F})$ the fields of sets whose complexes form a basis for a topology, and with $T(\mathbf{X})$ the corresponding topological space $(X, \mathcal{T})$, whose topology $\mathcal{T}$ is formed by taking arbitrary unions (and intersections) of complexes, then:

1. $T(\mathbf{X})$ is always a *zero-dimensional space* (i.e., graphically representable as a "point").
2. $T(\mathbf{X})$ is a *Hausdorff space* (i.e., whose points have disjoint neighborhoods) if $\mathbf{X}$ is separative.
3. $T(\mathbf{X})$ is a *compact space* with compact open sets $\mathcal{F}$ if $\mathbf{X}$ is compact.
4. $T(\mathbf{X})$ is a *Boolean space* or a *Stone space* with clopen sets $\mathcal{F}$ if $\mathbf{X}$ is both separative and compact.

From this reconstruction, the statement that a Stone space *S* is a *compact, Hausdorff, totally disconnected* [44] topological space becomes more understandable for philosophers. A topological space that has the same properties of the topological spaces on which the C*-subalgebras of Hilbert spaces in quantum physics formalism are defined, so that we can say that Stone RTBA is at the basis of any architecture of topological quantum computers, as we know.

In the light of what we said in this Appendix B and Section 5.1.1, Stone's RTBA stating the *isomorphism* and then the *Stone duality* between a BA *B* and a topological field of sets and the functorial dual equivalence between the correspondent categories also becomes easily understandable. Specifically, this is a field of sets constituting the basis of the *Stone space* associated with BA *B*, i.e., *S(B)*. The points in *S(B)* are indeed the ultrafilters on *B* or, equivalently, the *homomorphisms* from *B* to the two-element BA. Conversely, given any topological space $(X, \mathcal{T})$, its subsets that are clopen form BA.

## Notes

1    Following W. V. O. Quine's reconstruction ([29], pp. 133–136), according to this principle formulated by Russell to justify his solution of Cantor's and Frege's antinomies by a "ramified type theory", each object in logic, either individual or collection (set, class) can exist only as a *member* or *element* of the domain of a given predicate (i.e., of an "higher type" class), and finally, because it is so satisfying a *self-identity* relation, as a member of the *universal class V*, intended as the domain of the meta-predicate "being true". As Quine synthesizes, "*V* (stays) for '$\hat{x}(x = x)'$ *V* is, by definition, the class of all those elements which are self-identical, i.e., since everything is self-identical (...), *V* is simply the class of all elements" ([174], p. 144). It is worth emphasizing that all this is equivalent to affirm that in standard ST no set *self-membership* is allowed, i.e., no set can be an element of itself. A condition that in ZF, for example, is granted by the "foundation" and the "pairing" axioms, which, because of the consequent Zermelo's "well-ordering theorem", at the same time grant that every set has an "ordinal rank", according to Von Neumann's "ordinal cumulative hierarchy" construction (see [66] for a synthesis).

2    For instance, from the truth of the propositions: "Julius Caesar wrote the *De Bello Gallico*" and "Julius Caesar fought in Gallia", by applying the connective "and", we can deduce the truth of the composed proposition "Julius Caesar wrote the *De Bello Gallico* and fought in Gallia". On the contrary, we cannot deduce the truth of the composed proposition "Julius Cesar wrote the De Bello Gallico while he was fighting in Gallia", typical of the *tense-logic* that is one of the possible *alethic* interpretations—historically the first one since Aristotle—of the MC. For the truthfulness of tense-logic propositions it is necessary, indeed, in ML to consider the relationships between the *present* or "actual" state of the world, and other *past* and/or *future* "possible" states of the world.

3    The distinction between "optimality" and "maximality" conditions for the ethical constraints —where "optimal" stays for "good in all the possible worlds", and then for *all* the human groups/cultures, and "maximal" stays for "good in some possible worlds", and then for *some* human groups/cultures—where introduced in the contemporary debate by the 1998 Nobel Prize in Economy Amartya Sen. This distinction is the core of his theory of the *Comparative Distributive Justice,* based on the notion of *equity* (*fairness*), instead of the abstract (and false) "equality" in social sciences, of which he proposed also a *formal version* [162]. This was done in the framework of the newborn discipline of the *social choice theory*, he contributed to create, and to which he significantly dedicated his Nobel Lecture [175]. Today the "social choice theory" is a branch of the *formal philosophy*, the branch concerning the "decision theory and social philosophy" (see [176], pp. 611–725).

4    Effectively, in a formalized deontic logic in our global society and economy, the *optimal choice* (absolute) criterion must be substituted by a more effective and fair *maximal choice* criterion for *different social/economical situations* and *value systems,* relative to different groups in the society, according to a *comparative theory of distributive justice as fairness,* having in the *personal flourishing* of human individuals and group the "common good" to be pursued. This social theory was developed by the Nobel Prize in Economics Amartya Sen into the so-called *social choice theory*, conceived as a formal version of the political and social philosophy [162].

5    In the case of "balanced" open systems, the summands of the coproducts cannot commute with each other because representing the system and the thermal bath energy contributions in the calculation of the total energy of the quantum state.

6    Effectively, a difference occurs between them. The coproducts of the Bogoliubov construction in QFT for dissipative systems are *non-commutative*, so that the corresponding Boolean algebras are *non-commutative or skew Boolean algebras* that satisfy the same axioms of the general Boolean algebras except for the commutativity between the $\wedge$ and $\vee$ operators (see [160] for an extended examination).

7    For a connection with the actual use of "formal ontology" in computer science, it is sufficient to recall that "transcendental subject" in philosophy does *never* refer to a human *individual*, but to the common way of thinking and believing shared by a group of individuals, in the limit, by all the human individuals, as *conscious*—and then *intentional*—agents.

8    It is significant that for the conceptualist ontology there is *per se* no representative in the Ancient and Middle Ages, because it is typical of the Modern Age. It starts indeed with Descarte's foundation of the logical truth on the mental *evidence* and then on *consciousness,* and not on the "conformity" (homomorphism) of the structures of language with the structures of reality like in the Middle Age Platonic (logicism) and Aristotelian (naturalism) philosophies. Descarte's and modern conceptualist positions can therefore be synthesized with the slogan: "a statement is true because it is evident, and not it is evident because it is true", as it is in the logical and natural realisms.

9    Please, note that because $\in$ is not a primitive in CT, objects for existing in CT must not satisfy a self-identity relationship and then their membership to *V* like in ST, where they must satisfy Russell's set-elementhood principle for being consistently defined/demonstrated as existing in the theory (see Note 1).

10   For understanding immediately, the relevance of an arrow-theoretic way of thinking as to the set-theoretic one, let us think at the oldest proof method in the Western logic, which is the Aristotelian deductive (categorical) syllogism in its more fundamental form, the so-called *In Barbara* form. For instance: "If all humans (B) are mortal (A), and all the Greeks (C) are humans, then all the Greeks are mortal". Now, in the *extensional* interpretation that Leibniz (followed by Euler and Venn) gave of the Aristotelian scheme: "*AB* & *BC.·. AC*", this corresponds to stating predicatively: $((B \in A) \wedge (C \in B)) \rightarrow (C \in A)$. Such a predicative formula has according to Leibniz its extensional proof in the transitive inclusions of the respective classes $((\mathbf{B} \subseteq \mathbf{A}) \wedge (\mathbf{C} \subseteq \mathbf{B})) \rightarrow (\mathbf{C} \subseteq \mathbf{A})$. In CT where the set-elementhood is not a primitive, the universality of this demonstration takes the form of the commutative triangular diagram, *ABC* we discussed before, whose objects are *categories*—which in the Aristotelian syllogisms are always

"natural kinds"—and the morphisms are *functors* (see Definition 3. and [177] for this functorial interpretation of the syllogism "triadic" structure). Significantly, this categorical formalization of the syllogism can justify also the Aristotelian non-extensional (modal) syllogisms that, on the contrary, Leibniz's extensional interpretation cannot do, as J. Łukasiewicz first noticed [164].

11 On this regard, it is significant the fundamental work of R. Maddux [178] who demonstrated the strict relationship of Peirce's naïve triadic algebra of relations [179] with its axiomatic development into a calculus of relations by Tarski [69]. Not casually, indeed, the last book published by A. Tarski with S. Givant [180] (see also [181]) concerns precisely the demonstration of two fundamental results. (1) Before all, the demonstration that an *irreducible triadic algebra of relations* is sufficient for expressing faithfully any *first-order logic* (FOL) formula up to logical equivalence. That is, any FOL formula of the predicate and propositional calculi can be expressed faithfully in an *equation logic* (having arithmetic operators as connectives and numbers as their arguments) on a triadic basis. This fragment of FOL and the corresponding *variety* of relation algebras (**RA**)—i.e., the class of relation algebras defined by purely equational postulates—are therefore sufficient for expressing not only the *Peano arithmetic,* but also practically all *axiomatic set theories* ever proposed. Secondly, (2) just because of this expressive power, **RA** suffers in logic the same limitations imposed by Gödel's incompleteness theorems. I.e., the logic based on **RA** is incomplete and undecidable. However, and this is the second fundamental result, the Boolean FOL fragment of **RA** results to be *complete* and *decidable,* since its semantics is defined over *partially ordered* sets. All this means that we can express algebraically almost all mathematics in terms of a triadic **RA,** and more significantly we can express FOL without using quantifiers ($\forall, \exists$), connectives ($\wedge, \vee$), and turnstiles ($\vdash, \Vdash$), but essentially the *equation logic* of a Boolean algebra. If all this explains the odd title of Tarski's and Givant's book "A formalization of set theory without variables" [180], this algebraic construction of logic and mathematics is completed by the possibility in CT "arrow-theoretic" logic of demonstrating the natural number construction by *primitive recursion* without any (impredicative) reference to numbers as predicative numerals like in **ZF** (see [182–184] and [68], p. 285).

12 Effectively, the *present-time* event in the causal light-cone (see Figure 1) can be categorically interpreted as the *final* object *F* sharing with all the other events *A* belonging to its past/future light-cones a *dual* causal morphism in the sense of Definition 7. Indeed, in the case of the set of events $\{A\}$ belonging to the *past* light-cone, *F* plays the role of an initial object *I* defined by the unique morphism $\iota_A$ pointing to the set $\{A\}$ of its causes, i.e., $F = I := \iota_A : I \to A$. In the case of the set of events $\{A\}$ belonging to the *future* light-cone, *F* plays the role of a terminal object *T* defined by the unique morphism $\tau_A$ since the set $\{A\}$ of its effects are pointing to *F* as to their shared cause, i.e., $F = T := \tau_A : A \to T$.

13 Effectively, as Rieger rightly recalls, the foundation axiom is not *per se* necessary for avoiding Cantor's antinomies in the transfinite induction construction (see also [76]). The other axioms of **ZF,** first the "separation" and the "power-set" axioms, are sufficient for avoiding them. Effectively, the foundation axiom was introduced by Zermelo essentially for granting his well-ordering theorem.

14 All Aquinas' works are here quoted using their Latin title, according to the online edition of Aquinas' *Collected Works* in [86]. The translations into English of the different passages are mine. Effectively, in Aquinas' ontology the *physical* objects in nature or "substances", either "individuals" ("*primary* substances") or "species" ("*secondary* substances", for existing must satisfy a simple *reflexive/identity* relation (*reditio ad semetipsum,* "return onto itself") like objects in CT, and not a *double-reflexive/self-identity* relation like objects in ST. The self-identity relation, indeed, for Aquinas characterized the *logical* objects in mind, as far as they are *abstract* objects.

15 $\exists x P x$ stays in standard set-theoretic predicate logic for $x \in \mathbf{P}$, where **P** is the class connoted by the predicate *P* denoting the identity $\mathrm{Id}_x$ shared by all the elements *x* of the class, and where, therefore, the class **P** must be of a higher ordinal rank with respect to its elements *x*, i.e., belonging to the domain of the predicate *P,* if we must avoid the "Russell antinomy" in Frege's theory of classes (see Note 1).

16 Roughly speaking, this means that the state (classical mechanics) and/or the phase (statistical mechanics) space representing the system dynamics is *invariant* by exchanging each other the two canonical variables onto the orthogonal axes of their graphic vectorial representation.

17 Effectively it is a sort of "resonance" or "constructive interference" among statistical wave functions "oscillating coherently" with the same phase, as the famous "double-slit" experiment exemplifies very well also for the wave functions of only one particle (pure states).

18 In physics an "observable" is a physical magnitude that we can measure, for instance, the position and the momentum. In Classical Mechanics, an observable is a real-valued function on the set of all possible states. In quantum physics (QM and QFT), it is an "operator" because the properties of the quantum state, that is, the probability distributions for the outcomes of any possible measurement performed over it, can be determined only by some sequence of operations, e.g., by submitting the systems to the action of several electromagnetic fields, and then reading the resulting different values.

19 In functional analysis, a C*-algebra is a *Banach algebra*—that is, an associative algebra over the fields of real or complex numbers that is also a "Banach space", i.e., a space with a defined norm $||\cdot||$ complete in the metrics induced by the norm – together with an involution (a reversal of the morphisms like between $f(x)$ and $f^{-1}(x)$) satisfying the property of *adjointness*. In the specific case of quantum formalism, it is an *algebra B* over the complex number field $\mathbb{C}$ of *continuous linear operators* on a *complex Hilbert space*. In this case, the adjointness condition is strictly related to the Hermitian one (denoted by the symbol *), of the "inner products" $\langle \cdot, \cdot \rangle$ characterizing generally the Hilbert spaces. That is, without going deeper in the technicalities, given a linear operator $A : H_1 \to H_2$ between Hilbert spaces, the *adjoint* (dual) operator $A^* : H_2 \to H_1$ fulfills the condition between the relative inner

products: $\langle Ah_1, h_2 \rangle_{H_2} = \langle h_1, A^*h_2 \rangle_{H_2}$. In the case that the Hilbert spaces concerned are identical, $A$ is an endomorphism on the same Hilbert space satisfying therefore a *self-adjointness* property and then the duality between a Hilbert space $\mathcal{H}$ and its operator space $\mathcal{H}^*$. For this reason, Hilbert spaces are *self-dual*. In the case of C*-algebras, this adjointness condition is extended to the operators acting on Banach spaces $A : D \to E$, with corresponding norms $||\cdot||_D, ||\cdot||_E$. Its adjoint operator is $A^* : E^* \to D^*$. In this case, the Banach algebra $B$ satisfies two other properties: (1) it is *topologically closed* in the norm topology of the operators; (2) it is closed under the operation of taking adjoints of the *operators*. In the CT formalization, this means that the category of Hilbert spaces **Hilb** is effectively a *full subcategory* of the category of Banach spaces **Ban**, and then that **Hilb** is not *cocomplete* because all its colimits are in **Ban** not in **Hilb** (see [140] and below Section 4.8). Finally, one can extend the C*-algebra construction also to non-Hilbert C*-algebras. This class includes the algebras of the *continuous functions* $C_0(X)$, i.e., vanishing in the infinite limit. This justifies Landsman's reading of all classical and quantum physics in this framework of the algebra of operators, *per se* born, as we have seen, in the framework of quantum physics formalization.

[20]  The algebraic *tensor product* $V \otimes W$ between two vector spaces $V$ and $W$ over the same numerical field, is itself a vector space, endowed with the operation of *bilinear composition* denoted by $\otimes$ from ordered pairs in the Cartesian product $V \times W$ to $V \otimes W$, so to generalize to tensors the matrix *outer product*. Where: the "bilinear map" is a function combining elements of two vector spaces to yield elements of a third vector space, and it is linear in both of its arguments, while the "outer product" of two vectors of dimensions $n$ and $m$ is a $n \times m$ matrix. In the case of two tensors, the outer product is another tensor. A fundamental property of tensor products between finite dimensional vector spaces is that the resulting vector space has dimensions equal to the product of the dimensions of the two factors: $\dim(V \otimes W) = \dim V \times \dim W$. This distinguishes the tensor product from the *direct sum* vector space, whose dimension is the sum of the dimensions of the two summands: $\dim(V \otimes W) = \dim V + \dim W$. Just as – for giving another example of the direct sum operation in algebra well known by everybody –, the direct sum $\mathbb{R} \otimes \mathbb{R}$—where $\mathbb{R}$ is a coordinate space defined on real numbers—is the bidimensional *Cartesian plane.*

[21]  Effectively, the "quantum entanglement" acquires, in the light of the long-range correlations among quantum fields in QFT related to the Goldstone Theorem, an immediate intelligibility, showing that it does not imply any absurd "causal interaction" among quantum particles violating $c$ (the light velocity). That is, a physical signal propagating at a superluminal velocity, which is the deep reason for which Einstein refused the quantum "non-locality" (entanglement) in his famous discussion with Niels Bohr, during the 30's of the last century. For showing this, it is sufficient to recall the notion of "phase velocity" $V_P$ in the vacuum of SR that holds also in QFT. Now, $V_P = \frac{E}{p}$, where $P$ is the field phase, $E$ is the total energy, and $p$ is the momentum of a given physical signal. Therefore, in SR, $V_P = \frac{E}{p} = \frac{\gamma mc^2}{\gamma mv} = \frac{c^2}{v}$, where $\gamma$ is the Lorenz constant, $m$ is the mass, and $v$ is the velocity of the physical signal that is always less or also much less than $c$. This means that the phase propagation (or the propagation of correlation waves among quantum fields) in microphysics (QFT) is practically instantaneous without violating $c$.

[22]  Think at the everyday experience of the boiling water, exemplifying the continuously changing correlation-length among the water molecules, and then the continuously changing "dynamic boundary" of the vapor-liquid phase transition of water.

[23]  The dynamic mechanism according to which the water molecules, beyond a given density threshold, can condense into coherence domains (CDs) among their electric dipoles fields is today well known (see [120,185] for a more recent synthesis with several bibliographic references). The core of such a mechanism is that in each water CD the molecules oscillate *coherently* between two configurations of their electronic clouds, so to produce an electromagnetic field oscillating with the same frequency. The water CD can, therefore, attract by resonance a small number of "guests" molecules different from water, which share thus the energy stored in the CD. In this way, we have a much more efficient way than the random "diffusion process" introduced by the last work of A. M. Turing as the fundamental method of *morphogenesis* in biological matter [186], to make possible that *selective chemical reactions occur,* given that the chemical forces propagate only at short distances. For instance, this is the dynamic core of "cell specialization" in *epigenetics,* where only some sequences of the DNA that is the same for all the cells of a given organisms are activated/de-activated, because of the presence/absence of the proper molecules in the cell environment. In short, "the interplay between chemistry and electromagnetic field produces a collective oscillation of all the CDs that, according to the general theorem of quantum electro-dynamic coherence, gives rise to an extended coherence, where the CDs of water and "guest" molecules become the components of much more extended 'super-domains' which could just be the various organs" ([120], p. 37), at different level of the biological matter self-organization. Another well-studied phenomenon strictly related to the dipole CDs is the formation, propagation, and the reciprocal synchronization of *solitons*, that is a self-reinforcing solitary wave (a wave packet or pulse) that maintains its shape while it travels at constant speed. Solitons are caused by a cancellation of the nonlinear and dispersive effects in the medium. In macroscopic fluid dynamics, the formation of a "tsunami-wave" in the sea is a terrible example of "sea water soliton"! In biological matter electro-dynamics, the soliton presence is well established both in DNA and in protein dynamics, displaying a fundamental role for the efficiency of the cell metabolism through the cell microtubules, whose relevance for a quantum foundation of biology is today well recognized [120].

[24]  This notion, as Freeman and Vitiello explain elsewhere [40], is a critical reference to the much more famous theory of the "mirror neurons" by Giacomo Rizzolatti and his group [187]. The criticism consists in the fact that—apart from the fact that Rizzolatti's mirror neurons are limited to the brain interaction with the social environment—the measurements concerning the mirror system in the ape and in the human brains concern essentially the *passive* answer of neuron arrays of the motor neurons of one animal to stimulations deriving by the motor neurons of another animal, without explaining the underlying *dynamic mechanism* that, on

the contrary can have an elegant explanation at the fundamental physical level in the DDF principle of QFT as a result of the system-environment entanglement.

25   We recall that, following Von Neumann's construction of "cumulative hierarchy of ordinal number ranks" for justifying consistently the "transfinite induction" for infinite sets in **ZF**, $\omega_1$ is the limit ordinal number of the set of transfinite numbers with cardinality immediately successive to $\aleph_0$, i.e., to the "cardinality of the denumerable sets", that is, of all the infinite sets with the cardinality of $\mathbb{N}$.

26   We recall that a *Banach space* is a *Hilbert space if* it satisfies the "parallelogram law" characterizing the "inner structure" (inner product) of a Hilbert space, and for which it is self-dual.

27   In fact, *each* Hilbert space, precisely for its self-dual character, is *complete*, i.e., it contains all the *limits* necessary and sufficient for its computations in functional analysis. However, the category **Hilb** it is not *cocomplete* (i.e., it does not satisfy the fundamental "Cocompleteness theorem" (see Theorem A1 and the relative comments in Appendix A), because the category does not contain in itself the colimits for indexing the infinitely many inequivalent Hilbert spaces (see below).

28   For the reader convenience, I recall the statements of the two De Morgan laws of propositional logic: $(\neg(P \vee Q) \Leftrightarrow (\neg P \wedge \neg Q))$; and $(\neg(P \wedge Q) \Leftrightarrow (\neg P \vee \neg Q))$.

29   In parenthesis, is highly significant that complete BAs are a necessary ingredient for constructing Boolean-valued models of set theory using P. Cohen's *forcing* notion [188].

30   The connection of $\sigma$-algebras and measurable spaces in (statistical and quantum) physics is much more evident when we recall that any "probability space" in statistics is a probability triple $(\Omega, \mathcal{F}, P)$, where $\Omega$ is the set of all possible outcomes; $\mathcal{F}$ is an event space, which is a set of events $\mathcal{F}$, an event being a set of outcomes in the sample space; $P$ is a probability function, which assigns each event in the event space a probability that is a number between 0 and 1. Now, generally $\mathcal{F}$ is a $\sigma$-algebra, $\mathcal{F}$ consisting in the collection of all the events we would like to consider according to the type of the statistical analysis we want to perform on a given probability space. More generally, a $\sigma$-algebra or $\sigma$-field on a set $X$ is a collection $\Sigma$ of subsets of $X$, closed under complement, under countable unions, and under countable intersections so to constitute a *measurable space.*

31   We recall that a "set preorder" means that sets satisfy "reflexive" and "transitive" order relations $\leq$. So that, a set preorder is a "set partial order" if sets satisfy also "antisymmetric" order relations, while a set preorder is a "set equivalence class" if they satisfy also "symmetric" order relations.

32   This terminology originates with G. Fobenius in 1880s, who refers to a collection of elements of a group as a "complex".

33   Effectively, Stone in the demonstration of his theorem, does not use the Choice Axiom like ZFC but the Zorn Lemma that is an equivalent of this axiom for this type of application.

34   Generally, in TCS "concurrent computations" stay for two computational processes developing themselves *in parallel*—during overlapping time periods—instead of *sequentially*, with one completing itself before the other, and where the final state of the former gives the initial state to the latter according to a "circular" overall procedure.

35   For this historical reconstruction I am referring mainly to the final part of [32].

36   To Aquinas, indeed, is also attributed a short treatise about the modal propositional logic *De Propositionibus Modalibus* (available online in [86]), in which he offers an original interpretation of the *de re* and the *de dicto* modalities. Indeed, the propositional logic, unknown to Aristotle, given that it was defined and developed by his disciples, the Stoics, was well known in the Latin Scholasticism, because of the logical teaching of Anicius Manlius Severinus Boëthius (shortly, "Boethius": 477–524 A.D.). He not only translated into Latin Porphyry's *Isagoge* but wrote two treatises about the categorical (Aristotelian) syllogism and the hypothetical (Stoic) syllogism, by which the two logical calculi—the predicate and the propositional calculus, respectively—were introduced separately into the Medieval and then into the Modern logic, till their unification in Frege's formal calculus of classes.

37   Where, *by nature* means: those things that happen "or always or frequently" and then "not randomly" (*a casu*), i.e., by deterministic or statistical physical processes, as Aquinas explained before (see [86], *In Libros Physicorum,* II, lect. 13, n. 2).

38   That Aquinas is here referring to the notion of local truth (with relativized quantifiers) is evident from the following two quotations always from the *De Veritate* ("About Truth") book: «If therefore we take truth in the proper way, according to which the things are said true secondarily (i.e., relatively to an intellect), there are of several true objects (*plurium verorum*) several truths, and of a true object many truths in different intellects» (see [86], *Quaestiones Disputatae De Veritate,* I, 4 co.). «On the other hand, the truth that is in the human intellect is not related to things like an extrinsic and common measure to measured things [against Sophists, evidently], but like a measured to a measuring, ( . . . ) and therefore it must vary according to the variety of things» (see [86], *Quaestiones Disputatae De Veritate,* I, 4, ad 2).

39   For Aristotle, indeed, the predicate "being cause of" is not a category because resulting from the composition of three categories: "relation", "action", "passion" in his Table of Categories.

40   Particularly, it was the Scottish philosopher and logician Sir William Hamilton (1788–1856) who applied systematically, in his monumental work published posthumous *Lectures on Metaphysics and Logic*, the principle of sufficient reason for the construction of the whole metaphysical building, using a logic refusing explicitly, not only any symbolism, but also any *modality,* either real (*de re*) or logical (*de dicto*).

[41]　　The terms "cone" and "cocone" helps to understand intuitively the arrow-theoretic notions of "limits" and "colimits" as "terminal" and "initial" objects, respectively. Indeed, if we take a cone, its vertex with respect to its basis can be connoted, either as the unique "terminal object" (common target/codomain) of all the arrows having in each point of the basis their own sources/domains (i.e., "cones of morphisms"), or *dually* as the unique "initial object" (source/domain) of all the arrows having in each point of the basis their own targets/domains (i.e., "cocones of morphisms").

[42]　　We recall that generally in mathematics a *direct limit* is a way for constructing a *larger* object from many *smaller* objects "put together" in a specific way, generally by referring to an higher rank "class" of objects in the predicative set-theoretic approach in logic and mathematics. In CT these objects are from any category (e.g., **Set, Grp, Vect**$_k$**, Top,** . . . ) and the way for putting together the smaller objects is specified by the *homomorphisms* (or more generally, the morphisms) typical of the category concerned. For instance, in the case of sets, let $\{A_i : i \in I\}$ be a family of sets indexed by $I$, and $f_{ij} : A_i \to A_j$ be a homomorphism for all $i \le j$. Then, the pair $A_i, f_{ij}$ is called a *direct system* over $I$, the *direct limit* of the direct system is denoted by $\varinjlim A_i$, and its underlying set is constituted by the disjoint union (coproduct) of $A_i$'s, "modulo" a given equivalence relation $\sim$. I.e., $\varinjlim A_i = \cup_i A_i / \sim$. That is, if $x_i \in A_i$ and $x_j \in A_j$, then $x_i \sim x_j$ if there is some $k \in I$ with $i, j \le k$ such that $f_{ik}(x_i) = f_{jk}(x_j)$. From this definition, we derive the other definition of *canonical function* in terms of the homomorphism $\varphi_i : A_i \to \varinjlim A_i$ sending each element to its equivalence class. Dually, we can define the notion of *inverse limit* $\varprojlim A_i$ affirming that an element is equivalent to all its images under the maps of the direct system, i.e., $x_i \sim f_{ij}(x_i)$ for all $i \le j$. The duality between direct and inverse limits can be expressed as the following relation: $\mathrm{Hom}\left(\varinjlim X_i, Y\right) = \varprojlim \mathrm{Hom}(X_i, Y)$.

[43]　　We recall that a *small* category is a category whose objects are *sets* with *Card* < V, while a *large* category is a category whose objects are (Von Neumann's) *proper classes* with *Card* = V or even larger if we accept Gödel's generalized CH in **NBG**.

[44]　　In a zero-dimensional topological space, indeed, only one-point sets (i.e., the empty set and the unitary sets) are connected.

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
