# Peer review of "The Philosophy of Nature of the Natural Realism. The Operator Algebra from Physics to Logic"

_philosophies, doi:10.3390/philosophies7060121_

Round 1

Reviewer 1 Report

GENERAL COMMENT

This paper aims at the very ambitious project of providing grounds for formal philosophy in category theoretic terms. It is nonetheless poorly developed, promising too much and delivering too little. Instead of developing compelling reasons for category theory in this application, it simply states many definitions an results easily found in the literature without properly articulating with their goals.

The paper has many imprecisions and it needs a substantial review. It is far too long, spending time on historical issues already well developed in the standard literature. These topics, though important and interesting, besides being poorly developed, add little value to the arguments the paper propose to develop.

SPECIFIC COMMENTS

Abstract: It simply uses excerpts from the introductions without any additional value.

60 - 64: full paragraph in one long hard to sentence.

65 - 85: long paragraph; relevance to the thesis defended is not clear; many historical claims without proper justification.

86 - 100: this is full of jargon. I can imagine some of what is meant in this paragraph, but it is very hard to really get the point. The paragraph do not serves the purpose of introducing the subject.

101: typo "This main" --> "The main"

101 - 104: what is the main thesis? Instead of saying it, you say it will be said later. This does not help the reader.

131: I can see that the slogan states that Goldblatt prefers CT rather than ST, but it apparently does not have any meaning other than that. (Also, you should put the reference for this quote)

131  - 135: You say "Russell does this..., CT can do this...", but how this connects to any advantage is not written.

140 - 143: ok, this is something I want to see the argument for.

178 - 179: I don't think it adds to the text making this reference to Stoic and Aristotelian logic.

196: I don't understand what you want to say with 'operation' here.

196 - 197: I still don't understand the connection you are making here. I should by now understand why you require this ante-predicative character; and also aren't there other options, as, for instance, with modal theories?

202: I don't think we can associate Frege's intentions with linguistic turn; if anything can be said in this regard is that he is one of the basis of the linguistic turn.

221 - 224: very hard to understand. (moreover, the comment about Hilbert is not necessary and makes it harder to understand the phrasing)

236 - 258: I don't see the point of this long explanation. If for a beginner, it is too confusing; if for an expert, not necessary.

284: in the note: CH is what? Continuum hypothesis, right? If so, it has no connection to what is said in the paragraph. If not, you should make it clear what is the connection here.

285: is Hilbert's statement really related in anyway to von Neumann (von is lowercase, no?)? I never heard this relation and I thought it came much before von Neumann. Use some reference to this or remove this relation.

285 - 295: I don't see why this paragraph advances your arguments. Also, I think you need some references to argue these historical points (or else develop it more carefully -- which I don't recommend).

308: I think you could say the story of Frege and Russell is well-known. But as there is reference to attest this, it is better to use it. You can find the letters from and to Frege-Russell in the book "From Frege to Godel" by Jean van Heijenoort.

sections 2.1.1 and 2.1.2: I don't think these sections add value to the paper. They are reviews of the history of development of set theory that we can easily find in literature (e.g. Set theory and its philosophy by Potter). I could not identify an specific historical analysis that makes these sections valuable to be published in a journal paper. (FOR THIS REASON, I WILL NOT GIVE MORE SPECIFIC COMMENTS ON THESE SECTIONS)

583: the comment "self-inclusion is not self-membership" should be removed. This is a comment that adds value only to a beginner. This paper, talking about forcing and etc., is not directed to a beginner.

(I BELIEVE THIS PAPER STILL NEED SUBSTANTIAL DEVELOPMENT AND MUCH SHOULD BE REMOVED. SO I STOP HERE MAKING TOO MANY SPECIFIC COMMENTS FOR THEY WOULD BE TOO MANY AT THIS STAGE.)

667 - 675: very complicated paragraph. Also, why should one simply accept Aquinas commentary and, moreover, your interpretation of his commentary with respect to set theory. This needs to be much better developed to be convincing. (I don't see how you would argue that it CAN BE ONLY formalized in CT in 3.2., but if successful, it would be good.)

685 - 710: without proper development, this adds little value to the reader.

718 - 719: this claim is problematic in two respects: it depends on what you mean be "expressed" and "CT". If you mean expressed only in linguistic terms, than first order logic can also express ST -- but this is meaningless. If "expressed" is more substantial, you should say more about what CT you are referring to. Most CT axioms I know are only descriptive, meaning they only say the rules of objects and morphisms. And the existence of the categories usually being constructed in some set theory (often with large cardinals or classes). There are some developments of CT that incorporate existencial statements, but this is not standard; so you should be much more specific in this regard. Finally, what would you say if someone reply: "well, CT can also be expressed in ST terms?".

sections 2.2.1, 2.2.2, 2.2.3: too much on historical remarks that do not advance your argument. Most, if not all, that could read the other parts of your paper already know well enough this historical development (and, if not, would be satisfied in being directed to the relevant literature).

section 2.3 can be much smaller.

1337 - 1339: I don't understand this. I can guess what you mean, but you should make it much more clear.

1337 - 1375: it is not clear to me why you should mention this well known facts in order to introduce the history of CT. This should be summarized in one short paragraph.

1558: you say "Sets constitutes only one category". Ok, I can see that. But the foundational relation requires that CT can indeed gives this foundation. Otherwise, in order to have this category, one need to state the axioms of ST apart from the axioms of CT. (note that the other way around do not require that: one can define all the categories you list in ST and provide the existence of them).

section 3.2. Apart from some comments, this is an exposition of CT as one can find in any CT textbook. I suggest cutting all that and focusing on the additional comments.

1870: the claim "coalgebraic structures are becoming ever..." should be argued. I don't think every reader feel this way about it; and they will not be convinced simply because you say so.

1894: this quote goes more in the direction I usually see the foundational benefits of CT. It is the representation of universal phenomena in mathematics. It is not in the direction of "reducing" or "grounding existence".

1914: I am at this point in the text, and I still don't really know what is this benefits you promised many times...

1919: this comment about quantum theory is very vague and unnecessary. You talk too much about the part II. This is not the text we have at hand.

1933 - 1976: you promised to explain the benefits. But all I see here is the introduction of some traditional definitions of CT. It is fine to introduce something, but I cannot see what is your point in bringing all these definitions and results; how these things relate to the argument you want to make?

Section 3.3.2. At this point you say you are going to finally give us your argument. But then you keep referring to this Part II. At this point in the text you are still promising without delivering, as you are still saying things like "line 1988: I would like to complete Ehresmann's ...". Later you keep making more promises "In this way, I would like to give a concrete ..."

2003: "As an introduction... " at this point in the text?

2025: "In the II Part"... again you refer to part II without delivering in the promises for this part.

Author Response

Please, read the attached file that is effectively a new cover letter of the new submitted manuscript unifying the Part I and the Part II manuscripts of the old submission, taking into account reviewers suggestions. This cover letter contains also the answers to reviewers of the old Part I and Part II manuscripts. Thanks for your precious reviewing work.

Reviewer 2 Report

I am not competent to evaluate the philosophical interest of this paper. However, I find interesting to the application of modern mathematical structures, as Category Theory, Operator Algebras, and Quantum Field Theory concepts, to the philosophical context. 

Author Response

(The authors gave the same response as above.)

Reviewer 3 Report

This reader was not able to discern what the author was trying to say. 

Author Response

(The authors gave the same response as above.)

Round 2

Reviewer 1 Report

The paper is too long. I believe you should break it down before considering submitting this idea again.

I do not have further specific comments. 

Author Response

As to the different remarks requiring further improvements, I am not able unfortunately to do more for each of the points required, if not reducing the length of the paper. But this final requirement crashes with the above requirement of improvements, which on the other hand are too generic for suggesting a suitable revision of the paper. I am sorry.   

Reviewer 3 Report

I still do not understand what the author is attempting to say. 

Author Response

I am sorry if the referee was not able to understand what I tried to do. Probably there is some theoretical incompatibility between us. What I did was to give the paper to a philosopher of English mother language for improving the style of the paper. Essentially, she reduced the length of many paragraphs to make them more understandable. Perhaps for this reason, she understood the paper and what I tried to say (but this happened also to many colleagues to which I sent my manuscript). Again, I am very sorry.